# STEM: A Stochastic Two-Sided Momentum Algorithm Achieving Near-Optimal Sample and Communication Complexities for Federated Learning

**Prashant Khanduri**
University of Minnesota
khand095@umn.edu

**Pranay Sharma**
Carnegie Mellon University
pranaysh@andrew.cmu.edu

**Haibo Yang**
The Ohio State University
yang.5952@buckeyemail.osu.edu

**Mingyi Hong**\*
University of Minnesota
mhong@umn.edu

**Jia Liu**
The Ohio State University
liu@ece.osu.edu

**Ketan Rajawat**
Indian Institute of Technology Kanpur
ketan@iitk.ac.in

**Pramod K. Varshney**
Syracuse University
varshney@syr.edu

## Abstract

Federated Learning (FL) refers to the paradigm where multiple worker nodes (WNs) build a joint model by using local data. Despite extensive research, for a generic non-convex FL problem, it is not clear, how to choose the WNs' and the server's update directions, the minibatch sizes, and the number of local updates, so that the WNs use the minimum number of samples and communication rounds to achieve the desired solution. This work addresses the above question and considers a class of stochastic algorithms where the WNs perform a few local updates before communication. We show that when both the WN's and the server's directions are chosen based on certain stochastic momentum estimator, the algorithm requires $\tilde{\mathcal{O}}(\epsilon^{-3/2})$ samples and $\tilde{\mathcal{O}}(\epsilon^{-1})$ communication rounds to compute an $\epsilon$-stationary solution. To the best of our knowledge, this is the first FL algorithm that achieves such *near-optimal* sample and communication complexities simultaneously. Further, we show that there is a trade-off curve between the number of local updates and the minibatch sizes, on which the above sample and communication complexities can be maintained. Finally, we show that for the classical FedAvg (a.k.a. Local SGD, which is a momentum-less special case of the STEM), a similar trade-off curve exists, albeit with worse sample and communication complexities. Our insights on this trade-off provides guidelines for choosing the four important design elements for FL algorithms, the number of local updates, WNs' and server's update directions, and minibatch sizes to achieve the best performance.

## 1  Introduction

In Federated Learning (FL), multiple worker nodes (WNs) collaborate with the goal of learning a joint model, by only using local data. Therefore it has become popular for machine learning problems where datasets are massively distributed [1]. In FL, the data is often collected at or off-loaded to multiple WNs which in collaboration with a server node (SN) jointly aim to learn a centralized model

---

\*Corresponding Author: Mingyi Hong.

35th Conference on Neural Information Processing Systems (NeurIPS 2021).

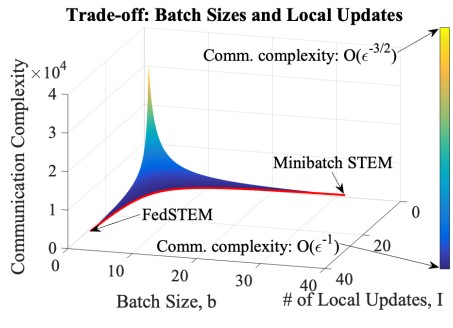

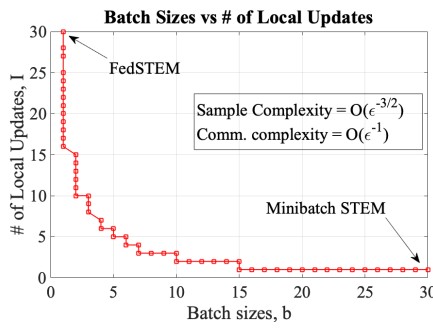

(a) Communication complexity.

(b) Minibatch sizes vs Local Updates.

Figure 1: The 3D surface in (a) plots the communication complexity of the proposed STEM for different minibatch sizes and number of local updates. The surface is generated such that each point represents STEM with a particular choice of $(b, I)$, so that it requires $\tilde{\mathcal{O}}(\epsilon^{-3/2})$ samples to achieve $\epsilon$-stationarity. Plot (b) shows the optimal trade off between the minibatch sizes and the number of local updates at each WN (i.e., achieving the lowest communication and sample complexities). Both plots are generated for an accuracy of $\epsilon = 10^{-3}$ and all the constants dependent on system parameters (variance of stochastic gradients, heterogeneity parameter, optimality gap, Lipschitz constants, etc.) are assumed to be $1$. Fed STEM is a special case of STEM where $\mathcal{O}(1)$ minibatch is used; Minibatch STEM is a special case of STEM where $\mathcal{O}(1)$ local updates are used.

[2, 3]. The local WNs share the computational load and since the data is local to each WN, FL also provides some level of data privacy [4]. A classical distributed optimization problem that $K$ WNs aim to solve:

$$\min_{x \in \mathbb{R}^d} \left\{ f(x) \coloneqq \frac{1}{K} \sum_{k=1}^{K} f^{(k)}(x) \coloneqq \frac{1}{K} \sum_{k=1}^{K} \mathbb{E}_{\xi^{(k)} \sim \mathcal{D}^{(k)}} [f^{(k)}(x; \xi^{(k)})] \right\}. \tag{1}$$

where $f^{(k)} : \mathbb{R}^d \to \mathbb{R}$ denotes the smooth (possibly non-convex) objective function and $\xi^{(k)} \sim \mathcal{D}^{(k)}$ represents the sample/s drawn from distribution $\mathcal{D}^{(k)}$ at the $k^{\text{th}}$ WN with $k \in [K]$. When the distributions $\mathcal{D}^{(k)}$ are different across the WNs, it is referred to as the *heterogeneous* data setting.

The optimization performance of non-convex FL algorithms is typically measured by the total number of samples accessed (cf. Definition 2.2) and the total rounds of communication (cf. Definition 2.3) required by each WN to achieve an $\epsilon$-stationary solution (cf. Definition 2.1). To minimize the sample and the communication complexities, FL algorithms rely on the following *four* key design elements: (i) the WNs' local model update directions, (ii) Minibatch size to compute each local direction, (iii) the number of local updates before WNs share their parameters, and (iv) the SN's update direction. How to find effective FL algorithms by (optimally) designing these parameters has received significant research interest recently.

**Contributions.** The main contributions of this work are listed below:

**1)** We propose the Stochastic Two-Sided Momentum (STEM) algorithm, that utilizes certain momentum-assisted stochastic gradient directions for *both* the WNs and SN updates. We show that there exists an *optimal* trade off between the minibatch sizes and number of local updates, such that on the trade-off curve STEM requires $\tilde{\mathcal{O}}(\epsilon^{-3/2})$[2] samples and $\tilde{\mathcal{O}}(\epsilon^{-1})$ communication rounds to reach an $\epsilon$-stationary solution; see Figure 1 for an illustration. These complexity results are the best achievable for first-order stochastic FL algorithms (under certain assumptions, cf. Assumption 1); see [5–8] and [9, 10], as well as Remark 1 of this paper for discussions regarding optimality. To the best of our knowledge, STEM is the first algorithm which – (i) *simultaneously* achieves the optimal sample and communication complexities for FL and (ii) can optimally trade off the minibatch sizes and the number of local updates.

**2)** A momentum-less special case of our STEM result further reveals some interesting insights of the classical FedAvg algorithm (a.k.a. the Local SGD) [11–13]. Specifically, we show that for FedAvg, there also exists a trade-off between the minibatch sizes and the number of local updates, such that it requires $\mathcal{O}(\epsilon^{-2})$ samples and $\mathcal{O}(\epsilon^{-3/2})$ communication rounds to achieve an $\epsilon$-stationary solution.

---

[2]The notation $\tilde{\mathcal{O}}(\cdot)$ hides the logarithmic factors.

| Algorithm | Work | Sample | Comm. | Minibatch ($b$) | Local Updates ($I$) /round |
|---|---|---|---|---|---|
| FedAvg$^\diamond$ | [12] /[14] | | $\mathcal{O}(\epsilon^{-3/2})$ | $\mathcal{O}(1)$ | $\mathcal{O}(\epsilon^{-1/2})$ |
| | [15]/[16] | $\mathcal{O}(\epsilon^{-2})$ | $\mathcal{O}(\epsilon^{-2})$ | $\mathcal{O}(1)$ | $\mathcal{O}(1)$ |
| | this work | | $\mathcal{O}(\epsilon^{-3/2})$ | $\mathcal{O}\big(\epsilon^{-\frac{2(1-\nu)}{(4-\nu)}}\big)$ | $\mathcal{O}\big(\epsilon^{-\frac{3\nu}{2(4-\nu)}}\big)$ |
| SCAFFOLD$^*$ | [15] | $\mathcal{O}(\epsilon^{-2})$ | $\mathcal{O}(\epsilon^{-2})$ | $\mathcal{O}(1)$ | $\mathcal{O}(1)$ |
| FedPD/FedProx$^\ddagger$ | [9]/ [10] | $\mathcal{O}(\epsilon^{-2})$ | $\mathcal{O}(\epsilon^{-1})$ | $\mathcal{O}(1)$ | $\mathcal{O}(\epsilon^{-1})$ |
| MIME$^\dagger$/FedGLOMO | [17]/[18] | $\mathcal{O}(\epsilon^{-3/2})$ | $\mathcal{O}(\epsilon^{-3/2})$ | $\mathcal{O}(1)$ | $\mathcal{O}(1)$ |
| STEM$^\diamond$ | this work | $\tilde{\mathcal{O}}(\epsilon^{-3/2})$ | $\tilde{\mathcal{O}}(\epsilon^{-1})$ | $\mathcal{O}\big(\epsilon^{-\frac{3(1-\nu)}{2(3-\nu)}}\big)$ | $\mathcal{O}\big(\epsilon^{-\frac{\nu}{(3-\nu)}}\big)$ |
| Fed STEM | | | | $\mathcal{O}(1)$ | $\mathcal{O}(\epsilon^{-1/2})$ |
| Minibatch STEM$^*$ | | | | $\mathcal{O}(\epsilon^{-1/2})$ | $\mathcal{O}(1)$ |

Table 1: Comparison of FedAvg and STEM with different FL algorithms for various choices of the minibatch sizes ($b$) and the number of per node local updates between two rounds of communication ($I$).
$^\diamond\nu \in [0, 1]$ trades off $b$ and $I$; $\nu = 1$ (resp. $\nu = 0$) uses multiple (resp. $\mathcal{O}(1)$) local updates and $\mathcal{O}(1)$ (resp. multiple) samples. Fed STEM and Minibatch STEM are two variants of the proposed STEM.
$^\ddagger$The data heterogeneity assumption is weaker than Assumption 2 (please see [9] for details).
$^\dagger$Requires bounded Hessian dissimilarity to model data heterogeneity across WNs.
$^*$Guarantees for Minibatch STEM with $I = 1$ and SCAFFOLD are independent of the data heterogeneity.

Collectively, our insights on the trade-offs provide practical guidelines for choosing different design elements for FL algorithms.

**Related Works.** FL algorithms were first proposed in the form of FedAvg [11], where the local update directions at each WN were chosen to be the SGD updates. Earlier works analyzed these algorithms in the homogeneous data setting [19–25], while many recent studies have focused on designing new algorithms to deal with heterogeneous data settings, as well as problems where the local loss functions are non-convex [9, 10, 12–16, 18, 26–32]. In [12], the authors showed that Parallel Restarted SGD (Local SGD or FedAvg [11]) achieves linear speed up while requiring $\mathcal{O}(\epsilon^{-2})$ samples and $\mathcal{O}(\epsilon^{-3/2})$ rounds of communication to reach an $\epsilon$-stationary solution. In [14], a Momentum SGD was proposed, which achieved the same sample and communication complexities as Parallel Restarted SGD [12], without requiring that the second moments of the gradients be bounded. Further, it was shown that under the homogeneous data setting, the communication complexity can be improved to $\mathcal{O}(\epsilon^{-1})$ while maintaining the same sample complexity. The works in [15, 16] conducted tighter analysis for FedAvg with partial WN participation with $\mathcal{O}(1)$ local updates and batch sizes. Their analysis showed that FedAvg's sample and communication complexities are both $\mathcal{O}(\epsilon^{-2})$. Additionally, SCAFFOLD was proposed in [15], which utilized variance reduction based local update directions [33] to achieve the same sample and communication complexities as FedAvg. Similarly, VRL-SGD proposed in [29] also utilized variance reduction and showed improved communication complexity of $\mathcal{O}(\epsilon^{-1})$, while requiring the same computations as FedAvg. Importantly, both SCAFFOLD and VRL-SGD's guarantees were independent of the data heterogeneity. The FedProx proposed in [10] used a penalty based method to improve the communication complexity of FedAvg (i.e., the Parallel Restarted and Momentum SGD [14, 12]) to $\mathcal{O}(\epsilon^{-1})$. FedProx used a gradient similarity assumption to model data heterogeneity which can be stringent for many practical applications. This assumption was relaxed by FedPD proposed in [9].

Recently, the works [17, 18] proposed to utilize hybrid momentum gradient estimators [7, 8]. The MIME algorithm [17] matched the optimal sample complexity (under certain smoothness assumptions) of $\mathcal{O}(\epsilon^{-3/2})$ of the centralized non-convex stochastic optimization algorithms [5–8]. Similarly, Fed-GLOMO [18] achieved the same sample complexity while employing compression to further reduce communication. Both MIME and Fed-GLOMO required $\mathcal{O}(\epsilon^{-3/2})$ communication rounds to achieve an $\epsilon$-stationary solution. Please see Table 1 for a summary of the above discussion.

The comparison of Local SGD (FedAvg) to Minibatch SGD for convex and strongly convex problems with homogeneous data setting was first conducted in [19] and later extended to heterogeneous setting in [13]. It was shown that Minibatch SGD almost always dominates the Local SGD. In contrast, it was shown in [24] that Local SGD dominates Minibatch SGD in terms of generalization performance. Although existing FL results are rich, but they are somehow ad hoc and there is a lack of principled

understanding of the algorithms. We note that the proposed STEM algorithmic framework provides a theoretical framework that unifies all existing FL results on sample and communication complexities.

**Notations.** The expected value of a random variable $X$ is denoted by $\mathbb{E}[X]$ and its expectation conditioned on an Event $A$ is denoted as $\mathbb{E}[X|\text{Event } A]$. We denote by $\mathbb{R}$ (and $\mathbb{R}^d$) the real line (and the $d$-dimensional Euclidean space). The set of natural numbers is denoted by $\mathbb{N}$. Given a positive integer $K \in \mathbb{N}$, we denote $[K] \triangleq \{1, 2, \ldots, K\}$. Notation $\| \cdot \|$ denotes the $\ell_2$-norm and $\langle \cdot, \cdot \rangle$ the Euclidean inner product. For a discrete set $\mathcal{B}$, $|\mathcal{B}|$ denotes the cardinality of the set. Uniform distribution over a discrete set $\{1, \ldots, T\}$ is denoted as $\mathcal{U}\{1, \ldots, T\}$.

## 2   Preliminaries

Before we proceed to the algorithms, we make the following assumptions about problem (1).

**Assumption 1** (Sample Gradient Lipschitz Smoothness)**.** The stochastic functions $f^{(k)}(\cdot, \xi^{(k)})$ with $\xi^{(k)} \sim \mathcal{D}^{(k)}$ for all $k \in [K]$, satisfy the mean squared smoothness property, i.e, we have

$$\mathbb{E}\|\nabla f^{(k)}(x; \xi^{(k)}) - \nabla f^{(k)}(y; \xi^{(k)})\|^2 \le L^2 \|x - y\|^2 \quad \text{for all } x, y \in \mathbb{R}^d.$$

**Assumption 2** (Unbiased gradient and Variance Bounds)**.** (i) Unbiased Gradient. The stochastic gradients computed at each WN are unbiased

$$\mathbb{E}[\nabla f^{(k)}(x; \xi^{(k)})] = \nabla f^{(k)}(x), \ \forall \ \xi^{(k)} \sim \mathcal{D}^{(k)}, \ \forall \ k \in [K].$$

(ii) Intra- and inter- node Variance Bound. The following bounds hold:

$$\mathbb{E}\|\nabla f^{(k)}(x; \xi^{(k)}) - \nabla f^{(k)}(x)\|^2 \le \sigma^2, \|\nabla f^{(k)}(x) - \nabla f^{(\ell)}(x)\|^2 \le \zeta^2, \ \forall \ \xi^{(k)} \sim \mathcal{D}^{(k)}, \forall k, \ell \in [K].$$

Note that Assumption 1 is stronger than directly assuming $f^{(k)}$'s are Lipschitz smooth (which we will refer to as the *averaged* gradient Lipschitz smooth condition), but it is still a rather standard assumption in SGD analysis. For example it has been used in analyzing centralized SGD algorithms such as SPIDER [5], SNVRG [6], STORM [7] (and many others) as well as in FL algorithms such as MIME [17] and Fed-GLOMO [18]. The second relation in Assumption 2-(ii) quantifies the data heterogeneity, and we call $\zeta > 0$ as the *heterogeneity parameter*. This is a typical assumption required to evaluate the performance of FL algorithms. If data distributions across individual WNs are identical, i.e., $\mathcal{D}^{(k)} = \mathcal{D}^{(\ell)}$ for all $k, \ell \in [K]$ then we have $\zeta = 0$.

Next, we define the $\epsilon$-stationary solution for non-convex optimization problems, as well as quantify the computation and communication complexities to achieve an $\epsilon$-stationary point.

**Definition 2.1** ($\epsilon$-Stationary Point)**.** A point $x$ is called $\epsilon$-stationary if $\|\nabla f(x)\|^2 \le \epsilon$. Moreover, a stochastic algorithm is said to achieve an $\epsilon$-stationary point in $t$ iterations if $\mathbb{E}[\|\nabla f(x_t)\|^2] \le \epsilon$, where the expectation is over the stochasticity of the algorithm until time instant $t$.

**Definition 2.2** (Sample complexity)**.** We assume an Incremental First-order Oracle (IFO) framework [34], where, given a sample $\xi^{(k)} \sim \mathcal{D}^{(k)}$ at the $k^{\text{th}}$ node and iterate $x$, the oracle returns $(f^{(k)}(x; \xi^{(k)}), \nabla f^{(k)}(x; \xi^{(k)}))$. Each access to the oracle is counted as a single IFO operation. We measure the sample (and computational) complexity in terms of the total number of calls to the IFO by all WNs to achieve an $\epsilon$-stationary point given in Definition 2.1.

**Definition 2.3** (Communication complexity)**.** We define a communication round as a one back-and-forth sharing of parameters between the WNs and the SN. Then the communication complexity is defined to be the total number of communication rounds between any WN and the SN required to achieve an $\epsilon$-stationary point given in Definition 2.1.

## 3   The STEM algorithm and the trade-off analysis

In this section, we discuss the proposed algorithm and present the main results. The key in the algorithm design is to carefully balance *all the four* design elements mentioned in Sec. 1, so that sufficient and useful progress can be made between two rounds of communication.

Let us discuss the key steps of STEM, listed in Algorithm 1. In Step 10, each node locally updates its model parameters using the local direction $d_t^k$, computed by using $b$ stochastic gradients at two

---
**Algorithm 1** The Stochastic Two-Sided Momemtum (STEM) Algorithm
---
1: **Input**: Parameters: $c > 0$, the number of local updates $I$, batch size $b$, stepsizes $\{\eta_t\}$.
2: **Initialize**: Iterate $x_1^{(k)} = \bar{x}_1 = \frac{1}{K}\sum_{k=1}^K x_1^{(k)}$, descent direction $d_1^{(k)} = \bar{d}_1 = \frac{1}{K}\sum_{k=1}^K d_1^{(k)}$
    with $d_1^{(k)} = \frac{1}{B}\sum_{\xi_1^{(k)} \in \mathcal{B}_1^{(k)}} \nabla f^{(k)}(x_1^{(k)}; \xi_1^{(k)})$ and $|\mathcal{B}_1^{(k)}| = B$ for $k \in [K]$.
3: Perform: $x_2^{(k)} = x_1^k - \eta_1 d_1^{(k)}, \ \forall \ k \in [K]$
4: **for** $t = 1$ to $T$ **do**
5:     **for** $k = 1$ to $K$ **do**                                               `#at the WN`
6:         $d_{t+1}^{(k)} = \frac{1}{b}\sum_{\xi_{t+1}^{(k)} \in \mathcal{B}_{t+1}^{(k)}} \nabla f^{(k)}(x_{t+1}^{(k)}; \xi_{t+1}^{(k)}) + (1 - a_{t+1})\Big(d_t^{(k)} - \frac{1}{b}\sum_{\xi_{t+1}^{(k)} \in \mathcal{B}_{t+1}^{(k)}} \nabla f^{(k)}(x_t^{(k)}; \xi_{t+1}^{(k)})\Big)$
        where we choose $|\mathcal{B}_{t+1}^{(k)}| = b$, and $a_{t+1} = c \cdot \eta_t^2$;
7:         **if** $t \bmod I = 0$ **then**                                    `#at the SN`
8:             $d_{t+1}^{(k)} = \bar{d}_{t+1} := \frac{1}{K}\sum_{k=1}^K d_{t+1}^{(k)}$
9:             $x_{t+2}^{(k)} := \bar{x}_{t+1} - \eta_{t+1}\bar{d}_{t+1} = \frac{1}{K}\sum_{k=1}^K x_{t+1}^{(k)} - \eta_{t+1}\bar{d}_{t+1}$   `#server-side momentum`
10:         **else** $x_{t+2}^{(k)} = x_{t+1}^{(k)} - \eta_{t+1}d_{t+1}^{(k)}$                         `#worker-side momentum`
11:         **end if**
12:     **end for**
13: **end for**
14: **Return**: $\bar{x}_a$ where $a \sim \mathcal{U}\{1, ..., T\}$.
---

consecutive iterates $x_{t+1}^{(k)}$ and $x_t^{(k)}$. After every $I$ local steps, the WNs share their current local models $\{x_{t+1}^{(k)}\}_{k=1}^K$ and directions $\{d_{t+1}^{(k)}\}_{k=1}^K$ with the SN. The SN aggregates these quantities, and performs a server-side momentum step, before returning $\bar{x}_{t+1}$ and $\bar{d}_{t+1}$ to all the WNs. Because both the WNs and the SN perform momentum based updates, we call the algorithm a stochastic *two-sided* momentum algorithm. The key parameters are: $b$ the minibatch size, $I$ the local update steps between two communication rounds, $\eta_t$ the stepsizes, and $a_t$ the momentum parameters.

One key technical innovation of our algorithm design is to identify the most suitable way to incorporate momentum based directions in FL algorithms. Although the momentum-based gradient estimator itself is not new and has been used in the literature before (see e.g., in [7, 8] and [17, 18] to improve the sample complexities of centralized and decentralized stochastic optimization problems, respectively), it is by no means clear if and how it can contribute to improve the communication complexity of FL algorithms. We show that in the FL setting, the local directions together with the local models have to be aggregated by the SN so to avoid being influenced too much by the local data. More importantly, besides the WNs, the SN also needs to perform updates using the (aggregated) momentum directions. Finally, such *two-sided* momentum updates have to be done carefully with the correct choice of minibatch size $b$, and the number of local updates $I$. Overall, it is the judicious choice of all these design elements that results in the optimal sample and communication complexities.

Next, we present the convergence guarantees of the STEM algorithm.

## 3.1 Main results: convergence guarantees for STEM

In this section, we analyze the performance of STEM. We first present our main result, and then provide discussions about a few parameter choices. In the next subsection, we discuss a special case of STEM related to the classical FedAvg and minibatch SGD algorithms.

**Theorem 3.1.** *Under the Assumptions 1 and 2, suppose the stepsize sequence is chosen as:*

$$\eta_t = \frac{\bar{\kappa}}{(w_t + \sigma^2 t)^{1/3}}, \tag{2}$$

*where we define :*

$$\bar{\kappa} = \frac{(bK)^{2/3}\sigma^{2/3}}{L}, \quad w_t = \max\left\{2\sigma^2, 4096L^3I^3\bar{\kappa}^3 - \sigma^2 t, \frac{c^3\bar{\kappa}^3}{4096L^3I^3}\right\}.$$

*Further, let us set $c = \frac{64L^2}{bK} + \frac{\sigma^2}{24\bar{\kappa}^3 LI} = L^2\left(\frac{64}{bK} + \frac{1}{24(bK)^2 I}\right)$, and set the initial batch size as $B = bI$; set the local updates $I$ and minibatch size $b$ as follows:*

$$I = \mathcal{O}\big((T/K^2)^{\nu/3}\big), \quad b = \mathcal{O}\big((T/K^2)^{1/2-\nu/2}\big) \tag{3}$$

*where $\nu$ satisfies $\nu \in [0,1]$. Then for* **STEM** *the following holds:*

*(i) For $\bar{x}_a$ chosen according to Algorithm 1, we have:*

$$\mathbb{E}\|\nabla f(\bar{x}_a)\|^2 = \mathcal{O}\left(\frac{f(\bar{x}_1) - f^*}{K^{2\nu/3}T^{1-\nu/3}}\right) + \tilde{\mathcal{O}}\left(\frac{\sigma^2}{K^{2\nu/3}T^{1-\nu/3}}\right) + \tilde{\mathcal{O}}\left(\frac{\zeta^2}{K^{2\nu/3}T^{1-\nu/3}}\right). \tag{4}$$

*(ii) For any $\nu \in [0,1]$, we have*

**Sample Complexity:** *The sample complexity of* **STEM** *is $\tilde{\mathcal{O}}(\epsilon^{-3/2})$. This implies that each WN requires at most $\tilde{\mathcal{O}}(K^{-1}\epsilon^{-3/2})$ gradient computations, thereby achieving linear speedup with the number of WNs present in the network.*

**Communication Complexity:** *The communication complexity of* **STEM** *is $\tilde{\mathcal{O}}(\epsilon^{-1})$.*

The proof of this result is relegated to the Supplemental Material. A few remarks are in order.

**Remark 1** (Near-Optimal sample and communication complexities). Theorem 3.1 suggests that when $I$ and $b$ are selected appropriately, then **STEM** achieves $\tilde{\mathcal{O}}(\epsilon^{-3/2})$ and $\tilde{\mathcal{O}}(\epsilon^{-1})$ sample and communication complexities. Taking them separately, these complexity bounds are the best achievable by the existing FL algorithms (upto logarithmic factors regardless of sample or batch Lipschitz smooth assumption) [35]; see Table 1. We note that the $\mathcal{O}(\epsilon^{-3/2})$ complexity is the best possible that can be achieved by centralized SGD with the sample Lipschitz gradient assumption; see [5]. On the other hand, the $\tilde{\mathcal{O}}(\epsilon^{-1})$ complexity bound is also *likely* to be the optimal, since in [9] the authors showed that even when the local steps use a class of (deterministic) first-order algorithms, $\mathcal{O}(\epsilon^{-1})$ is the best achievable communication complexity. The only difference is that [9] does not explicitly assume the inter-node variance bound (i.e., the second relation in Assumption 2-(ii)). We leave the precise characterization of the communication lower bound with inter-node variance as future work. $\square$

**Remark 2** (Large Batch Sizes and/or Local Updates). At first glance, it may seem that the requirement of **STEM** to compute large mini-batches and/or local updates (cf. Table 1) to achieve this (near) optimal performance is a drawback, however, we note that it is in fact an advantage of **STEM** that it allows the WNs to perform larger number of local updates (or compute large minibatches) without communicating often. This follows from the fact that irrespective of the number of local updates (or batch sizes) **STEM** achieves near-optimal communication complexity while attaining optimal *overall* sample complexity. Moreover, note that even with $b = I = \mathcal{O}(1)$ (i.e., $b$ and $I$ are chosen as constants), **STEM** achieves the same (optimal) sample and communication complexities as achieved by FedGLOMO [18] and MIME [17]. We further note that to the best of our knowledge the algorithms that achieve the communication complexity of $\mathcal{O}(\epsilon^{-1})$ either require the number of local updates or the batch-sizes that depend on the solution accuracy $\epsilon$. For example, FedProx [10], FedPD [9], and FedDyn [36] rely on solving the "local problems" to achieve an $\epsilon$-accuracy, which implies that the number of local updates (or the batch sizes) implicitly depends on the desired solution accuracy $\epsilon$, as is the case for **STEM**. Similarly, as shown in [12] and [14] the communication complexity of FedAvg and its momentum version can be improved from $\mathcal{O}(\epsilon^{-2})$ to $\mathcal{O}(\epsilon^{-3/2})$ when the number of local updates (or batch size) is chosen as $\mathcal{O}(\epsilon^{-1/2})$ (cf. Section 3.2 for a more detailed discussion). $\square$

**Remark 3** (The Optimal Batch Sizes and Local Updates Trade-off). The parameter $\nu \in [0,1]$ is used to balance the local minibatch sizes $b$, and the number of local updates $I$. Eqs. in (3) suggest that when $\nu$ increases from 0 to 1, $b$ decreases and $I$ increases. Specifically, if $\nu = 1$, then $b$ is a constant but $I = \mathcal{O}(T^{1/3}/K^{2/3})$. In this case, each WN chooses a small minibatch while executing multiple local updates, and **STEM** resembles a FedAvg (a.k.a. Local SGD) algorithm but with double-sided momentum update directions, and is referred to as Fed **STEM**. In contrast, if $\nu = 0$, then $b = \mathcal{O}(T^{1/2}/K)$ but $I$ is a constant. In this case, each WN chooses a large batch size while executing only a few, or even one, local updates, and **STEM** resembles the Minibatch SGD, but again with different update directions, and is referred to as Minibatch **STEM**. Such a trade-off can be seen in Fig. 1b. Due to space limitation, these two special cases will be precisely stated in the supplementary materials as corollaries of Theorem 3.1. $\square$

---

**Algorithm 2** The FedAvg Algorithm

---
1: **Input**: $\{\eta_t\}_{t=0}^T$; $I$, the # of local updates per communication round; $b$, the minibatch sizes.
2: **for** $t = 1$ to $T$ **do**
3:     **for** $k = 1$ to $K$ **do**
4:         $d_t^{(k)} = \frac{1}{b} \sum_{\xi_t^{(k)} \in \mathcal{B}_t^{(k)}} \nabla f^{(k)}(x_t^{(k)}; \xi_t^{(k)})$ with $|\mathcal{B}_t^{(k)}| = b$
5:         $x_{t+1}^{(k)} = x_t^{(k)} - \eta_t d_t^{(k)}$
6:         **if** $t \bmod I = 0$ **then**
7:             $x_{t+1}^{(k)} = \bar{x}_{t+1} = \frac{1}{K} \sum_{k=1}^K x_{t+1}^{(k)}$
8:         **end if**
9:     **end for**
10: **end for**
11: **Return**: $\bar{x}_a$ where $a \sim \mathcal{U}\{1, ..., T\}$.

---

**Remark 4** (The Sub-Optimal Batch Sizes and Local Updates Trade-off). From our proof (Theorem C.10 included in the supplemental material), we can see that STEM requires $\tilde{\mathcal{O}}\big( \max\big\{ (b \cdot I)\epsilon^{-1}, K^{-1}\epsilon^{-3/2} \big\} \big)$ samples and $\tilde{\mathcal{O}}\big( \max\big\{ \epsilon^{-1}, (b \cdot I)^{-1}K^{-1}\epsilon^{-3/2} \big\} \big)$ and communication rounds. According to the above expressions, if $b \cdot I$ increases beyond $\mathcal{O}(K^{-1}\epsilon^{-1/2})$, then the sample complexity will increase from the optimal $\tilde{\mathcal{O}}(\epsilon^{-3/2})$; otherwise, the optimal sample complexity $\tilde{\mathcal{O}}(\epsilon^{-3/2})$ is maintained. On the other hand, if $b \cdot I$ decreases beyond $\mathcal{O}(K^{-1}\epsilon^{-1/2})$, the communication complexity increases from $\tilde{\mathcal{O}}(\epsilon^{-1})$. For instance, if we choose $b = \mathcal{O}(1)$ and $I = \mathcal{O}(1)$ the communication complexity becomes $\tilde{\mathcal{O}}(\epsilon^{-3/2})$ while the optimal sample complexity $\tilde{\mathcal{O}}(\epsilon^{-3/2})$ is maintained. This trade-off is illustrated in Figure 1a, where we maintain the optimal sample complexity, while changing $b$ and $I$ to generate the trade-off surface. $\square$

**Remark 5** (Data Heterogeneity). The term $\tilde{\mathcal{O}}\big( \frac{\zeta^2}{K^{2\nu/3}T^{1-\nu/3}} \big)$ in the gradient bound (4) captures the effect of the heterogeneity of data across WNs, where $\zeta$ is the parameter characterizing the intra-node variance and has been defined in Assumption 2-(ii). Highly heterogeneous data with large $\zeta^2$ can adversely impact the performance of STEM. Note that such a dependency on $\zeta$ also appears in other existing FL algorithms, such as [9, 14, 18]. However, there is one special case of STEM that does not depend on the parameter $\zeta$. This is the case where $I = 1$, i.e., the minibatch SGD counterpart of STEM where only a single local iteration is performed between two communication rounds. We have the following corollary. $\square$

**Corollary 1** (Minibatch STEM). *Under Assumptions 1 and 2, and choose the algorithm parameters as in Theorem 3.1. At each WN, choose $I = 1$, $b = (T/K^2)^{1/2}$, and the initial batch size $B = b \cdot I$. Then STEM satisfies:*

*(i) For $\bar{x}_a$ chosen according to Algorithm 1, we have*

$$\mathbb{E}\|\nabla f(\bar{x}_a)\|^2 = \mathcal{O}\Big( \frac{f(\bar{x}_1) - f^*}{T} \Big) + \tilde{\mathcal{O}}\Big( \frac{\sigma^2}{T} \Big).$$

*(ii) Minibatch STEM achieves $\tilde{\mathcal{O}}(\epsilon^{-3/2})$ sample and $\tilde{\mathcal{O}}(\epsilon^{-1})$ communication complexity.*

Next, we show that FedAvg also exhibits a trade-off similar to that of STEM but with worse sample and communication complexities.

## 3.2 Special cases: The FedAvg algorithm

We briefly discuss another interesting special case of STEM, where the local momentum update is replaced by the conventional SGD (i.e., $a_t = 1, \ \forall \ t$), while the server does not perform the momentum update (i.e., $\bar{d}_t = 0, \forall \ t$). This is essentially the classical FedAvg algorithm, just that it balances the number of local updates $I$ and the minibatch size $b$. We show that this algorithm also exhibits a trade-off between $b$ and $I$ and on the trade-off curve it achieves $\mathcal{O}(\epsilon^{-2})$ sample complexity and $\mathcal{O}(\epsilon^{-3/2})$ communication complexity.

| Algorithm | Training Acc. | Testing Acc. |
|-----------|---------------|--------------|
| FedAvg | 78.2 | 74.1 |
| FedProx | 79.2 | 74.8 |
| FedDyn | 68.9 | 66.0 |
| SCAFFOLD | 71.9 | 74.0 |
| MIME | 82.6 | 76.8 |
| FedGLOMO | 76.1 | 72.8 |
| STEM | 80.1 | 78.8 |

(a) Mild heterogeneity, $b = 64$, and $I = 7$.

| Algorithm | Training Acc. | Testing Acc. |
|-----------|---------------|--------------|
| FedAvg | 73.6 | 75.4 |
| FedProx | 80.0 | 75.2 |
| FedDyn | 76.1 | 71.3 |
| SCAFFOLD | 72.5 | 73.7 |
| MIME | 61.5 | 58.6 |
| FedGLOMO | 10.0 | 10.0 |
| STEM | 81.1 | 78.5 |

(b) Moderate heterogeneity, $b = 8$, and $I = 61$.

Table 2: Training and testing accuracy of different algorithms on CIFAR-10 dataset for different batch-sizes, number of local updates, and heteregeneity settings.

**Theorem 3.2** (The FedAvg Algorithm). *Under Assumptions 1 and 2, suppose the stepsize is chosen as:* $\eta = \sqrt{\frac{bK}{T}}$; *Let us set:*

$$I = \mathcal{O}\big((T/K^3)^{\nu/4}\big), \quad b = \mathcal{O}\big((T/K^3)^{1/3-\nu/3}\big) \tag{5}$$

*where $\nu \in [0,1]$ is a constant. Then for FedAvg with $T \geq 81L^2I^2bK$, the following holds*

*(i) For $\bar{x}_a$ chosen according to Algorithm 2, we have*

$$\mathbb{E}\|\nabla f(\bar{x}_a)\|^2 = \mathcal{O}\left(\frac{f(\bar{x}_1) - f^*}{K^{\nu/2}T^{2/3-\nu/6}}\right) + \mathcal{O}\left(\frac{\sigma^2}{K^{\nu/2}T^{2/3-\nu/6}}\right) + \mathcal{O}\left(\frac{\zeta^2}{K^{\nu/2}T^{2/3-\nu/6}}\right).$$

*(ii) For any choice of $\nu \in [0,1]$ we have:*
   ***Sample Complexity:*** *The sample complexity of FedAvg is $\mathcal{O}(\epsilon^{-2})$. This implies that each WN requires at most $\mathcal{O}(K^{-1}\epsilon^{-2})$ gradient computations, thereby achieving linear speedup with the number of WNs in the network.*
   ***Communication Complexity:*** *The communication complexity of FedAvg is $\mathcal{O}(\epsilon^{-3/2})$.*

Note that the requirement on $T$ being lower bounded is only relevant for theoretical purposes, a similar requirement was also imposed in [14] to prove convergence. Again, the parameter $\nu \in [0,1]$ in the statement of Theorem 3.2 balances $I$ and $b$ at each WN while maintaining state-of-the-art sample and communication complexities; please see Table 1 for a comparison of those bounds with existing FedAvg bounds. For $\nu = 1$, FedAvg (cf. Theorem 3.2) reduces to FedAvg proposed in [12, 14] and for $\nu = 0$, the algorithm can be viewed as a large batch FedAvg with constant local updates [15, 16]. Note that similar to STEM, it is known that for $I = 1$, the Minibatch SGD's performance is independent of the heterogeneity parameter, $\zeta$ [13]. We also point out that if Algorithm 1 uses Nesterov's or Polyak's momentum [14] at local WNs instead of the recursive momentum estimator we get the same guarantees as in Theorem 3.2.

In summary, this section established that once the WN's and the SN's update directions (SGD in FedAvg and momentum based directions in STEM) are fixed, there exists a sequence of optimal choices of the number of local updates $I$, and the batch sizes $b$, which guarantees the best possible sample and communication complexities for the particular algorithm. The trade-off analysis presented in this section provides some useful guidelines for how to best select $b$ and $I$ in practice. Our subsequent numerical results will also verify that if $b$ or $I$ are not chosen judiciously, then the practical performance of the algorithms can degrade significantly.

## 4 Numerical results

In this section, we validate the proposed STEM algorithm and compare its performance with the de facto standard FedAvg [11], and the algorithms stated in Table 1. Note that instead of FedPD we include the performance comparison with FedDyn [36] since they are known to be very closely

| Algorithm | Training Acc. | Testing Acc. |
|---|---|---|
| FedAvg | 57.6 | 57.1 |
| FedProx | 59.1 | 58.5 |
| FedDyn | 51.2 | 51.3 |
| SCAFFOLD | 53.1 | 54.7 |
| MIME | 56.1 | 55.1 |
| FedGLOMO | 56.8 | 56.1 |
| STEM | 58.5 | 57.4 |

Table 3: Training and testing accuracy on CIFAR-10 dataset for high heterogeneity, $b = 128$ and $I = 6$.

| Algorithm | Training Acc. | Testing Acc. |
|---|---|---|
| FedAvg | 40.1 | 39.2 |
| FedProx | 43.5 | 43.2 |
| FedDyn | 43.7 | 43.2 |
| SCAFFOLD | 40.3 | 41.3 |
| MIME | 32.1 | 32.1 |
| FedGLOMO | 40.3 | 40.1 |
| STEM | 44.5 | 43.8 |

Table 4: Training and testing accuracy on Shakespeare dataset.

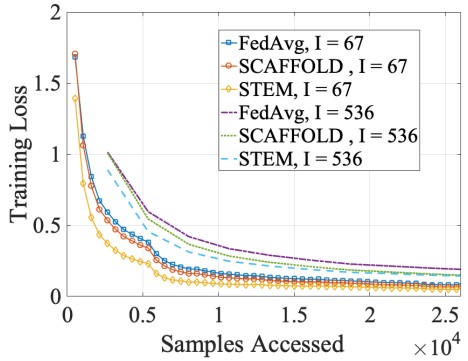 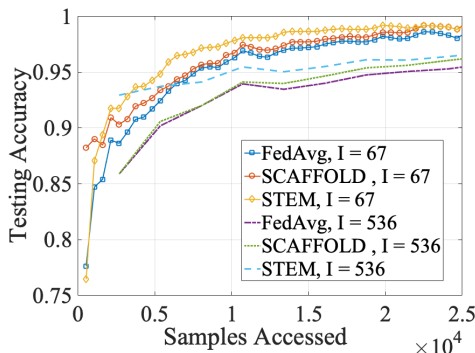

Figure 2: Training loss and the testing accuracy for classification on MNIST data set against the number of samples accessed at each WN for moderate heterogeneity setting with $b = 8$.

related. The goal of our experiments are three-fold: (1) To show that STEM performs on par, if not better, compared to other algorithms in different heterogeneity settings, (2) there are multiple ways to reach the desired solution accuracy, one can either choose a large batch size and perform only a few local updates or select a smaller batch size and perform multiple local updates, and finally, (3) if the local updates and the batch sizes are not chosen appropriately, the WNs might need to perform excessive computations to achieve the desired solution accuracy, thereby slowing down convergence.

**Data and Parameter Settings:** We compare the algorithms for image classification tasks on CIFAR-10 and MNIST data sets with 100 WNs, and for next character prediction task on Shakespeare dataset [37] with 143 WNs in the network. For both CIFAR-10 and MNIST, each WN implements a two-hidden-layer convolutional neural network (CNN) architecture followed by three linear layers for CIFAR-10 and two for MNIST. For CIFAR-10 (and MNIST) datset, we consider three settings with mild, moderate and high heterogeneity. For all the three settings, the data is partitioned into disjoint sets among the WNs. In the mild heterogeneity setting, the WNs have access to partitioned data from all the classes. In the moderate (resp. high) heterogeneity setting the data is partitioned such that each WN can access data from only 5 (resp. 2) out of 10 classes. For CIFAR-10 (resp. MNIST), each WN has access to 490 (resp. 540) samples for training and 90 (resp. 80) samples for testing purposes.

We also compare the performance of algorithms on a popular FL benchmarking dataset, Shakespeare dataset [37]. For this task, we adopt the settings from [10] and utilize a 2-Layer LSTM network with 100 hidden units and an 8-D embedding layer at each WN. Each WN has access to 3616 samples on average, and the samples are randomly split into an 80% training set and a 20% testing set. We randomly sample 10 nodes out of 143 for the training purpose. All the experiments are implemented on a single NVIDIA Quadro RTX 5000 GPU. More details are provided in appendix.

For the proposed STEM algorithm, recall that the step-size is $\eta_t = \bar{\kappa}/(w_t + \sigma^2 t)^{1/3}$ with momentum parameter defined as $a_t = c\eta_t^2$. The step-size is used to update the iterates while the momentum

parameter is used to construct the stochastic gradient estimate (cf. Algorithm 1 and Theorem 3.1). For the experiments, we set $w_t = \sigma^2 = 1$ and $c = \bar{c}/\bar{\kappa}^2$ and tune for $\bar{\kappa} \in [10^{-1}, 10^{-2}]$ for the CIFAR-10 dataset and for $\bar{\kappa} \in \{10^1, 10^0, 10^{-1}, 10^{-2}\}$ for the Shakespeare dataset. For both the datasets we tune for $\bar{c}$ in the range $[1, 10]$. For FedProx [10] and FedDyn [36] we choose the regularization constant to be 0.1. The momentum parameters for FedGLOMO [18] and MIME [17] are set based on the choices given in the respective papers. Specifically, for FedGLOMO we choose the parameter $\beta_k = 0.2$ and design the momentum gradient using a damping factor given in Appendix A.4 of FedGLOMO [18]. Moreover, for MIME we choose the momentum parameter as 0.9. For the rest of the algorithms (including FedAvg and SCAFFOLD), the step-size is tuned from the set $\{10^1, 10^0, 10^{-1}, 10^{-2}\}$.

**Discussion:** We evaluate the training and testing performance of STEM against multiple algorithms for different heterogeneity settings, minibatch sizes, and number of local updates. In Tables 2a, 2b and 3, we compare the training and testing accuracy of STEM to that of other algorithms on the CIFAR-10 dataset. Specific, heterogeneity settings, the choices of minibatches, and number of local updates are stated along with the tables. Note that STEM performs uniformly well under all the conditions. Moreover, note from Table 2b that FedGLOMO diverges once the number of local updates are high. Also, note from Table 3 that FedProx and STEM adapt well to high heterogeneity. Finally, with the next set of experiments we emphasize the importance of choosing $b$ and $I$ carefully. In Figure 2, we compare the training and testing performance of STEM, FedAvg and SCAFFOLD, against the number of samples accessed at each WN for the classification task on MNIST dataset with moderate heterogeneity. We fix $b = 8$ and conduct experiments under two settings, one with $I = 67$, and the other with $I = 536$ local updates at each WN. Note that although a large number of local updates might lead to fewer communication rounds but it can make the sample complexity extremely high as is demonstrated by Figure 2. For example, Figure 2 shows that to reach testing accuracy of $96 - 97\%$ with $I = 67$, STEM requires approximately $5000 - 6000$ samples, in contrast with $I = 536$ it requires more than $25000$ samples at each WN. Similar behavior can be observed if we fix $I > 1$ and increase the local batch sizes. This implies not choosing the local updates and the batch sizes judiciously might lead to increased sample complexity. Additional experiments are included in the supplementary material to further evaluate the performance of the proposed algorithms.

## Conclusion

In this work, we proposed a novel algorithm STEM, for distributed stochastic non-convex optimization with applications to FL. We showed that STEM reaches an $\epsilon$-stationary point with $\tilde{\mathcal{O}}(\epsilon^{-3/2})$ sample complexity while achieving linear speed-up with the number of WNs. Moreover, the algorithm achieves a communication complexity of $\tilde{\mathcal{O}}(\epsilon^{-1})$. We established a (optimal) trade-off that allows interpolation between varying choices of local updates and the batch sizes at each WN while maintaining (near optimal) sample and communication complexities. We showed that FedAvg (a.k.a LocalSGD) also exhibits a similar trade-off while achieving worse complexities. Our results provide guidelines to carefully choose the number of local updates, update directions, and minibatch sizes to achieve the best performance. The future directions of this work include developing lower bounds on communication complexity that establishes the tightness of the analysis conducted in this work.

## Acknowledgement

We thank the anonymous reviewers for their valuable comments and suggestions. The work of Prashant Khanduri and Mingyi Hong was supported by NSF grant CMMI-1727757, AFOSR grant 19RT0424 and ARO grant W911NF-19-1-0247. The work of Mingyi Hong was also supported by an IBM Faculty Research award. The work of Jia Liu has been supported in part by NSF grants CAREER CNS-2110259, CNS-2112471, CNS-2102233, CCF-2110252, ECCS-2140277, and a Google Faculty Research Award.

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
