Figure 3: Training loss and testing accuracy for classification on CIFAR-10 dataset against the number of communication rounds for mild heterogeneity setting with $b = 64$ and $I = 7$.

# Appendix

The organization of the Appendix is given below. In Appendix A we present the experimental details with along with additional numerical results on CIFAR-10 and MNIST datasets. Then in Appendix B, we present the proof of the convergence guarantees associated with the FedAvg algorithm given in Algorithm 2. Finally in Appendix C, we present the proof of the convergence for STEM given in Algorithm 1. Our proof is further divided into two parts, where in Appendix C.1 we present some useful lemmas, and the main body of the proof is given in Appendix C.2.

# A  Additional experiments

**Shakespeare Dataset.**    The Shakespeare dataset considers a classification problem of next character prediction with $80$ classes in total. We associate with each node a different speaking role (same setting as in [10]). We have a total of $143$ nodes with a total of $517,106$ samples that are unevenly split among $143$ nodes with each node having $3616$ samples on average. We randomly split the data at each node into an $80\%$ training set and a $20\%$ testing set. We randomly sample $10$ nodes out of $143$ for the training purpose. For this task, we utilize a 2-Layer LSTM network with $100$ hidden units and an 8-D embedding layer at each node. For each algorithm, we select a batch size of $128$ and tune for the rest of the hyperparameters as discussed in Section 4.

In this section, we present additional numerical results conducted for the classification task on CIFAR-10 and MNIST datasets. Here we focus on mild and moderate heterogeneity settings defined in Section 4. We compare the proposed STEM algorithm to two most popular baselines FedAvg and SCAFFOLD. We show that STEM outperforms both FedAvg and SCAFFOLD. Moreover, we corroborate the theoretical findings by showing that the algorithms converge in both cases, one where large batch sizes with a few local updates are used, and second where small batch sizes with a large number of local updates are employed. We utilize the same experimental settings as discussed in Section 4. Next, we present the results.

**Discussion.**    In Figures 4 and 3, we compare the training and testing performance of STEM with FedAvg and SCAFFOLD for CIFAR-10 dataset under mild heterogeneity setting. For Figure 4, we choose $b = 8$ and $I = 61$, whereas for Figure 3, we choose $b = 64$ and $I = 7$. We first note that for both cases STEM performs better than FedAvg and SCAFFOLD. Moreover, observe that for both settings, small batches with multiple local updates (Figure 4) and large batches with few local updates (Figure 3), the algorithms converge with approximately similar performance, corroborating the theoretical analysis (see Discussion in Section 1). Next, in Figure 5 we evaluate the performance of the proposed algorithms on CIFAR-10 with moderate heterogeneity setting for $b = 8$ and $I = 61$. We note that STEM outperforms FedAvg and SCAFFOLD in this setting as well.

Next, in Figure 6, we compare the training and testing performance of STEM and FedAvg against SCAFFOLD with the number of communication rounds. The figures are generated for local batch size of $b = 64$ while the number of local updates are chosen to be $I = 8$. The initial batch size, $B$, is chosen the same as $b$. Note form Figure 6 that STEM performs on par if not better than FedAvg under all settings. Moreover, STEM and FedAvg perform better than SCAFFOLD. In the next set

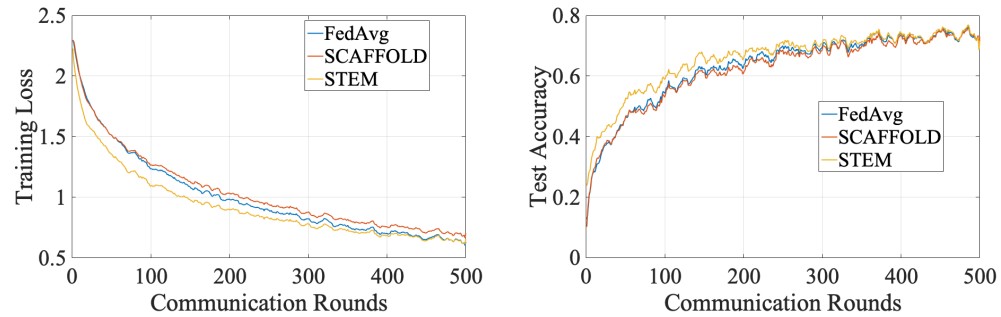

Figure 4: Training loss and testing accuracy for classification on CIFAR-10 dataset against the number of communication rounds for mild heterogeneity setting with $b = 8$ and $I = 61$.

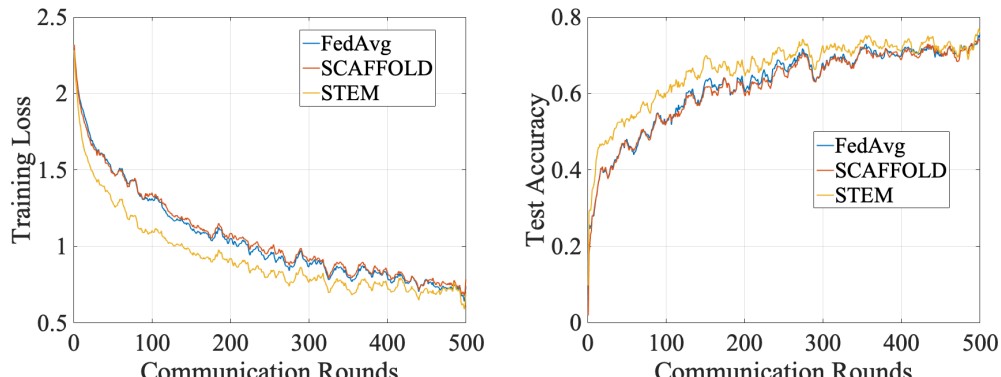

Figure 5: Training loss and testing accuracy for classification on CIFAR-10 dataset against the number of communication rounds for moderate heterogeneity setting with $b = 8$ and $I = 61$.

of simulations we trade the batch sizes for the number of local updates. Specifically, we choose $b = 8$ and $I = 67$, while choosing the same initial batch size, $B$, as $b$. The top two figures plot the performance of algorithms with mild heterogeneity setting while the lower two plot the performance for the moderate heterogeneity setting. Again note that STEM performs better than FedAvg and SCAFFOLD in both settings. Importantly, Figures 6 and 7 jointly imply that the algorithms can converge with acceptable performance while employing either "large batch sizes with few local updates" or "smaller batch sizes with multiple local updates".

Next, we present in detail the proofs of the results presented in the paper.

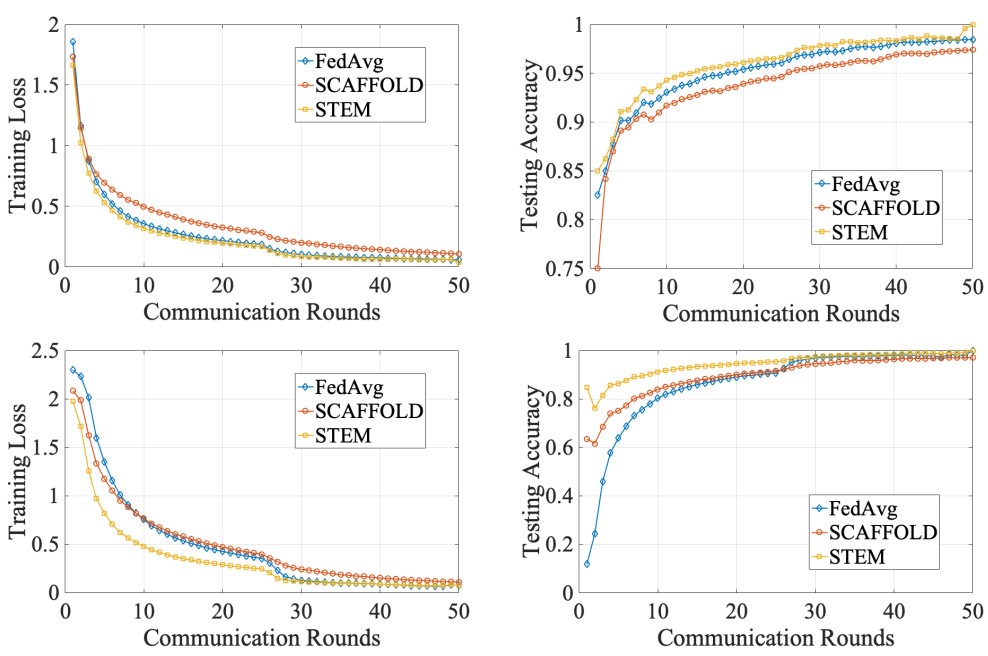

Figure 6: Training loss and the testing accuracy against the number of communication rounds with $b = 64$ and $I = 8$ for MNIST.

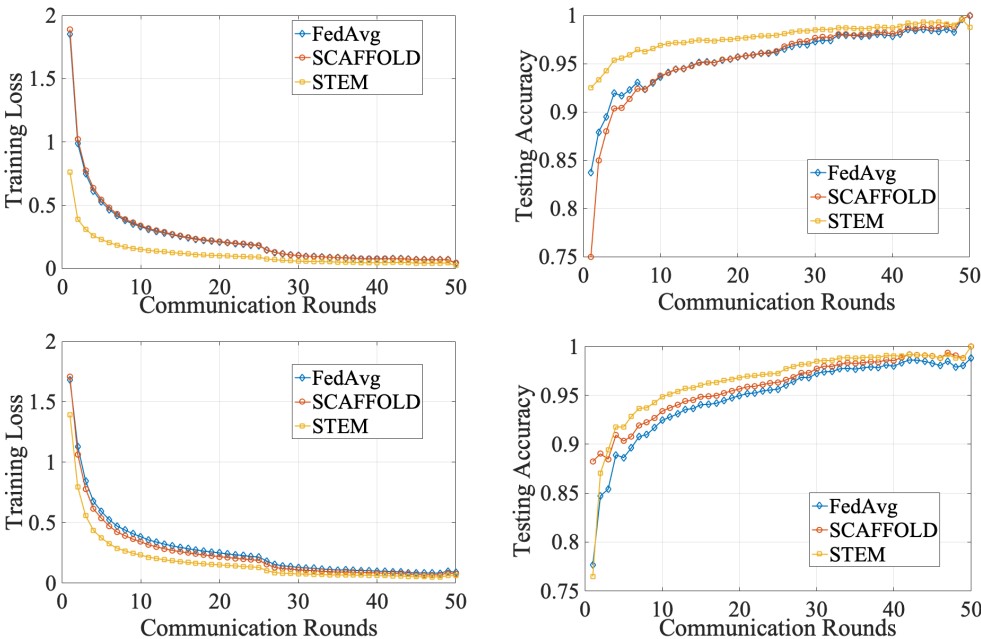

Figure 7: Training loss and the testing accuracy against the number of communication rounds with $b = 8$ and $I = 67$ for MNIST.

# B    Proofs of Convergence Guarantees for FedAvg

In this section, we present the proofs for the FedAvg algorithm. Before stating the proofs in detail we first present some preliminaries lemmas which shall be used for proving the main results of the paper. We first fix some notations:

We define $\bar{t}_s := sI + 1$ with $s \in [S]$. Note from Algorithm 2 that at $(s \times I)^{\text{th}}$ iteration, i.e., when $t \bmod I = 0$, the iterates, $\{x_t^{(k)}\}_{k=1}^K$ corresponding to $t = (\bar{t}_s)^{\text{th}}$ time instant are shared with the SN. We define the filtration $\mathcal{F}_t$ as the sigma algebra generated by iterates $x_1^{(k)}, x_2^{(k)}, \ldots, x_t^{(k)}$ as

$$\mathcal{F}_t = \sigma(x_1^{(k)}, x_2^{(k)}, \ldots, x_t^{(k)}, \text{ for all } k \in [K]).$$

Also, throughout the section we assume Assumptions 1 and 2 to hold.

## B.1 Preliminary Lemmas

**Lemma B.1.** *For $\bar{d}_t = \frac{1}{K} \sum_{k=1}^K d_t^{(k)}$ where $d_t^{(k)}$ for all $k \in [K]$ and $t \in [T]$ is chosen according to Algorithm 2, we have:*

$$\mathbb{E}\left\| \bar{d}_t - \frac{1}{K} \sum_{k=1}^K \nabla f^{(k)}(x_t^{(k)}) \right\|^2 \le \frac{\sigma^2}{bK},$$

*where the expectation is w.r.t the stochasticity of the the algorithm.*

*Proof.* Using the definition of $\bar{d}_t$ we have:

$$\mathbb{E}\left\| \bar{d}_t - \frac{1}{K} \sum_{k=1}^K \nabla f^{(k)}(x_t^{(k)}) \right\|^2$$

$$= \mathbb{E}\left\| \frac{1}{K} \sum_{k=1}^K \frac{1}{b} \sum_{\xi_t^{(k)} \in \mathcal{B}_t^{(k)}} \nabla f^{(k)}(x_t^{(k)}; \xi_t^{(k)}) - \frac{1}{K} \sum_{k=1}^K \nabla f^{(k)}(x_t^{(k)}) \right\|^2$$

$$= \mathbb{E}\left\| \frac{1}{K} \sum_{k=1}^K \frac{1}{b} \sum_{\xi_t^{(k)} \in \mathcal{B}_t^{(k)}} \left( \nabla f^{(k)}(x_t^{(k)}; \xi_t^{(k)}) - \nabla f^{(k)}(x_t^{(k)}) \right) \right\|^2$$

$$\overset{(a)}{=} \frac{1}{b^2 K^2} \sum_{k=1}^K \mathbb{E}\left\| \sum_{\xi_t^{(k)} \in \mathcal{B}_t^{(k)}} \left( \nabla f^{(k)}(x_t^{(k)}; \xi_t^{(k)}) - \nabla f^{(k)}(x_t^{(k)}) \right) \right\|^2$$

$$+ \frac{1}{b^2 K^2} \sum_{k \neq \ell} \mathbb{E}\left\langle \underbrace{\mathbb{E}\left[ \sum_{\xi_t^{(k)} \in \mathcal{B}_t^{(k)}} \left( \nabla f^{(k)}(x_t^{(k)}; \xi_t^{(k)}) - \nabla f^{(k)}(x_t^{(k)}) \right) \Big| \mathcal{F}_t \right]}_{=0}, \underbrace{\mathbb{E}\left[ \sum_{\xi_t^{(\ell)} \in \mathcal{B}_t^{(\ell)}} \left( \nabla f^{(\ell)}(x_t^{(\ell)}; \xi_t^{(\ell)}) - \nabla f^{(\ell)}(x_t^{(\ell)}) \right) \Big| \mathcal{F}_t \right]}_{=0} \right\rangle$$

$$\overset{(b)}{=} \frac{1}{b^2 K^2} \sum_{k=1}^K \sum_{\xi_t^{(k)} \in \mathcal{B}_t^{(k)}} \mathbb{E}\left\| \nabla f^{(k)}(x_t^{(k)}; \xi_t^{(k)}) - \nabla f^{(k)}(x_t^{(k)}) \right\|^2$$

$$+ \frac{1}{b^2 K^2} \sum_{k=1}^K \sum_{\xi_t^{(k)} \neq \zeta_t^{(k)}} \mathbb{E}\left\langle \underbrace{\mathbb{E}\left[ \nabla f^{(k)}(x_t^{(k)}; \xi_t^{(k)}) - \nabla f^{(k)}(x_t^{(k)}) | \mathcal{F}_t \right]}_{=0}, \underbrace{\mathbb{E}\left[ \nabla f^{(k)}(x_t^{(k)}; \zeta_t^{(k)}) - \nabla f^{(k)}(x_t^{(k)}) | \mathcal{F}_t \right]}_{=0} \right\rangle$$

$$\overset{(c)}{\le} \frac{\sigma^2}{bK},$$

where $(a)$ follows from Assumption 2 that given $\mathcal{F}_t$ we have: $\mathbb{E}\left[ \nabla f^{(k)}(x_t^{(k)}; \xi_t^{(k)}) \right] = \nabla f^{(k)}(x_t^{(k)})$, for all $k \in [K]$. Moreover, given $\mathcal{F}_t$ the samples $\xi_t^{(k)}$ and $\xi_t^{(\ell)}$ at the $k^{\text{th}}$ and the $\ell^{\text{th}}$ WNs are chosen uniformly randomly, and independent of each other for all $k, \ell \in [K]$ and $k \neq \ell$, therefore we have

$$\mathbb{E}\left[ \left\langle \sum_{\xi_t^{(k)} \in \mathcal{B}_t^{(k)}} \left( \nabla f^{(k)}(x_t^{(k)}; \xi_t^{(k)}) - \nabla f^{(k)}(x_t^{(k)}) \right), \sum_{\xi_t^{(\ell)} \in \mathcal{B}_t^{(\ell)}} \left( \nabla f^{(\ell)}(x_t^{(\ell)}; \xi_t^{(\ell)}) - \nabla f^{(\ell)}(\bar{x}_t) \right) \right\rangle \right]$$

$$= \mathbb{E}\left[ \left\langle \sum_{\xi_t^{(k)} \in \mathcal{B}_t^{(k)}} \underbrace{\mathbb{E}\left[ \nabla f^{(k)}(x_t^{(k)}; \xi_t^{(k)}) - \nabla f^{(k)}(x_t^{(k)}) | \mathcal{F}_t \right]}_{=0}, \sum_{\xi_t^{(\ell)} \in \mathcal{B}_t^{(\ell)}} \underbrace{\mathbb{E}\left[ \nabla f^{(\ell)}(x_t^{(\ell)}; \xi_t^{(\ell)}) - \nabla f^{(\ell)}(x_t^{(\ell)}) | \mathcal{F}_t \right]}_{=0} \right\rangle \right]$$

$$= 0.$$

The equality $(b)$ follows from the fact that $\xi_1^{(k)}$ and $\zeta_1^{(k)}$ for all $k \in [K]$ are chosen independently of each other. Then we conclude $(b)$ from an argument similar to that of $(a)$. Finally, $(c)$ results from the intra-node variance bound given in Assumption 2(ii).

Hence, the lemma is proved. $\qquad\square$

**Lemma B.2.** *For a finite sequence $x^{(k)} \in \mathbb{R}^d$ for $k \in [K]$ define $\bar{x} := \frac{1}{K} \sum_{k=1}^{K} x^{(k)}$, we then have*

$$\sum_{k=1}^{K} \|x^{(k)} - \bar{x}\|^2 \le \sum_{k=1}^{K} \|x^{(k)}\|^2.$$

*Proof.* Using the notation $\mathbf{x} = \left[ \left(x^{(1)}\right)^T, \left(x^{(2)}\right)^T, \dots, \left(x^{(K)}\right)^T \right]^T \in \mathbb{R}^{Kd}$, denoting $\mathbf{I}_d \in \mathbb{R}^{d \times d}$ and $\mathbf{I}_{Kd} \in \mathbb{R}^{Kd \times Kd}$ as identity matrices and representing $\mathbf{1} \in \mathbb{R}^K$ as the vector of all ones. We rewrite the left hand side of the statement as

$$\sum_{k=1}^{K} \|x^{(k)} - \bar{x}\|^2 = \left\| \mathbf{x} - \left( \mathbf{I} \otimes \frac{\mathbf{11}^T}{K} \right) \mathbf{x} \right\|^2$$

$$= \left\| \left( \mathbf{I}_{Kd} - \left( \mathbf{I}_d \otimes \frac{\mathbf{11}^T}{K} \right) \right) \mathbf{x} \right\|^2$$

$$\overset{(a)}{\le} \|\mathbf{x}\|^2 = \sum_{k=1}^{K} \|x^{(k)}\|^2,$$

where $(a)$ follows from the fact that the induced matrix norm $\left\| \mathbf{I}_{Kd} - \left( \mathbf{I}_d \otimes \frac{\mathbf{11}^T}{K} \right) \right\| \le 1$. $\qquad\square$

**Lemma B.3** (From [7]). *Let $a_0 > 0$ and $a_1, a_2, \dots, a_T \ge 0$. We have*

$$\sum_{t=1}^{T} \frac{a_t}{a_0 + \sum_{i=t}^{t} a_i} \le \ln \left( 1 + \frac{\sum_{i=1}^{t} a_i}{a_0} \right).$$

**Lemma B.4.** *For $X_1, X_2, \dots, X_n \in \mathbb{R}^d$, we have*

$$\|X_1 + X_2 + \dots + X_n\|^2 \le n\|X_1\|^2 + n\|X_2\|^2 + \dots + n\|X_n\|^2.$$

Next, we present the proof of Theorem 3.2. The proof follows in few steps which are discussed next.

## B.2 Proof of Main Results: FedAvg

**Lemma B.5** (Error Accumulation from Iterates). *For the choice of stepsize $\eta \le 1/9 \cdot L \cdot I$, the iterates $x_t^{(k)}$ for each $k \in [K]$ generated from Algorithm 2 satisfy:*

$$\sum_{t=1}^{T} \frac{1}{K} \sum_{k=1}^{K} \mathbb{E}\|x_t^{(k)} - \bar{x}_t\|^2 \le 3\eta^2 (I-1)\sigma^2 T + 5\eta^2 (I-1)^2 \zeta^2 T,$$

*where the expectation is w.r.t the stochasticity of the algorithm.*

*Proof.* Note from Algorithm 2 and the definition of $\bar{t}_s$ that at $t = \bar{t}_{s-1}$ with $s \in [S]$, $x_t^{(k)} = \bar{x}_t$, for all $k$. This implies

$$\frac{1}{K} \sum_{k=1}^{K} \|x_{\bar{t}_{s-1}}^{(k)} - \bar{x}_{\bar{t}_{s-1}}\|^2 = 0.$$

Therefore, the statement of the lemma holds trivially. Moreover, for $t \in [\bar{t}_{s-1} + 1, \bar{t}_s - 1]$, with $s \in [S]$, we have from Algorithm 2: $x_t^{(k)} = x_{t-1}^{(k)} - \eta d_{t-1}^{(k)}$, this implies that:

$$x_t^{(k)} = x_{\bar{t}_{s-1}}^{(k)} - \sum_{\ell=\bar{t}_{s-1}}^{t-1} \eta d_\ell^{(k)} \quad \text{and} \quad \bar{x}_t = \bar{x}_{\bar{t}_{s-1}} - \sum_{\ell=\bar{t}_{s-1}}^{t-1} \eta \bar{d}_\ell.$$

This implies that for $t \in [\bar{t}_{s-1} + 1, \bar{t}_s - 1]$, with $s \in [S]$ we have

$$\frac{1}{K}\sum_{k=1}^K \|x_t^{(k)} - \bar{x}_t\|^2 = \frac{1}{K}\sum_{k=1}^K \left\| x_{\bar{t}_{s-1}}^{(k)} - \bar{x}_{\bar{t}_{s-1}} - \Big( \sum_{\ell=\bar{t}_{s-1}}^{t-1} \eta d_\ell^{(k)} - \sum_{\ell=\bar{t}_{s-1}}^{t-1} \eta \bar{d}_\ell \Big) \right\|^2$$

$$\overset{(a)}{=} \frac{\eta^2}{K}\sum_{k=1}^K \left\| \sum_{\ell=\bar{t}_{s-1}}^{t-1} (d_\ell^{(k)} - \bar{d}_\ell) \right\|^2$$

$$\overset{(b)}{=} \frac{\eta^2}{K}\sum_{k=1}^K \left\| \sum_{\ell=\bar{t}_{s-1}}^{t-1} \Big( \frac{1}{b}\sum_{\xi_\ell^{(k)} \in \mathcal{B}_\ell^{(k)}} \nabla f^{(k)}(x_\ell^{(k)}; \xi_\ell^{(k)}) - \frac{1}{K}\sum_{j=1}^K \frac{1}{b}\sum_{\xi_\ell^{(j)} \in \mathcal{B}_\ell^{(j)}} \nabla f^{(j)}(x_\ell^{(j)}; \xi_\ell^{(j)}) \Big) \right\|^2$$

$$\overset{(c)}{\leq} \frac{2\eta^2}{K}\sum_{k=1}^K \left\| \sum_{\ell=\bar{t}_{s-1}}^{t-1} \left[ \Big( \frac{1}{b}\sum_{\xi_\ell^{(k)} \in \mathcal{B}_\ell^{(k)}} \nabla f^{(k)}(x_\ell^{(k)}; \xi_\ell^{(k)}) - \nabla f^{(k)}(x_\ell^{(k)}) \Big) \right. \right.$$

$$\left. \left. - \frac{1}{K}\sum_{j=1}^K \Big( \frac{1}{b}\sum_{\xi_\ell^{(j)} \in \mathcal{B}_\ell^{(j)}} \nabla f^{(j)}(x_\ell^{(j)}; \xi_\ell^{(j)}) - \nabla f^{(j)}(x_\ell^{(j)}) \Big) \right] \right\|^2$$

$$+ \frac{2\eta^2}{K}\sum_{k=1}^K \left\| \sum_{\ell=\bar{t}_{s-1}}^{t-1} \Big( \nabla f^{(k)}(x_\ell^{(k)}) - \frac{1}{K}\sum_{j=1}^K \nabla f^{(j)}(x_\ell^{(j)}) \Big) \right\|^2$$

$$\overset{(d)}{\leq} \frac{2\eta^2}{K}\sum_{k=1}^K \left\| \sum_{\ell=\bar{t}_{s-1}}^{t-1} \Big( \frac{1}{b}\sum_{\xi_\ell^{(k)} \in \mathcal{B}_\ell^{(k)}} \nabla f^{(k)}(x_\ell^{(k)}; \xi_\ell^{(k)}) - \nabla f^{(k)}(x_\ell^{(k)}) \Big) \right\|^2$$

$$+ \frac{2\eta^2}{K}\sum_{k=1}^K \left\| \sum_{\ell=\bar{t}_{s-1}}^{t-1} \Big( \nabla f^{(k)}(x_\ell^{(k)}) - \frac{1}{K}\sum_{j=1}^K \nabla f^{(j)}(x_\ell^{(j)}) \Big) \right\|^2, \quad (6)$$

where the equality $(a)$ follows from the fact that $x_{\bar{t}_{s-1}}^{(k)} = \bar{x}_{\bar{t}_{s-1}}$ for $t = \bar{t}_{s-1}$; $(b)$ results from the definition of the stochastic gradient employed by FedAvg in Algorithm 2; $(c)$ uses Lemma B.4 and $(d)$ follows from the application of Lemma B.2.

Taking expectation on both sides and let us next consider each term of (6) above separately, we have for any $k \in [K]$ from the first term of (6) above

$$\mathbb{E}\left\| \sum_{\ell=\bar{t}_{s-1}}^{t-1} \Big( \frac{1}{b}\sum_{\xi_\ell^{(k)} \in \mathcal{B}_\ell^{(k)}} \nabla f^{(k)}(x_\ell^{(k)}; \xi_\ell^{(k)}) - \nabla f^{(k)}(x_\ell^{(k)}) \Big) \right\|^2 \overset{(a)}{=} \sum_{\ell=\bar{t}_{s-1}}^{t-1} \mathbb{E}\left\| \frac{1}{b}\sum_{\xi_\ell^{(k)} \in \mathcal{B}_\ell^{(k)}} \nabla f^{(k)}(x_\ell^{(k)}; \xi_\ell^{(k)}) - \nabla f^{(k)}(x_\ell^{(k)}) \right\|^2$$

$$\overset{(b)}{=} \sum_{\ell=\bar{t}_{s-1}}^{t-1} \frac{1}{b^2}\sum_{\xi_\ell^{(k)} \in \mathcal{B}_\ell^{(k)}} \mathbb{E}\|\nabla f^{(k)}(x_\ell^{(k)}; \xi_\ell^{(k)}) - \nabla f^{(k)}(x_\ell^{(k)})\|^2$$

$$\overset{(c)}{\leq} \frac{(I-1)}{b}\sigma^2$$

$$\overset{(d)}{\leq} (I-1)\sigma^2, \quad (7)$$

where $(a)$ results from the fact that $\mathbb{E}\big[ \frac{1}{b}\sum_{\xi_\ell^{(k)} \in \mathcal{B}_\ell^{(k)}} \nabla f^{(k)}(x_\ell^{(k)}; \xi_\ell^{(k)}) - \nabla f^{(k)}(x_\ell^{(k)}) \big| \mathcal{F}_{\bar{\ell}} \big] = 0$ for any $\bar{\ell} < \ell$; $(b)$ uses the fact that $\mathbb{E}\big[ \nabla f^{(k)}(x_\ell^{(k)}; \xi_\ell^{(k)}) - \nabla f^{(k)}(x_\ell^{(k)}) \big| \nabla f^{(k)}(x_\ell^{(k)}; \zeta_\ell^{(k)}) - \nabla f^{(k)}(x_\ell^{(k)}) \big] = 0$ for samples $\xi_\ell^{(k)}, \zeta_\ell^{(k)} \sim \mathcal{D}^{(k)}$ chosen independent; $(c)$ utilizes intra-node variance bound in

Assumption 2(ii) and the fact that $(t-1) - \bar{t}_{s-1} \le I - 1$ for $t \in [\bar{t}_{s-1}+1, \bar{t}_s - 1]$; and finally, $(d)$ uses the fact that $b \ge 1$.

Next, we consider the second term of (6) for any $k \in [K]$, we have

$$\sum_{k=1}^{K} \mathbb{E} \left\| \sum_{\ell=\bar{t}_{s-1}}^{t-1} \left( \nabla f^{(k)}(x_\ell^{(k)}) - \frac{1}{K}\sum_{j=1}^{K} \nabla f^{(j)}(x_\ell^{(j)}) \right) \right\|^2$$

$$\overset{(a)}{\le} (I-1) \sum_{\ell=\bar{t}_{s-1}}^{t-1} \sum_{k=1}^{K} \mathbb{E} \left\| \nabla f^{(k)}(x_\ell^{(k)}) - \frac{1}{K}\sum_{j=1}^{K} \nabla f^{(j)}(x_\ell^{(j)}) \right\|^2$$

$$\overset{(b)}{\le} (I-1) \sum_{\ell=\bar{t}_{s-1}}^{t-1} \left[ 4\sum_{k=1}^{K} \mathbb{E}\|\nabla f^{(k)}(x_\ell^{(k)}) - \nabla f^{(k)}(\bar{x}_\ell)\|^2 + 4\sum_{k=1}^{K} \mathbb{E} \left\| \nabla f(\bar{x}_\ell) - \frac{1}{K}\sum_{j=1}^{K}\nabla f(x_\ell^{(j)}) \right\|^2 \right.$$

$$\left. + 2\sum_{k=1}^{K} \mathbb{E}\|\nabla f^{(k)}(\bar{x}_\ell) - \nabla f(\bar{x}_\ell)\|^2 \right]$$

$$\overset{(c)}{\le} (I-1) \sum_{\ell=\bar{t}_{s-1}}^{t-1} \left[ 8L^2\sum_{k=1}^{K} \mathbb{E}\|x_\ell^{(k)} - \bar{x}_\ell\|^2 + 2\sum_{k=1}^{K} \mathbb{E} \left\| \nabla f^{(k)}(\bar{x}_\ell) - \frac{1}{K}\sum_{j=1}^{K}\nabla f^{(j)}(\bar{x}_\ell) \right\|^2 \right]$$

$$\overset{(d)}{\le} 8L^2(I-1) \sum_{\ell=\bar{t}_{s-1}}^{t-1} \sum_{k=1}^{K} \mathbb{E}\|x_\ell^{(k)} - \bar{x}_\ell\|^2 + 2K(I-1)^2\zeta^2, \tag{8}$$

where $(a)$ utilizes the fact that $(t-1) - \bar{t}_{s-1} \le I-1$ for $t \in [\bar{t}_{s-1}+1, \bar{t}_s - 1]$; $(b)$ results from the application of Lemma B.4; $(c)$ follows from Assumption 1; and $(d)$ utilizes the inter-node variance Assumption 2 and the fact that $(t-1) - \bar{t}_{s-1} \le I-1$ for $t \in [\bar{t}_{s-1}+1, \bar{t}_s - 1]$.

Substituting (7) and (8) in (6) and taking expectation on both sides we get

$$\frac{1}{K}\sum_{k=1}^{K} \mathbb{E}\|x_t^{(k)} - \bar{x}_t\|^2 \le 2\eta^2(I-1)\sigma^2 + 4\eta^2(I-1)^2\zeta^2$$

$$+ 16L^2(I-1)\eta^2 \sum_{\ell=\bar{t}_{s-1}}^{t-1} \frac{1}{K}\sum_{k=1}^{k} \mathbb{E}\|x_\ell^{(k)} - \bar{x}_\ell\|^2.$$

Summing both sides from $t = \bar{t}_{s-1}$ to $\bar{t}_s - 1$, we get

$$\sum_{t=\bar{t}_{s-1}}^{\bar{t}_s-1} \frac{1}{K}\sum_{k=1}^{K} \mathbb{E}\|x_t^{(k)} - \bar{x}_t\|^2$$

$$\le 2\eta^2(I-1)\sigma^2 I + 4\eta^2(I-1)^2\zeta^2 I + 16L^2(I-1)\eta^2 \sum_{t=\bar{t}_{s-1}}^{\bar{t}_s-1} \sum_{\ell=\bar{t}_{s-1}}^{t-1} \frac{1}{K}\sum_{k=1}^{K} \mathbb{E}\|x_\ell^{(k)} - \bar{x}_\ell\|^2$$

$$\overset{(a)}{\le} 2\eta^2(I-1)\sigma^2 I + 4\eta^2(I-1)^2\zeta^2 I + 16L^2(I-1)\eta^2 \sum_{t=\bar{t}_{s-1}}^{\bar{t}_s-1} \sum_{\ell=\bar{t}_{s-1}}^{\bar{t}_s-1} \frac{1}{K}\sum_{k=1}^{K} \mathbb{E}\|x_\ell^{(k)} - \bar{x}_\ell\|^2$$

$$\overset{(b)}{\le} 2\eta^2(I-1)\sigma^2 I + 4\eta^2(I-1)^2\zeta^2 I + 16L^2(I-1)\eta^2 I \sum_{t=\bar{t}_{s-1}}^{\bar{t}_s-1} \frac{1}{K}\sum_{k=1}^{K} \mathbb{E}\|x_t^{(k)} - \bar{x}_t\|^2,$$

where $(a)$ uses that fact that $t \le \bar{t}_s - 1$; $(b)$ results from $t_s - t_{s-1} \le I$ for all $s \in [S]$. Finally, summing over $s \in [S]$ and using $T = SI$ we get

$$\sum_{t=1}^{T} \frac{1}{K}\sum_{k=1}^{K} \mathbb{E}\|x_t^{(k)} - \bar{x}_t\|^2 \le 2\eta^2(I-1)\sigma^2 T + 4\eta^2(I-1)^2\zeta^2 T + 16L^2 I^2\eta^2 \sum_{t=1}^{T} \frac{1}{K}\sum_{k=1}^{K} \mathbb{E}\|x_t^{(k)} - \bar{x}_t\|^2.$$

Rearranging the terms, we get

$$(1 - 16L^2 I^2 \eta^2) \sum_{t=1}^{T} \frac{1}{K} \sum_{k=1}^{K} \mathbb{E}\|x_t^{(k)} - \bar{x}_t\|^2 \leq 2\eta^2(I-1)\sigma^2 T + 4\eta^2(I-1)^2\zeta^2 T.$$

Finally, using the fact that $\eta \leq 1/9 \cdot L \cdot I$ we have $1 - 16L^2 I^2 \eta^2 \geq 4/5$. Multiplying, both sides by $5/4$ we get

$$\sum_{t=1}^{T} \frac{1}{K} \sum_{k=1}^{K} \mathbb{E}\|x_t^{(k)} - \bar{x}_t\|^2 \leq 3\eta^2(I-1)\sigma^2 T + 5\eta^2(I-1)^2\zeta^2 T.$$

Therefore, the lemma is proved. $\qquad\square$

**Lemma B.6** (Descent Lemma)**.** *For all $t \in [\bar{t}_{s-1}, \bar{t}_s - 1]$ and $s \in [S]$, with the choice of stepsizes $\eta \leq 1/9 \cdot L \cdot I$, the iterates generated by Algorithm 2 satisfy:*

$$\mathbb{E}f(\bar{x}_{t+1}) \leq \mathbb{E}f(\bar{x}_t) - \frac{\eta}{2}\mathbb{E}\|\nabla f(\bar{x}_t)\|^2 + \frac{\eta L^2}{2K}\sum_{k=1}^{K}\mathbb{E}\|x_t^{(k)} - \bar{x}_t\|^2 + \frac{\eta^2 L}{bK}\sigma^2,$$

*where the expectation is w.r.t the stochasticity of the algorithm.*

*Proof.* Using the smoothness of $f$ (Assumption 1) we have:

$$\mathbb{E}[f(\bar{x}_{t+1})]$$

$$\leq \mathbb{E}\left[f(\bar{x}_t) + \langle \nabla f(\bar{x}_t), \bar{x}_{t+1} - \bar{x}_t \rangle + \frac{L}{2}\|\bar{x}_{t+1} - \bar{x}_t\|^2\right]$$

$$\stackrel{(a)}{=} \mathbb{E}\left[f(\bar{x}_t) - \eta\langle \nabla f(\bar{x}_t), \bar{d}_t \rangle + \frac{\eta^2 L}{2}\|\bar{d}_t\|^2\right]$$

$$\stackrel{(b)}{=} \mathbb{E}\left[f(\bar{x}_t) - \eta\left\langle \nabla f(\bar{x}_t), \frac{1}{K}\sum_{k=1}^{K}\nabla f^{(k)}(x_t^{(k)}) \right\rangle + \frac{\eta^2 L}{2}\|\bar{d}_t\|^2\right]$$

$$\stackrel{(c)}{=} \mathbb{E}\left[f(\bar{x}_t) - \frac{\eta}{2}\left\|\frac{1}{K}\sum_{k=1}^{K}\nabla f^{(k)}(x_t^{(k)})\right\|^2 - \frac{\eta}{2}\|\nabla f(\bar{x}_t)\|^2 + \frac{\eta}{2}\left\|\nabla f(\bar{x}_t) - \frac{1}{K}\sum_{k=1}^{K}\nabla f^{(k)}(x_t^{(k)})\right\|^2 \right.$$

$$\left. + \eta^2 L\left\|\bar{d}_t - \frac{1}{K}\sum_{k=1}^{K}\nabla f^{(k)}(x_t^{(k)})\right\|^2 + \eta^2 L\left\|\frac{1}{K}\sum_{k=1}^{K}\nabla f^{(k)}(x_t^{(k)})\right\|^2\right]$$

$$\stackrel{(d)}{\leq} \mathbb{E}\left[f(\bar{x}_t) - \left(\frac{\eta}{2} - \eta^2 L\right)\left\|\frac{1}{K}\sum_{k=1}^{K}\nabla f^{(k)}(x_t^{(k)})\right\|^2 - \frac{\eta}{2}\|\nabla f(\bar{x}_t)\|^2 + \frac{\eta L^2}{2K}\sum_{k=1}^{K}\|x_t^{(k)} - \bar{x}_t\|^2 + \frac{\eta^2 L}{bK}\sigma^2\right]$$

$$\stackrel{(e)}{\leq} \mathbb{E}\left[f(\bar{x}_t) - \frac{\eta}{2}\|\nabla f(\bar{x}_t)\|^2 + \frac{\eta L^2}{2K}\sum_{k=1}^{K}\|x_t^{(k)} - \bar{x}_t\|^2 + \frac{\eta^2 L}{bK}\sigma^2\right],$$

where equality $(a)$ follows from the iterate update given in Step 5 of Algorithm 2; $(b)$ results from $\mathbb{E}[\nabla f^{(k)}(x_t^{(k)}; \xi_t^{(k)})|\mathcal{F}_t] = \nabla f^{(k)}(x_t^{(k)})$; $(c)$ uses $\langle a, b \rangle = \frac{1}{2}[\|a\|^2 + \|b\|^2 - \|a - b\|^2]$ and Lemma B.4; $(d)$ results from (9) below and Lemma B.1; and $(e)$ results from the stepsize choice of $\eta \leq 1/9LI$.

$$\mathbb{E}\left\|\frac{1}{K}\sum_{k=1}^{K}\left(\nabla f^{(k)}(x_t^{(k)}) - \nabla f^{(k)}(\bar{x}_t)\right)\right\|^2 \leq \frac{1}{K}\sum_{k=1}^{K}\mathbb{E}\|\nabla f^{(k)}(x_t^{(k)}) - \nabla f^{(k)}(\bar{x}_t)\|^2$$

$$\leq \frac{L^2}{K}\sum_{k=1}^{K}\mathbb{E}\|x_t^{(k)} - \bar{x}_t\|^2, \qquad (9)$$

where the first inequality follows from Lemma B.4, and the second follows from the $L$-Smoothness of $f^{(k)}(\cdot)$ (Assumption 1).

Hence, the lemma is proved. $\qquad\square$

### B.2.1 Proof of Theorem 3.2

The proof of Theorem 3.2 follows by replacing the choices of $b$ and $I$ given in (5) in the following result.

**Theorem B.7.** *Under Assumptions 1 and 2, with stepsize $\eta = \sqrt{\frac{bk}{T}}$. Then for $T \geq 81L^2I^2bK$ with any choice of minibatch sizes, $b \geq 1$, and number of local updates, $I \geq 1$, the iterates generated from Algorithm 2 satisfy*

$$\mathbb{E}\|\nabla f(\bar{x}_a)\|^2 \leq \frac{2(f(\bar{x}_t)) - f^*)}{(bk)^{1/2}T^{1/2}} + \frac{2L}{(bk)^{1/2}T^{1/2}}\sigma^2 + \frac{3L^2bK(I-1)}{T}\sigma^2 + \frac{5L^2bK(I-1)^2}{T}\zeta^2.$$

*Proof.* Summing the result of Lemma B.6 for $t = [T]$ and multiplying both sides by $2/\eta T$ we get

$$\frac{1}{T}\sum_{t=1}^{T}\mathbb{E}\|\nabla f(\bar{x}_t)\|^2 \leq \frac{2(f(\bar{x}_t) - f(\bar{x}_{t+1}))}{\eta T} + \frac{2\eta L}{bK}\sigma^2 + \frac{L^2}{T}\sum_{t=1}^{T}\frac{1}{K}\sum_{k=1}^{K}\mathbb{E}\|x_t^{(k)} - \bar{x}_t\|^2$$

$$\leq \frac{2(f(\bar{x}_t) - f^*)}{\eta T} + \frac{2\eta L}{bK}\sigma^2 + \frac{L^2}{T}\sum_{t=1}^{T}\frac{1}{K}\sum_{k=1}^{K}\mathbb{E}\|x_t^{(k)} - \bar{x}_t\|^2$$

where the second inequality uses $f(\bar{x}_{t-1}) \geq f^*$. Next, using Lemma B.5 we get

$$\frac{1}{T}\sum_{t=1}^{T}\mathbb{E}\|\nabla f(\bar{x}_t)\|^2 \leq \frac{2(f(\bar{x}_t) - f^*)}{\eta T} + \frac{2\eta L}{bK}\sigma^2 + 3L^2\eta^2(I-1)\sigma^2 + 5L^2\eta^2(I-1)^2\zeta^2.$$

Finally, using the definition of $\bar{x}_a$ from Algorithm 2 and the choice of $\eta = \sqrt{\frac{bK}{T}}$, we get

$$\mathbb{E}\|\nabla f(\bar{x}_a)\|^2 \leq \frac{2(f(\bar{x}_t) - f^*)}{(bK)^{1/2}T^{1/2}} + \frac{2L}{(bK)^{1/2}T^{1/2}}\sigma^2 + \frac{3L^2bK(I-1)}{T}\sigma^2 + \frac{5L^2bK(I-1)^2}{T}\zeta^2.$$

Therefore, we have the theorem. $\qquad\square$

Finally, substituting the choice of $I$ and $b$ given in (5) we get the statement of Theorem 3.2. Next two remarks characterize the behavior of FedAvg for two extreme choices of $I$ and $b$.

**Remark 6** (FedAvg: multiple local updates)**.** Choosing $\nu = 1$ in Theorem 3.2 implies $I = (T/b^3K^3)^{1/4}$ and $b = \mathcal{O}(1)$, we have

$$\mathbb{E}\|\nabla f(\bar{x}_a)\|^2 = \mathcal{O}\left(\frac{f(\bar{x}_1) - f^*}{K^{1/2}T^{1/2}}\right) + \mathcal{O}\left(\frac{\sigma^2}{K^{1/2}T^{1/2}}\right) + \mathcal{O}\left(\frac{\zeta^2}{K^{1/2}T^{1/2}}\right),$$

while the sample and communication complexities are still $\mathcal{O}(\epsilon^{-2})$ and $\mathcal{O}(\epsilon^{-3/2})$, respectively. Note that these are the same guarantees for FedAvg analyzed in [14, 20]. $\qquad\square$

**Remark 7** (FedAvg: large batch)**.** Choosing $\nu = 0$ in Theorem 3.2 implies $I = \mathcal{O}(1) > 1$ (we allow multiple local updates, i.e. $I > 1$) and $b = (T/I^4K^3)^{1/3}$, then we have

$$\mathbb{E}\|\nabla f(\bar{x}_a)\|^2 = \mathcal{O}\left(\frac{f(\bar{x}_1) - f^*}{T^{2/3}}\right) + \mathcal{O}\left(\frac{\sigma^2}{T^{2/3}}\right) + \mathcal{O}\left(\frac{\zeta^2}{T^{2/3}}\right).$$

while the sample and communication complexities are again $\mathcal{O}(\epsilon^{-2})$ and $\mathcal{O}(\epsilon^{-3/2})$, respectively. $\quad\square$

**Minibatch SGD:** When the parameters are shared after each local update, for such case we have $I = 1$ and for the choice of $b = \mathcal{O}(T/K)$ we have:

$$\mathbb{E}\|\nabla f(\bar{x}_a)\|^2 = \mathcal{O}\left(\frac{f(\bar{x}_1) - f^*}{T}\right) + \mathcal{O}\left(\frac{\sigma^2}{T}\right).$$

This implies that the sample and communication complexitiess are $\mathcal{O}(\epsilon^{-2})$ and $\mathcal{O}(\epsilon^{-1})$. Again, this result is independent of the heterogeniety parameter $\zeta$ (cf. Assumption 2) as the algorithm for $I = 1$ is essentially a centralized algorithm.

Next, we present the main result of the work presented in Theorem 3.1.

---

**Algorithm 3** The Stochastic Two-Sided Momemtum (STEM) Algorithm

---

1: **Input**: Parameters: $c > 0$, the number of local updates $I$, batch size $b$, stepsizes $\{\eta_t\}$.
2: **Initialize**: Iterate $x_1^{(k)} = \bar{x}_1 = \frac{1}{K}\sum_{k=1}^K x_1^{(k)}$, descent direction $d_1^{(k)} = \bar{d}_1 = \frac{1}{K}\sum_{k=1}^K d_1^{(k)}$
    with $d_1^{(k)} = \frac{1}{B}\sum_{\xi_1^{(k)}\in\mathcal{B}_1^{(k)}} \nabla f^{(k)}(x_1^{(k)};\xi_1^{(k)})$ and $|\mathcal{B}_1^{(k)}| = B$ for $k \in [K]$.
3: **Perform**: $x_2^{(k)} = x_1^k - \eta_1 d_1^{(k)}$, $\forall\, k \in [K]$
4: **for** $t = 1$ to $T$ **do**
5:     **for** $k = 1$ to $K$ **do**                                               `#at the WN`
6:         $d_{t+1}^{(k)} = \frac{1}{b}\sum_{\xi_{t+1}^{(k)}\in\mathcal{B}_{t+1}^{(k)}} \nabla f^{(k)}(x_{t+1}^{(k)};\xi_{t+1}^{(k)}) + (1-a_{t+1})\Big(d_t^{(k)} - \frac{1}{b}\sum_{\xi_{t+1}^{(k)}\in\mathcal{B}_{t+1}^{(k)}} \nabla f^{(k)}(x_t^{(k)};\xi_{t+1}^{(k)})\Big)$
    where we choose $|\mathcal{B}_{t+1}^{(k)}| = b$, and $a_{t+1} = c \cdot \eta_t^2$;
7:         **if** $t \bmod I = 0$ **then**                                    `#at the SN`
8:           $d_{t+1}^{(k)} = \bar{d}_{t+1} := \frac{1}{K}\sum_{k=1}^K d_{t+1}^{(k)}$
9:           $x_{t+2}^{(k)} := \bar{x}_{t+1} - \eta_{t+1}\bar{d}_{t+1} = \frac{1}{K}\sum_{k=1}^K x_{t+1}^{(k)} - \eta_{t+1}\bar{d}_{t+1}$ `#server-side momentum`
10:         **else** $x_{t+2}^{(k)} = x_{t+1}^{(k)} - \eta_{t+1}d_{t+1}^{(k)}$                        `#worker-side momentum`
11:         **end if**
12:     **end for**
13: **end for**
14: **Return**: $\bar{x}_a$ where $a \sim \mathcal{U}\{1,...,T\}$.

---

# C   Proofs of Convergence Guarantees for **STEM**

In this section we present the proofs of the convergence of STEM. First, we present some preliminary lemmas to be utilized throughout the proof. For reader's convenience here we restate the steps of the Algorithm 1 in Algorithm 3.

## C.1   Preliminary Lemmas

**Lemma C.1.** *Define* $\bar{e}_t := \bar{d}_t - \frac{1}{K}\sum_{k=1}^K \nabla f^{(k)}(x_t^{(k)})$, *then the iterates generated according to Algorithm 3 satisfy*

$$\mathbb{E}\left[\left\langle (1-a_t)\bar{e}_{t-1}, \frac{1}{K}\sum_{k=1}^K \frac{1}{b}\sum_{\xi_t^{(k)}\in\mathcal{B}_t^{(k)}} \left[\left(\nabla f^{(k)}(x_t^{(k)};\xi_t^{(k)}) - \nabla f^{(k)}(x_t^{(k)})\right)\right.\right.\right.$$
$$\left.\left.\left. - (1-a_t)\left(\nabla f^{(k)}(x_{t-1}^{(k)};\xi_t^{(k)}) - \nabla f^{(k)}(x_{t-1}^{(k)})\right)\right]\right\rangle\right] = 0,$$

*where the expectation is w.r.t. the stochasticity of the algorithm.*

*Proof.* Note that, given the filtration

$$\mathcal{F}_t = \sigma(x_1^{(k)}, x_2^{(k)}, \ldots, x_t^{(k)}, d_1^{(k)}, d_2^{(k)}, \ldots, d_{t-1}^{(k)} \text{ for all } k \in [K]),$$

the gradient error term, $\bar{e}_{t-1}$, is fixed. The only randomness in the left hand side of the statement of the Lemma is with respect to $\xi_t^{(k)}$, for all $k \in [K]$. This implies that we can write it as

$$\mathbb{E}\left[\left\langle (1-a_t)\bar{e}_{t-1}, \frac{1}{K}\sum_{k=1}^K \frac{1}{b}\sum_{\xi_t^{(k)}\in\mathcal{B}_t^{(k)}} \left[\left(\nabla f^{(k)}(x_t^{(k)};\xi_t^{(k)}) - \nabla f^{(k)}(x_t^{(k)})\right)\right.\right.\right.$$
$$\left.\left.\left. - (1-a_t)\left(\nabla f^{(k)}(x_{t-1}^{(k)};\xi_t^{(k)}) - \nabla f^{(k)}(x_{t-1}^{(k)})\right)\right]\right\rangle\right]$$

$$= \mathbb{E}\left[\left\langle (1-a_t)\bar{e}_{t-1}, \frac{1}{K}\sum_{k=1}^{K}\mathbb{E}\left[\frac{1}{b}\sum_{\xi_t^{(k)}\in\mathcal{B}_t^{(k)}}\left[\left(\nabla f^{(k)}(x_t^{(k)};\xi_t^{(k)}) - \nabla f^{(k)}(x_t^{(k)})\right)\right.\right.\right.\right.$$
$$\left.\left.\left.\left. - (1-a_t)\left(\nabla f^{(k)}(x_{t-1}^{(k)};\xi_t^{(k)}) - \nabla f^{(k)}(x_{t-1}^{(k)})\right)\right]\Big|\mathcal{F}_t\right]\right\rangle\right].$$

The result then follows from the fact that $\xi_t^{(k)}$ is chosen uniformly randomly at each $k \in [K]$, and we have from (Assumption 2) that: $\mathbb{E}\left[\nabla f^{(k)}(x_t^{(k)};\xi_t^{(k)})\right] = \nabla f^{(k)}(x_t^{(k)})$. This implies we have

$$\mathbb{E}\left[\frac{1}{b}\sum_{\xi_t^{(k)}\in\mathcal{B}_t^{(k)}}\left[\left(\nabla f^{(k)}(x_t^{(k)};\xi_t^{(k)}) - \nabla f^{(k)}(x_t^{(k)})\right) - (1-a_t)\left(\nabla f^{(k)}(x_{t-1}^{(k)};\xi_t^{(k)}) - \nabla f^{(k)}(x_{t-1}^{(k)})\right)\right]\Big|\mathcal{F}_t\right] = 0$$

for all $k \in [K]$.

Therefore the lemma is proved. $\qquad\square$

**Lemma C.2.** *For $k, \ell \in [K]$ with $k \neq \ell$, the iterates generated according to Algorithm 3 satisfy*

$$\mathbb{E}\left[\left\langle \sum_{\xi_t^{(k)}\in\mathcal{B}_t^{(k)}}\left[\left(\nabla f^{(k)}(x_t^{(k)};\xi_t^{(k)}) - \nabla f^{(k)}(x_t^{(k)})\right) - (1-a_t)\left(\nabla f^{(k)}(x_{t-1}^{(k)};\xi_t^{(k)}) - \nabla f^{(k)}(x_{t-1}^{(k)})\right)\right],\right.\right.$$
$$\left.\left. \sum_{\xi_t^{(\ell)}\in\mathcal{B}_t^{(\ell)}}\left[\left(\nabla f^{(\ell)}(x_t^{(\ell)};\xi_t^{(\ell)}) - \nabla f^{(\ell)}(x_t^{(\ell)})\right) - (1-a_t)\left(\nabla f^{(\ell)}(x_{t-1}^{(\ell)};\xi_t^{(\ell)}) - \nabla f^{(\ell)}(x_{t-1}^{(\ell)})\right)\right]\right\rangle\right] = 0$$

*Proof.* Again note from the fact that conditioned on $\mathcal{F}_t$ the batches $\mathcal{B}_t^{(k)}$ and $\mathcal{B}_t^{(\ell)}$ for all $k, \ell \in [K]$ with $k \neq \ell$ across WNs are chosen independently of each other. Therefore, we have

$$\mathbb{E}\left[\left\langle \sum_{\xi_t^{(k)}\in\mathcal{B}_t^{(k)}}\left[\left(\nabla f^{(k)}(x_t^{(k)};\xi_t^{(k)}) - \nabla f^{(k)}(x_t^{(k)})\right) - (1-a_t)\left(\nabla f^{(k)}(x_{t-1}^{(k)};\xi_t^{(k)}) - \nabla f^{(k)}(x_{t-1}^{(k)})\right)\right],\right.\right.$$
$$\left.\left. \sum_{\xi_t^{(\ell)}\in\mathcal{B}_t^{(\ell)}}\left[\left(\nabla f^{(\ell)}(x_t^{(\ell)};\xi_t^{(\ell)}) - \nabla f^{(\ell)}(x_t^{(\ell)})\right) - (1-a_t)\left(\nabla f^{(\ell)}(x_{t-1}^{(\ell)};\xi_t^{(\ell)}) - \nabla f^{(\ell)}(x_{t-1}^{(\ell)})\right)\right]\right\rangle\right]$$
$$= \mathbb{E}\left[\left\langle \mathbb{E}\left[\sum_{\xi_t^{(k)}\in\mathcal{B}_t^{(k)}}\left[\left(\nabla f^{(k)}(x_t^{(k)};\xi_t^{(k)}) - \nabla f^{(k)}(x_t^{(k)})\right) - (1-a_t)\left(\nabla f^{(k)}(x_{t-1}^{(k)};\xi_t^{(k)}) - \nabla f^{(k)}(x_{t-1}^{(k)})\right)\right]\Big|\mathcal{F}_t\right],\right.\right.$$
$$\left.\left. \mathbb{E}\left[\sum_{\xi_t^{(\ell)}\in\mathcal{B}_t^{(\ell)}}\left[\left(\nabla f^{(\ell)}(x_t^{(\ell)};\xi_t^{(\ell)}) - \nabla f^{(\ell)}(x_t^{(\ell)})\right) - (1-a_t)\left(\nabla f^{(\ell)}(x_{t-1}^{(\ell)};\xi_t^{(\ell)}) - \nabla f^{(\ell)}(x_{t-1}^{(\ell)})\right)\right]\Big|\mathcal{F}_t\right]\right\rangle\right].$$

The result then follows from the fact that $\xi_t^{(k)}$ is chosen uniformly randomly across $k \in [K]$ and we have from the unbiased gradient Assumption 2 that: $\mathbb{E}\left[\nabla f^{(k)}(x_t^{(k)};\xi_t^{(k)})\right] = \nabla f^{(k)}(x_t^{(k)})$. This implies we have

$$\mathbb{E}\left[\sum_{\xi_t^{(k)}\in\mathcal{B}_t^{(k)}}\left[\left(\nabla f^{(k)}(x_t^{(k)};\xi_t^{(k)}) - \nabla f^{(k)}(x_t^{(k)})\right) - (1-a_t)\left(\nabla f^{(k)}(x_{t-1}^{(k)};\xi_t^{(k)}) - \nabla f^{(k)}(x_{t-1}^{(k)})\right)\right]\Big|\mathcal{F}_t\right] = 0$$

for all $k \in [K]$.

Therefore, the lemma is proved. $\qquad\square$

**Lemma C.3.** *For $\bar{e}_1 := \bar{d}_1 - \frac{1}{K}\sum_{k=1}^{K}\nabla f^{(k)}(x_1^{(k)})$ where $\bar{d}_1$ chosen according to Algorithm 3, we have:*

$$\mathbb{E}\|\bar{e}_1\|^2 \leq \frac{\sigma^2}{KB}.$$

*Proof.* The proof follows from an argument similar to that of Lemma B.1

$\square$

Next, using the preliminary lemmas developed in this section we prove the main results of the work.

## C.2   Proof of Main Results: **STEM**

In this section, we utilize the results developed in earlier sections to derive the main result of the paper presented in Section 3.1. Throughout the section we assume Assumptions 1 and 2 to hold. Before proceeding, we first define some notations.

We define $\bar{t}_s \coloneqq sI + 1$ with $s \in [S]$. Note from Algorithm 3 that at $(s \times I)^{\text{th}}$ iteration, i.e., when $t \bmod I = 0$, the descent directions, $\{d_t^{(k)}\}_{k=1}^K$, corresponding to $t = (\bar{t}_s)^{\text{th}}$ time instant are shared with the SN. At the same time instant, the iterates, $\{x_t^{(k)}\}_{k=1}^K$ are also shared and the SN performs the "server side momentum step" (cf. Step 9 of Algorithm 3).

### C.2.1   Proof of Descent Lemma

In the first step, we bound the error accumulation via the iterates generated by Algorithm 3.

**Lemma C.4** (Error Accumulation from Iterates). *For each $t \in [\bar{t}_{s-1}, \bar{t}_s - 1]$ and $s \in [S]$, the iterates $x_t^{(k)}$ for each $k \in [K]$ generated from Algorithm 3 satisfy:*

$$\sum_{k=1}^K \mathbb{E}\|x_t^{(k)} - \bar{x}_t\|^2 \le (I-1) \sum_{\ell=\bar{t}_{s-1}}^t \eta_\ell^2 \sum_{k=1}^K \mathbb{E}\|d_\ell^{(k)} - \bar{d}_\ell\|^2,$$

*where the expectation is w.r.t the stochasticity of the algorithm.*

*Proof.* Note from Algorithm 3 and the definition of $\bar{t}_s$ that at $t = \bar{t}_{s-1}$ with $s \in [S]$, $x_t^{(k)} = \bar{x}_t$, for all $k$. This implies

$$\sum_{k=1}^K \|x_{\bar{t}_{s-1}}^{(k)} - \bar{x}_{\bar{t}_{s-1}}\|^2 = 0.$$

Therefore, the statement of the lemma holds trivially. Moreover, for $t \in [\bar{t}_{s-1} + 1, \bar{t}_s - 1]$, with $s \in [S]$, we have from Algorithm 3: $x_t^{(k)} = x_{t-1}^{(k)} - \eta_{t-1} d_{t-1}^{(k)}$, this implies that:

$$x_t^{(k)} = x_{\bar{t}_{s-1}}^{(k)} - \sum_{\ell=\bar{t}_{s-1}}^{t-1} \eta_\ell d_\ell^{(k)} \quad \text{and} \quad \bar{x}_t = \bar{x}_{\bar{t}_{s-1}} - \sum_{\ell=\bar{t}_{s-1}}^{t-1} \eta_\ell \bar{d}_\ell.$$

This implies that for $t \in [\bar{t}_{s-1} + 1, \bar{t}_s - 1]$, with $s \in [S]$ we have

$$\sum_{k=1}^K \|x_t^{(k)} - \bar{x}_t\|^2 = \sum_{k=1}^K \left\| x_{\bar{t}_{s-1}}^{(k)} - \bar{x}_{\bar{t}_{s-1}} - \left( \sum_{\ell=\bar{t}_{s-1}}^{t-1} \eta_\ell d_\ell^{(k)} - \sum_{\ell=\bar{t}_{s-1}}^{t-1} \eta_\ell \bar{d}_\ell \right) \right\|^2$$

$$\stackrel{(a)}{=} \sum_{k=1}^K \left\| \sum_{\ell=\bar{t}_{s-1}}^{t-1} \left( \eta_\ell d_\ell^{(k)} - \eta_\ell \bar{d}_\ell \right) \right\|^2$$

$$\stackrel{(b)}{\le} (I-1) \sum_{\ell=\bar{t}_{s-1}}^{t-1} \eta_\ell^2 \sum_{k=1}^K \|d_\ell^{(k)} - \bar{d}_\ell\|^2$$

$$\le (I-1) \sum_{\ell=\bar{t}_{s-1}}^t \eta_\ell^2 \sum_{k=1}^K \|d_\ell^{(k)} - \bar{d}_\ell\|^2,$$

where the equality $(a)$ follows from the fact that $x_{\bar{t}_{s-1}}^{(k)} = \bar{x}_{\bar{t}_{s-1}}$ and inequality $(b)$ uses the Lemma B.4 along with the fact that we have $d_t^{(k)} = \bar{d}_t$ for $t = \bar{t}_{s-1}$.

Taking expectation on both sides yields the statement of the lemma. $\square$

Next, we utilize Lemma C.4 along with the smoothness of the function $f(\cdot)$ (Assumption 1) to show descent in the objective function value at consecutive iterates.

**Lemma C.5** (Descent Lemma). *With $\bar{e}_t := \bar{d}_t - \frac{1}{K} \sum_{k=1}^{K} \nabla f^{(k)}(x_t^{(k)})$, for all $t \in [\bar{t}_{s-1}, \bar{t}_s - 1]$ and $s \in [S]$, then the iterates generated by Algorithm 3 satisfy:*

$$\mathbb{E}f(\bar{x}_{t+1}) \leq \mathbb{E}f(\bar{x}_t) - \left(\frac{\eta_t}{2} - \frac{\eta_t^2 L}{2}\right) \mathbb{E}\|\bar{d}_t\|^2 - \frac{\eta_t}{2}\mathbb{E}\|\nabla f(\bar{x}_t)\|^2 + \eta_t \mathbb{E}\|\bar{e}_t\|^2$$

$$+ \frac{\eta_t L^2 (I-1)}{K} \sum_{\ell=\bar{t}_{s-1}}^{t} \eta_\ell^2 \sum_{k=1}^{K} \mathbb{E}\|d_\ell^{(k)} - \bar{d}_\ell\|^2,$$

*where the expectation is w.r.t the stochasticity of the algorithm.*

*Proof.* Using the smoothness of $f$ (Assumption 1) we have:

$$f(\bar{x}_{t+1}) \leq f(\bar{x}_t) + \langle \nabla f(\bar{x}_t), \bar{x}_{t+1} - \bar{x}_t \rangle + \frac{L}{2}\|\bar{x}_{t+1} - \bar{x}_t\|^2$$

$$\overset{(a)}{=} f(\bar{x}_t) - \eta_t \langle \nabla f(\bar{x}_t), \bar{d}_t \rangle + \frac{\eta_t^2 L}{2}\|\bar{d}_t\|^2$$

$$\overset{(b)}{=} f(\bar{x}_t) - \eta_t\|\bar{d}_t\|^2 + \eta_t \langle \bar{d}_t - \nabla f(\bar{x}_t), \bar{d}_t \rangle + \frac{\eta_t^2 L}{2}\|\bar{d}_t\|^2$$

$$\overset{(c)}{=} f(\bar{x}_t) - \left(\frac{\eta_t}{2} - \frac{\eta_t^2 L}{2}\right)\|\bar{d}_t\|^2 - \frac{\eta_t}{2}\|\nabla f(\bar{x}_t)\|^2 + \frac{\eta_t}{2}\|\bar{d}_t - \nabla f(\bar{x}_t)\|^2$$

$$\overset{(d)}{\leq} f(\bar{x}_t) - \left(\frac{\eta_t}{2} - \frac{\eta_t^2 L}{2}\right)\|\bar{d}_t\|^2 - \frac{\eta_t}{2}\|\nabla f(\bar{x}_t)\|^2 + \eta_t\left\|\bar{d}_t - \frac{1}{K}\sum_{k=1}^{K}\nabla f^{(k)}(x_t^{(k)})\right\|^2$$

$$+ \eta_t\left\|\frac{1}{K}\sum_{k=1}^{K}\left(\nabla f^{(k)}(x_t^{(k)}) - \nabla f^{(k)}(\bar{x}_t)\right)\right\|^2, \quad (10)$$

where equality $(a)$ follows from the iterate update given in Step 10 of Algorithm 3, $(b)$ results by adding and subtracting $\bar{d}_t$ to $\nabla f(\bar{x}_t)$ in the inner product term and using the linearity of the inner product, $(c)$ follows from the relation $\langle x, y \rangle = \frac{1}{2}\|x\|^2 + \frac{1}{2}\|y\|^2 - \frac{1}{2}\|x - y\|^2$, finally inequality $(d)$ results from adding and subtracting $\frac{1}{K}\sum_{k=1}^{K}\nabla f^{(k)}(x_t^{(k)})$ in the last term of $(c)$ and using Lemma B.4.

Taking expectation on both sides and considering the last term of (10), we have

$$\mathbb{E}\left\|\frac{1}{K}\sum_{k=1}^{K}\left(\nabla f^{(k)}(x_t^{(k)}) - \nabla f^{(k)}(\bar{x}_t)\right)\right\|^2 \leq \frac{1}{K}\sum_{k=1}^{K}\mathbb{E}\left\|\nabla f^{(k)}(x_t^{(k)}) - \nabla f^{(k)}(\bar{x}_t)\right\|^2$$

$$\leq \frac{L^2}{K}\sum_{k=1}^{K}\mathbb{E}\|x_t^{(k)} - \bar{x}_t\|^2, \quad (11)$$

where the first inequality follows from Lemma B.4, and the second follows from the $L$-smoothness of $f^{(k)}(\cdot)$ (Assumption 1).

Substituting (11) in (10) and using the definition $\bar{e}_t := \bar{d}_t - \frac{1}{K}\sum_{k=1}^{K}\nabla f^{(k)}(x_t^{(k)})$ we get:

$$\mathbb{E}f(\bar{x}_{t+1}) \leq \mathbb{E}f(\bar{x}_t) - \left(\frac{\eta_t}{2} - \frac{\eta_t^2 L}{2}\right)\mathbb{E}\|\bar{d}_t\|^2 - \frac{\eta_t}{2}\mathbb{E}\|\nabla f(\bar{x}_t)\|^2 + \eta_t\mathbb{E}\|\bar{e}_t\|^2$$

$$+ \frac{\eta_t L^2}{K}\sum_{k=1}^{K}\mathbb{E}\|x_t^{(k)} - \bar{x}_t\|^2. \quad (12)$$

Finally, using Lemma C.4 to bound the last term of (12), we get:

$$\mathbb{E}f(\bar{x}_{t+1}) \leq \mathbb{E}f(\bar{x}_t) - \left(\frac{\eta_t}{2} - \frac{\eta_t^2 L}{2}\right)\mathbb{E}\|\bar{d}_t\|^2 - \frac{\eta_t}{2}\mathbb{E}\|\nabla f(\bar{x}_t)\|^2 + \eta_t\mathbb{E}\|\bar{e}_t\|^2$$
$$+ \frac{\eta_t L^2(I-1)}{K}\sum_{\ell=\bar{t}_{s-1}}^{t}\eta_\ell^2\sum_{k=1}^{K}\mathbb{E}\|d_\ell^{(k)} - \bar{d}_\ell\|^2.$$

Hence, the lemma is proved. $\qquad\square$

Lemma C.5 shows that the expected descent in the function $f$ depends on the magnitude of the expected gradient error term $\bar{e}_t$, and the expected gradient drift across WNs, i.e., $\mathbb{E}\|d_\ell^{(k)} - \bar{d}_\ell\|^2$. This implies that to ensure sufficient descent we need to control the gradient error, and the gradient drift across WNs. We achieve this by carefully designing the number of local updates, $I$, at each WN, and the batch-sizes $b$ (and initial batch size $B$), that each WN uses to compute the descent direction.

Next, we present the error contraction lemma which analyzes how the term $\mathbb{E}\|\bar{e}_t\|^2$ contracts across time.

### C.2.2 Proof of Gradient Error Contraction

**Lemma C.6** (Gradient Error Contraction). *Define $\bar{e}_t := \bar{d}_t - \frac{1}{K}\sum_{k=1}^{K}\nabla f^{(k)}(x_t^{(k)})$, then for every $t \in [T]$ the iterates generated by Algorithm 3 satisfy*

$$\mathbb{E}\|\bar{e}_{t+1}\|^2 \leq (1-a_{t+1})^2\mathbb{E}\|\bar{e}_t\|^2 + \frac{8(1-a_{t+1})^2 L^2}{bK^2}\frac{(I-1)}{I}\eta_t^2\sum_{k=1}^{K}\mathbb{E}\|d_t^{(k)} - \bar{d}_t\|^2$$
$$+ \frac{4(1-a_{t+1})^2 L^2\eta_t^2}{bK}\mathbb{E}\|\bar{d}_t\|^2 + \frac{2a_{t+1}^2\sigma^2}{bK},$$

*where the expectation is w.r.t the stochasticity of the algorithm.*

*Proof.* Consider the error term $\|\bar{e}_t\|^2$ as

$$\mathbb{E}\|\bar{e}_t\|^2 = \mathbb{E}\left\|\bar{d}_t - \frac{1}{K}\sum_{k=1}^{K}\nabla f^{(k)}(x_t^{(k)})\right\|^2$$

$$\stackrel{(a)}{=} \mathbb{E}\left\|\frac{1}{K}\sum_{k=1}^{K}\frac{1}{b}\sum_{\xi_t^{(k)}\in\mathcal{B}_t^{(k)}}\nabla f^{(k)}(x_t^{(k)};\xi_t^{(k)}) + (1-a_t)\left(\bar{d}_{t-1} - \frac{1}{K}\sum_{k=1}^{K}\frac{1}{b}\sum_{\xi_t^{(k)}\in\mathcal{B}_t^{(k)}}\nabla f^{(k)}(x_{t-1}^{(k)};\xi_t^{(k)})\right)\right.$$
$$\left. - \frac{1}{K}\sum_{k=1}^{K}\nabla f^{(k)}(x_t^{(k)})\right\|^2$$

$$\stackrel{(b)}{=} \mathbb{E}\left\|\frac{1}{K}\sum_{k=1}^{K}\frac{1}{b}\sum_{\xi_t^{(k)}\in\mathcal{B}_t^{(k)}}\left[\left(\nabla f^{(k)}(x_t^{(k)};\xi_t^{(k)}) - \nabla f^{(k)}(x_t^{(k)})\right)\right.\right.$$
$$\left.\left. - (1-a_t)\left(\nabla f^{(k)}(x_{t-1}^{(k)};\xi_t^{(k)}) - \nabla f^{(k)}(x_{t-1}^{(k)})\right)\right] + (1-a_t)\bar{e}_{t-1}\right\|^2,$$

where $(a)$ follows from the definition of descent direction given in Step 6 of Algorithm 3; $(b)$ follows by adding and subtracting $(1-a_t)\frac{1}{K}\sum_{k=1}^{K}\nabla f^{(k)}(x_{t-1}^{(k)})$ and using the definition of $\bar{e}_{t-1}$. Further simplifying the above expression, we get

$$\mathbb{E}\|\bar{e}_t\|^2 \stackrel{(c)}{=} (1-a_t)^2\mathbb{E}\|\bar{e}_{t-1}\|^2 + \frac{1}{b^2 K^2}\mathbb{E}\left\|\sum_{k=1}^{K}\sum_{\xi_t^{(k)}\in\mathcal{B}_t^{(k)}}\left[\left(\nabla f^{(k)}(x_t^{(k)};\xi_t^{(k)}) - \nabla f^{(k)}(x_t^{(k)})\right)\right.\right.$$
$$\left.\left. - (1-a_t)\left(\nabla f^{(k)}(x_{t-1}^{(k)};\xi_t^{(k)}) - \nabla f^{(k)}(x_{t-1}^{(k)})\right)\right]\right\|^2$$

$$\overset{(d)}{=} (1-a_t)^2 \mathbb{E}\|\bar{e}_{t-1}\|^2 + \frac{1}{b^2 K^2} \sum_{k=1}^{K} \mathbb{E}\left\| \sum_{\xi_t^{(k)} \in \mathcal{B}_t^{(k)}} \left[ \left( \nabla f^{(k)}(x_t^{(k)}; \xi_t^{(k)}) - \nabla f^{(k)}(x_t^{(k)}) \right) \right. \right.$$
$$\left. \left. - (1-a_t) \left( \nabla f^{(k)}(x_{t-1}^{(k)}; \xi_t^{(k)}) - \nabla f^{(k)}(x_{t-1}^{(k)}) \right) \right] \right\|^2,$$

$$\overset{(e)}{=} (1-a_t)^2 \mathbb{E}\|\bar{e}_{t-1}\|^2 + \frac{1}{b^2 K^2} \sum_{k=1}^{K} \sum_{\xi_t^{(k)} \in \mathcal{B}_t^{(k)}} \mathbb{E}\left\| \left( \nabla f^{(k)}(x_t^{(k)}; \xi_t^{(k)}) - \nabla f^{(k)}(x_t^{(k)}) \right) \right.$$
$$\left. - (1-a_t) \left( \nabla f^{(k)}(x_{t-1}^{(k)}; \xi_t^{(k)}) - \nabla f^{(k)}(x_{t-1}^{(k)}) \right) \right\|^2,$$
$$\tag{13}$$

where $(c)$ results from expanding the norm using inner product and noting that the cross terms are zero in expectation from Lemma C.1; $(d)$ follows from expanding the norm using the inner products across $k \in [K]$ and noting that the cross term is zero in expectation from Lemma C.2; finally, $(e)$ results from expanding the norm using the inner product across samples used to compute the minibatch gradients and the inner product is zero since at each node $k \in [K]$, the samples in the minibatch $\mathcal{B}_t^{(k)}$ are sampled independently of each other.

Now considering the 2nd term of (13) above, we have

$$\mathbb{E}\left\| \left( \nabla f^{(k)}(x_t^{(k)}; \xi_t^{(k)}) - \nabla f^{(k)}(x_t^{(k)}) \right) - (1-a_t) \left( \nabla f^{(k)}(x_{t-1}^{(k)}; \xi_t^{(k)}) - \nabla f^{(k)}(x_{t-1}^{(k)}) \right) \right\|^2$$
$$= \mathbb{E}\left\| (1-a_t) \left[ \left( \nabla f^{(k)}(x_t^{(k)}; \xi_t^{(k)}) - \nabla f^{(k)}(x_t^{(k)}) \right) - \left( \nabla f^{(k)}(x_{t-1}^{(k)}; \xi_t^{(k)}) - \nabla f^{(k)}(x_{t-1}^{(k)}) \right) \right] \right.$$
$$\left. + a_t \left( \nabla f^{(k)}(x_t^{(k)}; \xi_t^{(k)}) - \nabla f^{(k)}(x_t^{(k)}) \right) \right\|^2$$
$$\overset{(a)}{\leq} 2(1-a_t)^2 \mathbb{E}\left\| \left( \nabla f^{(k)}(x_t^{(k)}; \xi_t^{(k)}) - \nabla f^{(k)}(x_{t-1}^{(k)}; \xi_t^{(k)}) \right) - \left( \nabla f^{(k)}(x_t^{(k)}) - \nabla f^{(k)}(x_{t-1}^{(k)}) \right) \right\|^2$$
$$+ 2a_t^2 \mathbb{E}\left\| \nabla f^{(k)}(x_t^{(k)}; \xi_t^{(k)}) - \nabla f^{(k)}(x_t^{(k)}) \right\|^2$$
$$\overset{(b)}{\leq} 2(1-a_t)^2 \mathbb{E}\left\| \nabla f^{(k)}(x_t^{(k)}; \xi_t^{(k)}) - \nabla f^{(k)}(x_{t-1}^{(k)}; \xi_t^{(k)}) \right\|^2 + 2a_t^2 \sigma^2$$
$$\overset{(c)}{\leq} 2(1-a_t)^2 L^2 \mathbb{E}\|x_t^{(k)} - x_{t-1}^{(k)}\|^2 + 2a_t^2 \sigma^2$$
$$\overset{(d)}{\leq} 2(1-a_t)^2 L^2 \eta_{t-1}^2 \mathbb{E}\|d_{t-1}^{(k)}\|^2 + 2a_t^2 \sigma^2$$
$$\overset{(e)}{\leq} 8(1-a_t)^2 L^2 \frac{(I-1)}{I} \eta_{t-1}^2 \mathbb{E}\|d_{t-1}^{(k)} - \bar{d}_{t-1}\|^2 + 4(1-a_t)^2 L^2 \eta_{t-1}^2 \mathbb{E}\|\bar{d}_{t-1}\|^2 + 2a_t^2 \sigma^2,$$
$$\tag{14}$$

where $(a)$ follows from Lemma B.4; $(b)$ results from use of Assumption 2 and mean variance inequality: For a random variable $Z$ we have $\mathbb{E}\|Z - \mathbb{E}[Z]\|^2 \leq \mathbb{E}\|Z\|^2$; $(c)$ follows from the Lipschitz continuity of the gradient given in Assumption 1; $(d)$ results from the iterate update equation given in Step 10 of Algorithm 3; finally, $(e)$ uses the fact that: $(i)$ for $I = 1$ we have $d_t^{(k)} = \bar{d}_t$ for all $t \in [T]$ and $(ii)$ for $I \geq 2$ we use Lemma B.4 and the fact that $(I-1)/I \geq 1/2$.

Substituting (14) in (13) we get:

$$\mathbb{E}\|\bar{e}_t\|^2 \leq (1-a_t)^2 \mathbb{E}\|\bar{e}_{t-1}\|^2 + \frac{8(1-a_t)^2 L^2}{bK^2} \frac{(I-1)}{I} \eta_{t-1}^2 \sum_{k=1}^{K} \mathbb{E}\|d_{t-1}^{(k)} - \bar{d}_{t-1}\|^2$$
$$+ \frac{4(1-a_t)^2 L^2 \eta_{t-1}^2}{bK} \mathbb{E}\|\bar{d}_{t-1}\|^2 + \frac{2a_t^2 \sigma^2}{bK}.$$

Finally, the lemma is proved by replacing $t$ by $t+1$. □

Lemma C.6 shows that the gradient error contracts in each iteration. Next, we first define a potential function and then utilize Lemmas C.5 and C.6 to show descent in the potential function.

### C.2.3 Descent in Potential Function

We define the potential function as a linear combination of the objective function and the gradient estimation error: $\bar{e}_t := \bar{d}_t - \frac{1}{K}\sum_{k=1}^{K}\nabla f^{(k)}(x_t^{(k)})$

$$\Phi_t := f(\bar{x}_t) + \frac{bK}{64L^2}\frac{\|\bar{e}_t\|^2}{\eta_{t-1}}. \tag{15}$$

Next, we characterize the descent in the potential function.

**Lemma C.7** (Potential Function Descent). *For $\bar{t} \in [\bar{t}_{s-1}, \bar{t}_s - 1]$ and for $\eta_t \leq \frac{1}{16LI}$ we have*

$$\mathbb{E}[\Phi_{\bar{t}+1} - \Phi_{\bar{t}_{s-1}}] \leq -\sum_{t=\bar{t}_{s-1}}^{\bar{t}}\left(\frac{7\eta_t}{16} - \frac{\eta_t^2 L}{2}\right)\mathbb{E}\|\bar{d}_t\|^2 - \sum_{t=\bar{t}_{s-1}}^{\bar{t}}\frac{\eta_t}{2}\mathbb{E}\|\nabla f(\bar{x}_t)\|^2 + \frac{\sigma^2 c^2}{32L^2}\sum_{t=\bar{t}_{s-1}}^{\bar{t}}\eta_t^3$$

$$+ \frac{33}{256K}\frac{(I-1)}{I}\sum_{t=\bar{t}_{s-1}}^{\bar{t}}\eta_t\sum_{k=1}^{K}\mathbb{E}\|d_t^{(k)} - \bar{d}_t\|^2$$

*where the expectation is w.r.t the stochasticity of the algorithm.*

*Proof.* To get the descent on the the potential function, i.e. $\mathbb{E}[\Phi_{t+1} - \Phi_t]$, we first consider the term: $\frac{\mathbb{E}\|\bar{e}_{t+1}\|^2}{\eta_t} - \frac{\mathbb{E}\|\bar{e}_t\|^2}{\eta_{t-1}}$.

Using Lemma C.6 we get

$$\frac{\mathbb{E}\|\bar{e}_{t+1}\|^2}{\eta_t} - \frac{\mathbb{E}\|\bar{e}_t\|^2}{\eta_{t-1}} \leq \left[\frac{(1-a_{t+1})^2}{\eta_t} - \frac{1}{\eta_{t-1}}\right]\mathbb{E}\|\bar{e}_t\|^2 + \frac{8(1-a_{t+1})^2 L^2}{bK^2}\frac{(I-1)}{I}\eta_t\sum_{k=1}^{K}\mathbb{E}\|d_t^{(k)} - \bar{d}_t\|^2$$

$$+ \frac{4(1-a_{t+1})^2 L^2\eta_t}{bK}\mathbb{E}\|\bar{d}_t\|^2 + \frac{2a_{t+1}^2\sigma^2}{\eta_t bK}$$

$$\overset{(a)}{\leq} (\eta_t^{-1} - \eta_{t-1}^{-1} - c\eta_t)\mathbb{E}\|\bar{e}_t\|^2 + \frac{8L^2}{bK^2}\frac{(I-1)}{I}\eta_t\sum_{k=1}^{K}\mathbb{E}\|d_t^{(k)} - \bar{d}_t\|^2$$

$$+ \frac{4L^2\eta_t}{bK}\mathbb{E}\|\bar{d}_t\|^2 + \frac{2\sigma^2 c^2\eta_t^3}{bK}, \tag{16}$$

where inequality $(a)$ utilizes the fact that $(1-a_t)^2 \leq 1 - a_t \leq 1$ for all $t \in [T]$.

Let us consider $\eta_t^{-1} - \eta_{t-1}^{-1}$ in the first term of the inequality in (16) and using the definition of the stepsize $\eta_t$ from Theorem 3.1, we have

$$\eta_t^{-1} - \eta_{t-1}^{-1} = \frac{(w_t + \sigma^2 t)^{1/3}}{\bar{\kappa}} - \frac{(w_{t-1} + \sigma^2(t-1))^{1/3}}{\bar{\kappa}}$$

$$\overset{(a)}{\leq} \frac{(w_t + \sigma^2 t)^{1/3}}{\bar{\kappa}} - \frac{(w_t + \sigma^2(t-1))^{1/3}}{\bar{\kappa}}$$

$$\overset{(b)}{\leq} \frac{\sigma^2}{3\bar{\kappa}(w_t + \sigma^2(t-1))^{2/3}}$$

$$\overset{(c)}{\leq} \frac{2^{2/3}\sigma^2\bar{\kappa}^2}{3\bar{\kappa}^3(w_t + \sigma^2 t)^{2/3}}$$

$$\overset{(d)}{=} \frac{2^{2/3}\sigma^2}{3\bar{\kappa}^3}\eta_t^2$$

$$\overset{(e)}{\leq} \frac{\sigma^2}{24\bar{\kappa}^3 LI}\eta_t, \tag{17}$$

where inequality $(a)$ follows from the fact that we choose $w_t \leq w_{t-1}$ (see definition of $w_t$ in Theorem 3.1), $(b)$ results from the concavity of $x^{1/3}$ as:

$$(x+y)^{1/3} - x^{1/3} \leq \frac{y}{3x^{2/3}}.$$

In inequality $(c)$, we have used the fact that $w_t \geq 2\sigma^2$, finally, $(d)$ and $(e)$ utilize the definition of $\eta_t$ and the fact that $\eta_t \leq 1/16LI$ for all $t \in [T]$, respectively.

Now combining the first term of inequality in (16) with (17) and choosing $c = \dfrac{64L^2}{bK} + \dfrac{\sigma^2}{24\bar{\kappa}^3 LI}$ we get:

$$\eta_t^{-1} - \eta_{t-1}^{-1} - c\eta_t \leq -\frac{64L^2}{bK}\eta_t.$$

Therefore, we have from (16):

$$\frac{\mathbb{E}\|\bar{e}_{t+1}\|^2}{\eta_t} - \frac{\mathbb{E}\|\bar{e}_t\|^2}{\eta_{t-1}} \leq -\frac{64L^2\eta_t}{bK}\mathbb{E}\|\bar{e}_t\|^2 + \frac{8L^2}{bK^2}\frac{(I-1)}{I}\eta_t\sum_{k=1}^{K}\mathbb{E}\|d_t^{(k)} - \bar{d}_t\|^2$$

$$+ \frac{4L^2\eta_t}{bK}\mathbb{E}\|\bar{d}_t\|^2 + \frac{2\sigma^2c^2\eta_t^3}{bK}$$

$$\frac{bK}{64L^2}\left(\frac{\mathbb{E}\|\bar{e}_{t+1}\|^2}{\eta_t} - \frac{\mathbb{E}\|\bar{e}_t\|^2}{\eta_{t-1}}\right) \leq -\eta_t\mathbb{E}\|\bar{e}_t\|^2 + \frac{1}{8K}\frac{(I-1)}{I}\eta_t\sum_{k=1}^{K}\mathbb{E}\|d_t^{(k)} - \bar{d}_t\|^2 + \frac{\eta_t}{16}\mathbb{E}\|\bar{d}_t\|^2 + \frac{\sigma^2c^2\eta_t^3}{32L^2}.$$

Finally, using Lemma C.5 and the definition of potential function given in (15), using the above we get the descent in the potential function for any $t \in [\bar{t}_{s-1}, \bar{t}_s - 1]$ with $s \in [S]$ as:

$$\mathbb{E}[\Phi_{t+1} - \Phi_t] \leq -\left(\frac{7\eta_t}{16} - \frac{\eta_t^2 L}{2}\right)\mathbb{E}\|\bar{d}_t\|^2 - \frac{\eta_t}{2}\mathbb{E}\|\nabla f(\bar{x}_t)\|^2 + \frac{\eta_t L^2(I-1)}{K}\sum_{\ell=\bar{t}_{s-1}}^{t}\eta_\ell^2\sum_{k=1}^{K}\mathbb{E}\|d_\ell^{(k)} - \bar{d}_\ell\|^2$$

$$+ \frac{1}{8K}\frac{(I-1)}{I}\eta_t\sum_{k=1}^{K}\mathbb{E}\|d_t^{(k)} - \bar{d}_t\|^2 + \frac{\sigma^2c^2\eta_t^3}{32L^2}.$$

Summing the above over $t = \bar{t}_{s-1}$ to $\bar{t}$ for $\bar{t} \in [\bar{t}_{s-1}, \bar{t}_s - 1]$, we get:

$$\mathbb{E}[\Phi_{\bar{t}+1} - \Phi_{\bar{t}_{s-1}}] \leq -\sum_{t=\bar{t}_{s-1}}^{\bar{t}}\left(\frac{7\eta_t}{16} - \frac{\eta_t^2 L}{2}\right)\mathbb{E}\|\bar{d}_t\|^2 - \sum_{t=\bar{t}_{s-1}}^{\bar{t}}\frac{\eta_t}{2}\mathbb{E}\|\nabla f(\bar{x}_t)\|^2 + \frac{\sigma^2c^2}{32L^2}\sum_{t=\bar{t}_{s-1}}^{\bar{t}}\eta_t^3$$

$$+ \frac{L^2(I-1)}{K}\sum_{t=\bar{t}_{s-1}}^{\bar{t}}\eta_t\sum_{\ell=\bar{t}_{s-1}}^{t}\eta_\ell^2\sum_{k=1}^{K}\mathbb{E}\|d_\ell^{(k)} - \bar{d}_\ell\|^2 + \frac{1}{8K}\frac{(I-1)}{I}\sum_{t=\bar{t}_{s-1}}^{\bar{t}}\eta_t\sum_{k=1}^{K}\mathbb{E}\|d_t^{(k)} - \bar{d}_t\|^2$$

$$\leq -\sum_{t=\bar{t}_{s-1}}^{\bar{t}}\left(\frac{7\eta_t}{16} - \frac{\eta_t^2 L}{2}\right)\mathbb{E}\|\bar{d}_t\|^2 - \sum_{t=\bar{t}_{s-1}}^{\bar{t}}\frac{\eta_t}{2}\mathbb{E}\|\nabla f(\bar{x}_t)\|^2 + \frac{\sigma^2c^2}{32L^2}\sum_{t=\bar{t}_{s-1}}^{\bar{t}}\eta_t^3$$

$$+ \frac{L^2(I-1)}{K}\left(\sum_{t=\bar{t}_{s-1}}^{\bar{t}}\eta_t\right)\left(\sum_{\ell=\bar{t}_{s-1}}^{\bar{t}}\eta_\ell^2\sum_{k=1}^{K}\mathbb{E}\|d_\ell^{(k)} - \bar{d}_\ell\|^2\right)$$

$$+ \frac{1}{8K}\frac{(I-1)}{I}\sum_{t=\bar{t}_{s-1}}^{\bar{t}}\eta_t\sum_{k=1}^{K}\mathbb{E}\|d_t^{(k)} - \bar{d}_t\|^2.$$

Finally, using the fact that we have: $\eta_t \leq 1/16LI$ for all $t \in [T]$, we get:

$$\mathbb{E}[\Phi_{\bar{t}+1} - \Phi_{\bar{t}_{s-1}}] \leq -\sum_{t=\bar{t}_{s-1}}^{\bar{t}}\left(\frac{7\eta_t}{16} - \frac{\eta_t^2 L}{2}\right)\mathbb{E}\|\bar{d}_t\|^2 - \sum_{t=\bar{t}_{s-1}}^{\bar{t}}\frac{\eta_t}{2}\mathbb{E}\|\nabla f(\bar{x}_t)\|^2 + \frac{\sigma^2c^2}{32L^2}\sum_{t=\bar{t}_{s-1}}^{\bar{t}}\eta_t^3$$

$$+ \frac{L^2(I-1)}{K}\left(I \times \frac{1}{16LI} \times \frac{1}{16LI}\right)\sum_{t=\bar{t}_{s-1}}^{\bar{t}}\eta_t\sum_{k=1}^{K}\mathbb{E}\|d_t^{(k)} - \bar{d}_t\|^2$$

$$+ \frac{1}{8K}\frac{(I-1)}{I}\sum_{t=\bar{t}_{s-1}}^{\bar{t}}\eta_t\sum_{k=1}^{K}\mathbb{E}\|d_t^{(k)} - \bar{d}_t\|^2$$

$$= -\sum_{t=\bar{t}_{s-1}}^{\bar{t}} \left( \frac{7\eta_t}{16} - \frac{\eta_t^2 L}{2} \right) \mathbb{E}\|\bar{d}_t\|^2 - \sum_{t=\bar{t}_{s-1}}^{\bar{t}} \frac{\eta_t}{2} \mathbb{E}\|\nabla f(\bar{x}_t)\|^2 + \frac{\sigma^2 c^2}{32L^2} \sum_{t=\bar{t}_{s-1}}^{\bar{t}} \eta_t^3$$

$$+ \frac{33}{256K} \frac{(I-1)}{I} \sum_{t=\bar{t}_{s-1}}^{\bar{t}} \eta_t \sum_{k=1}^{K} \mathbb{E}\|d_t^{(k)} - \bar{d}_t\|^2.$$

Therefore, the lemma is proved. $\qquad\square$

Multiple local updates at each WN on heterogeneous data can cause the local descent directions to drift away from each other. Next, we bound this error accumulated via gradient drift across WNs.

### C.2.4 Accumulated Gradient Consensus Error

We first upper bound the gradient consensus error given by term $\sum_{k=1}^{K} \mathbb{E}\|d_t^{(k)} - \bar{d}_t\|^2$.

**Lemma C.8** (Gradient Consensus Error). *For every $t \in [T]$ and some $\beta > 0$ we have*

$$\sum_{k=1}^{K} \mathbb{E}\|d_t^{(k)} - \bar{d}_t\|^2 \leq \left[ (1-a_t)^2(1+\beta) + 4L^2 \left( 1 + \frac{1}{\beta} \right) \eta_{t-1}^2 \right] \sum_{k=1}^{K} \mathbb{E}\|d_{t-1}^{(k)} - \bar{d}_{t-1}\|^2$$

$$+ 4KL^2 \left( 1 + \frac{1}{\beta} \right) \eta_{t-1}^2 \mathbb{E}\|\bar{d}_{t-1}\|^2 + \frac{4K\sigma^2}{b} \left( 1 + \frac{1}{\beta} \right) a_t^2 + 8K\zeta^2 \left( 1 + \frac{1}{\beta} \right) a_t^2$$

$$+ 32L^2 \left( 1 + \frac{1}{\beta} \right) (I-1) a_t^2 \sum_{\bar{\ell}=\bar{t}_{s-1}}^{t-1} \eta_{\bar{\ell}}^2 \sum_{k=1}^{K} \mathbb{E}\|d_{\bar{\ell}}^{(k)} - \bar{d}_{\bar{\ell}}\|^2.$$

*where the expectation is w.r.t. the stochasticity of the algorithm.*

*Proof.* Using the definition of the descent direction $d_t^{(k)}$ from Algorithm 3 we have

$$\sum_{k=1}^{K} \mathbb{E}\|d_t^{(k)} - \bar{d}_t\|^2 \tag{18}$$

$$= \sum_{k=1}^{K} \mathbb{E}\left\| \frac{1}{b} \sum_{\xi_t^{(k)} \in \mathcal{B}_t^{(k)}} \nabla f^{(k)}(x_t^{(k)}; \xi_t^{(k)}) + (1-a_t)\left(d_{t-1}^{(k)} - \frac{1}{b} \sum_{\xi_t^{(k)} \in \mathcal{B}_t^{(k)}} \nabla f^{(k)}(x_{t-1}^{(k)}; \xi_t^{(k)})\right) \right.$$

$$\left. - \left( \frac{1}{K} \sum_{j=1}^{K} \frac{1}{b} \sum_{\xi_t^{(j)} \in \mathcal{B}_t^{(j)}} \nabla f^{(j)}(x_t^{(j)}; \xi_t^{(j)}) + (1-a_t)\left(\bar{d}_{t-1} - \frac{1}{K} \sum_{j=1}^{K} \frac{1}{b} \sum_{\xi_t^{(j)} \in \mathcal{B}_t^{(j)}} \nabla f^{(j)}(x_{t-1}^{(j)}; \xi_t^{(j)})\right) \right) \right\|^2$$

$$= \sum_{k=1}^{K} \mathbb{E}\left\| (1-a_t)\left(d_{t-1}^{(k)} - \bar{d}_{t-1}\right) + \frac{1}{b} \sum_{\xi_t^{(k)} \in \mathcal{B}_t^{(k)}} \nabla f^{(k)}(x_t^{(k)}; \xi_t^{(k)}) - \frac{1}{K} \sum_{j=1}^{K} \frac{1}{b} \sum_{\xi_t^{(j)} \in \mathcal{B}_t^{(j)}} \nabla f^{(j)}(x_t^{(j)}; \xi_t^{(j)}) \right.$$

$$\left. - (1-a_t)\left( \frac{1}{b} \sum_{\xi_t^{(k)} \in \mathcal{B}_t^{(k)}} \nabla f^{(k)}(x_{t-1}^{(k)}; \xi_t^{(k)}) - \frac{1}{K} \sum_{j=1}^{K} \frac{1}{b} \sum_{\xi_t^{(j)} \in \mathcal{B}_t^{(j)}} \nabla f^{(j)}(x_{t-1}^{(j)}; \xi_t^{(j)}) \right) \right\|^2$$

$$\overset{(a)}{\leq} (1+\beta)(1-a_t)^2 \sum_{k=1}^{K} \mathbb{E}\|d_{t-1}^{(k)} - \bar{d}_{t-1}\|^2$$

$$+ \left( 1 + \frac{1}{\beta} \right) \sum_{k=1}^{K} \mathbb{E}\left\| \frac{1}{b} \sum_{\xi_t^{(k)} \in \mathcal{B}_t^{(k)}} \nabla f^{(k)}(x_t^{(k)}; \xi_t^{(k)}) - \frac{1}{K} \sum_{j=1}^{K} \frac{1}{b} \sum_{\xi_t^{(j)} \in \mathcal{B}_t^{(j)}} \nabla f^{(j)}(x_t^{(j)}; \xi_t^{(j)}) \right.$$

$$\left. - (1-a_t)\left( \frac{1}{b} \sum_{\xi_t^{(k)} \in \mathcal{B}_t^{(k)}} \nabla f^{(k)}(x_{t-1}^{(k)}; \xi_t^{(k)}) - \frac{1}{K} \sum_{j=1}^{K} \frac{1}{b} \sum_{\xi_t^{(j)} \in \mathcal{B}_t^{(j)}} \nabla f^{(j)}(x_{t-1}^{(j)}; \xi_t^{(j)}) \right) \right\|^2$$

$$\tag{19}$$

where inequality $(a)$ follows from the Young's inequality for some $\beta > 0$. Now considering the second term in (19), we get

$$
\sum_{k=1}^{K} \mathbb{E} \left\| \frac{1}{b} \sum_{\xi_t^{(k)} \in \mathcal{B}_t^{(k)}} \nabla f^{(k)}(x_t^{(k)}; \xi_t^{(k)}) - \frac{1}{K} \sum_{j=1}^{K} \frac{1}{b} \sum_{\xi_t^{(j)} \in \mathcal{B}_t^{(j)}} \nabla f^{(j)}(x_t^{(j)}; \xi_t^{(j)}) \right.
$$

$$
\left. - (1 - a_t) \left( \frac{1}{b} \sum_{\xi_t^{(k)} \in \mathcal{B}_t^{(k)}} \nabla f^{(k)}(x_{t-1}^{(k)}; \xi_t^{(k)}) - \frac{1}{K} \sum_{j=1}^{K} \frac{1}{b} \sum_{\xi_t^{(j)} \in \mathcal{B}_t^{(j)}} \nabla f^{(j)}(x_{t-1}^{(j)}; \xi_t^{(j)}) \right) \right\|^2
$$

$$
= \sum_{k=1}^{K} \mathbb{E} \left\| \frac{1}{b} \sum_{\xi_t^{(k)} \in \mathcal{B}_t^{(k)}} \nabla f^{(k)}(x_t^{(k)}; \xi_t^{(k)}) - \frac{1}{K} \sum_{j=1}^{K} \frac{1}{b} \sum_{\xi_t^{(j)} \in \mathcal{B}_t^{(j)}} \nabla f^{(j)}(x_t^{(j)}; \xi_t^{(j)}) \right.
$$

$$
- \left( \frac{1}{b} \sum_{\xi_t^{(k)} \in \mathcal{B}_t^{(k)}} \nabla f^{(k)}(x_{t-1}^{(k)}; \xi_t^{(k)}) - \frac{1}{K} \sum_{j=1}^{K} \frac{1}{b} \sum_{\xi_t^{(j)} \in \mathcal{B}_t^{(j)}} \nabla f^{(j)}(x_{t-1}^{(j)}; \xi_t^{(j)}) \right)
$$

$$
\left. + a_t \left( \frac{1}{b} \sum_{\xi_t^{(k)} \in \mathcal{B}_t^{(k)}} \nabla f^{(k)}(x_{t-1}^{(k)}; \xi_t^{(k)}) - \frac{1}{K} \sum_{j=1}^{K} \frac{1}{b} \sum_{\xi_t^{(j)} \in \mathcal{B}_t^{(j)}} \nabla f^{(j)}(x_{t-1}^{(j)}; \xi_t^{(j)}) \right) \right\|^2
$$

$$
\overset{(a)}{\leq} 2 \sum_{k=1}^{K} \mathbb{E} \left\| \frac{1}{b} \sum_{\xi_t^{(k)} \in \mathcal{B}_t^{(k)}} \nabla f^{(k)}(x_t^{(k)}; \xi_t^{(k)}) - \frac{1}{K} \sum_{j=1}^{K} \frac{1}{b} \sum_{\xi_t^{(j)} \in \mathcal{B}_t^{(j)}} \nabla f^{(j)}(x_t^{(j)}; \xi_t^{(j)}) \right.
$$

$$
\left. - \left( \frac{1}{b} \sum_{\xi_t^{(k)} \in \mathcal{B}_t^{(k)}} \nabla f^{(k)}(x_{t-1}^{(k)}; \xi_t^{(k)}) - \frac{1}{K} \sum_{j=1}^{K} \frac{1}{b} \sum_{\xi_t^{(j)} \in \mathcal{B}_t^{(j)}} \nabla f^{(j)}(x_{t-1}^{(j)}; \xi_t^{(j)}) \right) \right\|^2
$$

$$
+ 2 a_t^2 \sum_{k=1}^{K} \mathbb{E} \left\| \frac{1}{b} \sum_{\xi_t^{(k)} \in \mathcal{B}_t^{(k)}} \nabla f^{(k)}(x_{t-1}^{(k)}; \xi_t^{(k)}) - \frac{1}{K} \sum_{j=1}^{K} \frac{1}{b} \sum_{\xi_t^{(j)} \in \mathcal{B}_t^{(j)}} \nabla f^{(j)}(x_{t-1}^{(j)}; \xi_t^{(j)}) \right\|^2
$$

$$
\overset{(b)}{\leq} 2 \sum_{k=1}^{K} \mathbb{E} \left\| \frac{1}{b} \sum_{\xi_t^{(k)} \in \mathcal{B}_t^{(k)}} \left( \nabla f^{(k)}(x_t^{(k)}; \xi_t^{(k)}) - \nabla f^{(k)}(x_{t-1}^{(k)}; \xi_t^{(k)}) \right) \right\|^2
$$

$$
+ 2 a_t^2 \sum_{k=1}^{K} \mathbb{E} \left\| \frac{1}{b} \sum_{\xi_t^{(k)} \in \mathcal{B}_t^{(k)}} \nabla f^{(k)}(x_{t-1}^{(k)}; \xi_t^{(k)}) - \frac{1}{K} \sum_{j=1}^{K} \frac{1}{b} \sum_{\xi_t^{(j)} \in \mathcal{B}_t^{(j)}} \nabla f^{(j)}(x_{t-1}^{(j)}; \xi_t^{(j)}) \right\|^2
$$

$$
\overset{(c)}{\leq} 2 \sum_{k=1}^{K} \frac{1}{b} \sum_{\xi_t^{(k)} \in \mathcal{B}_t^{(k)}} \mathbb{E} \left\| \nabla f^{(k)}(x_t^{(k)}; \xi_t^{(k)}) - \nabla f^{(k)}(x_{t-1}^{(k)}; \xi_t^{(k)}) \right\|^2
$$

$$
+ 2 a_t^2 \sum_{k=1}^{K} \mathbb{E} \left\| \frac{1}{b} \sum_{\xi_t^{(k)} \in \mathcal{B}_t^{(k)}} \nabla f^{(k)}(x_{t-1}^{(k)}; \xi_t^{(k)}) - \frac{1}{K} \sum_{j=1}^{K} \frac{1}{b} \sum_{\xi_t^{(j)} \in \mathcal{B}_t^{(j)}} \nabla f^{(j)}(x_{t-1}^{(j)}; \xi_t^{(j)}) \right\|^2
$$

$$
\overset{(d)}{\leq} 2 L^2 \sum_{k=1}^{K} \mathbb{E} \| x_t^{(k)} - x_{t-1}^{(k)} \|^2 + 2 a_t^2 \sum_{k=1}^{K} \mathbb{E} \left\| \frac{1}{b} \sum_{\xi_t^{(k)} \in \mathcal{B}_t^{(k)}} \nabla f^{(k)}(x_{t-1}^{(k)}; \xi_t^{(k)}) - \frac{1}{K} \sum_{j=1}^{K} \frac{1}{b} \sum_{\xi_t^{(j)} \in \mathcal{B}_t^{(j)}} \nabla f^{(j)}(x_{t-1}^{(j)}; \xi_t^{(j)}) \right\|^2,
$$

$$
\tag{20}
$$

where inequality $(a)$ above follows from Lemma B.4, $(b)$ follows from Lemma B.2, inequality $(c)$ again uses Lemma B.4 and $(d)$ follows from the Lipschitz-smoothness of the individual functions $f^{(k)}$ (Assumption 1).

Now considering the second term in (20) above, we have

$$\sum_{k=1}^{K} \mathbb{E}\left\|\frac{1}{b}\sum_{\xi_t^{(k)}\in\mathcal{B}_t^{(k)}}\nabla f^{(k)}(x_{t-1}^{(k)};\xi_t^{(k)}) - \frac{1}{K}\sum_{j=1}^{K}\frac{1}{b}\sum_{\xi_t^{(j)}\in\mathcal{B}_t^{(j)}}\nabla f^{(j)}(x_{t-1}^{(j)};\xi_t^{(j)})\right\|^2$$

$$\overset{(a)}{=} \sum_{k=1}^{K}\mathbb{E}\left\|\frac{1}{b}\sum_{\xi_t^{(k)}\in\mathcal{B}_t^{(k)}}\left(\nabla f^{(k)}(x_{t-1}^{(k)};\xi_t^{(k)}) - \nabla f^{(k)}(x_{t-1}^{(k)})\right)\right.$$
$$\left. - \frac{1}{K}\sum_{j=1}^{K}\frac{1}{b}\sum_{\xi_t^{(j)}\in\mathcal{B}_t^{(j)}}\left(\nabla f^{(j)}(x_{t-1}^{(j)};\xi_t^{(j)}) - \nabla f^{(j)}(x_{t-1}^{(j)})\right) + \nabla f^{(k)}(x_{t-1}^{(k)}) - \frac{1}{K}\sum_{j=1}^{K}\nabla f^{(j)}(x_{t-1}^{(j)})\right\|^2$$

$$\overset{(b)}{\leq} 2\sum_{k=1}^{K}\mathbb{E}\left\|\frac{1}{b}\sum_{\xi_t^{(k)}\in\mathcal{B}_t^{(k)}}\left(\nabla f^{(k)}(x_{t-1}^{(k)};\xi_t^{(k)}) - \nabla f^{(k)}(x_{t-1}^{(k)})\right)\right.$$
$$\left. - \frac{1}{K}\sum_{j=1}^{K}\frac{1}{b}\sum_{\xi_t^{(j)}\in\mathcal{B}_t^{(j)}}\left(\nabla f^{(j)}(x_{t-1}^{(j)};\xi_t^{(j)}) - \nabla f^{(j)}(x_{t-1}^{(j)})\right)\right\|^2$$
$$+ 2\sum_{k=1}^{K}\mathbb{E}\left\|\nabla f^{(k)}(x_{t-1}^{(k)}) - \frac{1}{K}\sum_{j=1}^{K}\nabla f^{(j)}(x_{t-1}^{(j)})\right\|^2$$

$$\overset{(c)}{\leq} 2\sum_{k=1}^{K}\mathbb{E}\left\|\frac{1}{b}\sum_{\xi_t^{(k)}\in\mathcal{B}_t^{(k)}}\left(\nabla f^{(k)}(x_{t-1}^{(k)};\xi_t^{(k)}) - \nabla f^{(k)}(x_{t-1}^{(k)})\right)\right\|^2$$
$$+ 2\sum_{k=1}^{K}\mathbb{E}\left\|\nabla f^{(k)}(x_{t-1}^{(k)}) - \frac{1}{K}\sum_{j=1}^{K}\nabla f^{(j)}(x_{t-1}^{(j)})\right\|^2$$

$$\overset{(d)}{\leq} 2\sum_{k=1}^{K}\frac{1}{b^2}\sum_{\xi_t^{(k)}\in\mathcal{B}_t^{(k)}}\mathbb{E}\left\|\left(\nabla f^{(k)}(x_{t-1}^{(k)};\xi_t^{(k)}) - \nabla f^{(k)}(x_{t-1}^{(k)})\right)\right\|^2 + 4\sum_{k=1}^{K}\mathbb{E}\left\|\nabla f^{(k)}(\bar{x}_{t-1}) - \nabla f(\bar{x}_{t-1})\right\|^2$$
$$+ 8\sum_{k=1}^{K}\mathbb{E}\left\|\nabla f^{(k)}(x_{t-1}^{(k)}) - \nabla f^{(k)}(\bar{x}_{t-1})\right\|^2 + 8\sum_{k=1}^{K}\mathbb{E}\left\|\nabla f(\bar{x}_{t-1}) - \frac{1}{K}\sum_{j=1}^{K}\nabla f^{(j)}(x_{t-1}^{(j)})\right\|^2$$

$$\overset{(e)}{\leq} \frac{2K\sigma^2}{b} + 4\sum_{k=1}^{K}\frac{1}{K}\sum_{j=1}^{K}\mathbb{E}\|\nabla f^{(k)}(\bar{x}_{t-1}) - \nabla f^{(j)}(\bar{x}_{t-1})\|^2 + 16L^2\sum_{k=1}^{K}\mathbb{E}\|x_{t-1}^{(k)} - \bar{x}_{t-1}\|^2$$

$$\overset{(g)}{\leq} \frac{2K\sigma^2}{b} + 4K\zeta^2 + 16L^2\sum_{k=1}^{K}\mathbb{E}\|x_{t-1}^{(k)} - \bar{x}_{t-1}\|^2, \tag{21}$$

where equality $(a)$ follows from adding and subtracting $\nabla f^{(k)}(x_{t-1}^{(k)})$ and $\frac{1}{K}\sum_{j=1}^{K}\nabla f^{(j)}(x_{t-1}^{(j)})$ inside the norm; inequality $(b)$ uses Lemma B.4; inequality $(c)$ results from the use of Lemma B.2; inequality $(d)$ expands the sum of the first term using inner products and utilizes the fact that the cross product terms are zero in expectation. This follows from the fact that conditioned on $\mathcal{F}_t$ we have $\mathbb{E}[\nabla f^{(k)}(x_t^{(k)};\xi_t^{(k)})] = \nabla f^{(k)}(x_t^{(k)})$ for all $k \in [K]$ and $t \in [T]$; inequality $(e)$ utilizes Intra-Node Variance Bound (Assumption 2), and Lemma B.4; finally, $(g)$ follows from Inter-Node Variance Bound (Assumption 2).

Finally, substituting (21) and (20) in (19), we get

$$\sum_{k=1}^{K}\mathbb{E}\|d_t^{(k)} - \bar{d}_t\|^2 \leq (1-a_t)^2(1+\beta)\sum_{k=1}^{K}\mathbb{E}\|d_{t-1}^{(k)} - \bar{d}_{t-1}\|^2 + 2L^2\left(1+\frac{1}{\beta}\right)\sum_{k=1}^{K}\mathbb{E}\|x_t^{(k)} - x_{t-1}^{(k)}\|^2$$
$$+ \frac{4K\sigma^2}{b}\left(1+\frac{1}{\beta}\right)a_t^2 + 8K\zeta^2\left(1+\frac{1}{\beta}\right)a_t^2 + 32L^2\left(1+\frac{1}{\beta}\right)a_t^2\sum_{k=1}^{K}\mathbb{E}\|x_{t-1}^{(k)} - \bar{x}_{t-1}\|^2$$

$$\overset{(a)}{\leq} (1-a_t)^2(1+\beta) \sum_{k=1}^{K} \mathbb{E}\|d_{t-1}^{(k)} - \bar{d}_{t-1}\|^2 + 2L^2\left(1 + \frac{1}{\beta}\right)\eta_{t-1}^2 \sum_{k=1}^{K} \mathbb{E}\|d_{t-1}^{(k)}\|^2$$

$$+ \frac{4K\sigma^2}{b}\left(1 + \frac{1}{\beta}\right)a_t^2 + 8K\zeta^2\left(1 + \frac{1}{\beta}\right)a_t^2$$

$$+ 32L^2\left(1 + \frac{1}{\beta}\right)(I-1)a_t^2 \sum_{\bar{\ell}=\bar{t}_{s-1}}^{t-1} \eta_{\bar{\ell}}^2 \sum_{k=1}^{K} \mathbb{E}\|d_{\bar{\ell}}^{(k)} - \bar{d}_{\bar{\ell}}\|^2$$

$$\overset{(b)}{\leq} (1-a_t)^2(1+\beta) \sum_{k=1}^{K} \mathbb{E}\|d_{t-1}^{(k)} - \bar{d}_{t-1}\|^2 + 4L^2\left(1 + \frac{1}{\beta}\right)\eta_{t-1}^2 \sum_{k=1}^{K} \mathbb{E}\|d_{t-1}^{(k)} - \bar{d}_{t-1}\|^2$$

$$+ 4L^2\left(1 + \frac{1}{\beta}\right)\eta_{t-1}^2 \sum_{k=1}^{K} \mathbb{E}\|\bar{d}_{t-1}\|^2 + \frac{4K\sigma^2}{b}\left(1 + \frac{1}{\beta}\right)a_t^2 + 8K\zeta^2\left(1 + \frac{1}{\beta}\right)a_t^2$$

$$+ 32L^2\left(1 + \frac{1}{\beta}\right)(I-1)a_t^2 \sum_{\bar{\ell}=\bar{t}_{s-1}}^{t-1} \eta_{\bar{\ell}}^2 \sum_{k=1}^{K} \mathbb{E}\|d_{\bar{\ell}}^{(k)} - \bar{d}_{\bar{\ell}}\|^2$$

$$= \left[(1-a_t)^2(1+\beta) + 4L^2\left(1 + \frac{1}{\beta}\right)\eta_{t-1}^2\right] \sum_{k=1}^{K} \mathbb{E}\|d_{t-1}^{(k)} - \bar{d}_{t-1}\|^2$$

$$+ 4KL^2\left(1 + \frac{1}{\beta}\right)\eta_{t-1}^2 \mathbb{E}\|\bar{d}_{t-1}\|^2 + \frac{4K\sigma^2}{b}\left(1 + \frac{1}{\beta}\right)a_t^2 + 8K\zeta^2\left(1 + \frac{1}{\beta}\right)a_t^2$$

$$+ 32L^2\left(1 + \frac{1}{\beta}\right)(I-1)a_t^2 \sum_{\bar{\ell}=\bar{t}_{s-1}}^{t-1} \eta_{\bar{\ell}}^2 \sum_{k=1}^{K} \mathbb{E}\|d_{\bar{\ell}}^{(k)} - \bar{d}_{\bar{\ell}}\|^2,$$

where inequality $(a)$ follows from the iterate update given in Step 10 of Algorithm 3 and inequality $(b)$ utilizes Lemma B.4. $\qquad\square$

Using the above Lemma C.8, we bound the accumulated gradient consensus error in the potential function's descent derived in Lemma C.7.

**Lemma C.9** (Accumulated Gradient Consensus Error). *For $\bar{t} \in [\bar{t}_{s-1}, \bar{t}_s - 1]$ with $s \in [S]$ we have*

$$\frac{33}{256K}\frac{(I-1)}{I} \sum_{t=\bar{t}_{s-1}}^{\bar{t}} \eta_t \sum_{k=1}^{K} \mathbb{E}\|d_t^{(k)} - \bar{d}_t\|^2 \leq \sum_{t=\bar{t}_{s-1}}^{\bar{t}} \frac{\eta_t}{64} \mathbb{E}\|\bar{d}_t\|^2 + \frac{\sigma^2 c^2}{64bL^2} \sum_{t=\bar{t}_{s-1}}^{\bar{t}} \eta_t^3 + \frac{\zeta^2 c^2}{32L^2}\frac{(I-1)}{I} \sum_{t=\bar{t}_{s-1}}^{\bar{t}} \eta_t^3.$$

*Proof.* First, from the statement of Lemma C.8, considering the coefficient of first term on the right hand side of the expression, we have:

$$(1-a_t)^2(1+\beta) + 4L^2\left(1 + \frac{1}{\beta}\right)\eta_{t-1}^2 \overset{(a)}{\leq} 1 + \beta + 4L^2\left(1 + \frac{1}{\beta}\right)\eta_{t-1}^2$$

$$\overset{(b)}{\leq} 1 + \frac{1}{I} + 4L^2(I+1)\eta_{t-1}^2$$

$$\overset{(c)}{\leq} 1 + \frac{1}{I} + \frac{I+1}{64I^2}$$

$$\overset{(d)}{\leq} 1 + \frac{33}{32I},$$

where inequality $(a)$ uses the fact that $(1-a_t)^2 \leq 1$; the second inequality $(b)$ follows from taking $\beta = 1/I$, inequality $(c)$ uses the bound $\eta_t \leq 1/16LI$ for all $t \in [T]$. Finally, the last inequality $(d)$ results by using the fact that we have $I + 1 \leq 2I$. Substituting in the statement of Lemma C.8 above, we get

$$\sum_{k=1}^{K}\mathbb{E}\|d_t^{(k)}-\bar{d}_t\|^2 \leq \left(1+\frac{33}{32I}\right)\sum_{k=1}^{K}\mathbb{E}\|d_{t-1}^{(k)}-\bar{d}_{t-1}\|^2 + 4KL^2\left(1+\frac{1}{\beta}\right)\eta_{t-1}^2\mathbb{E}\|\bar{d}_{t-1}\|^2 + \frac{4K\sigma^2}{b}\left(1+\frac{1}{\beta}\right)a_t^2$$

$$+ 8K\zeta^2\left(1+\frac{1}{\beta}\right)a_t^2 + 32L^2\left(1+\frac{1}{\beta}\right)(I-1)a_t^2\sum_{\ell=\bar{t}_{s-1}}^{t-1}\eta_\ell^2\sum_{k=1}^{K}\mathbb{E}\|d_\ell^{(k)}-\bar{d}_\ell\|^2.$$

$$\overset{(a)}{\leq} \left(1+\frac{33}{32I}\right)\sum_{k=1}^{K}\mathbb{E}\|d_{t-1}^{(k)}-\bar{d}_{t-1}\|^2 + 8KL^2I\eta_{t-1}^2\mathbb{E}\|\bar{d}_{t-1}\|^2 + \frac{8KI\sigma^2}{b}c^2\eta_{t-1}^4$$

$$+ 16KI\zeta^2c^2\eta_{t-1}^4 + 64L^2I^2c^2\eta_{t-1}^4\sum_{\ell=\bar{t}_{s-1}}^{t-1}\eta_\ell^2\sum_{k=1}^{K}\mathbb{E}\|d_\ell^{(k)}-\bar{d}_\ell\|^2$$

$$\overset{(b)}{\leq} \left(1+\frac{33}{32I}\right)\sum_{k=1}^{K}\mathbb{E}\|d_{t-1}^{(k)}-\bar{d}_{t-1}\|^2 + \frac{KL}{2}\eta_{t-1}\mathbb{E}\|\bar{d}_{t-1}\|^2 + \frac{K\sigma^2c^2}{2bL}\eta_{t-1}^3$$

$$+ \frac{K\zeta^2c^2}{L}\eta_{t-1}^3 + 64L^2I^2c^2\eta_{t-1}^4\sum_{\ell=\bar{t}_{s-1}}^{t-1}\eta_\ell^2\sum_{k=1}^{K}\mathbb{E}\|d_\ell^{(k)}-\bar{d}_\ell\|^2 \tag{22}$$

where $(a)$ follows from using $\beta=1/I$, the fact that $I+1\leq 2I$ and the definition of $a_t$ from Algorithm 3.

Note form Algorithm 3 that we have $d_t^{(k)}=\bar{d}_t$ for $t=\bar{t}_{s-1}$ with $s\in[S]$. This implies that for $t=\bar{t}_{s-1}$ with $s\in[S]$, we have, $\sum_{k=1}^{K}\|d_t^{(k)}-\bar{d}_t\|^2=0$. Applying (22) above recursively for $t\in[\bar{t}_{s-1}+1,\bar{t}_s-1]$ we get:

$$\sum_{k=1}^{K}\mathbb{E}\|d_t^{(k)}-\bar{d}_t\|^2 \leq \frac{KL}{2}\sum_{\ell=\bar{t}_{s-1}}^{t-1}\left(1+\frac{33}{32I}\right)^{t-1-\ell}\eta_\ell\mathbb{E}\|\bar{d}_\ell\|^2 + \frac{K\sigma^2c^2}{2bL}\sum_{\ell=\bar{t}_{s-1}}^{t-1}\left(1+\frac{33}{32I}\right)^{t-1-\ell}\eta_\ell^3$$

$$+ \frac{K\zeta^2c^2}{L}\sum_{\ell=\bar{t}_{s-1}}^{t-1}\left(1+\frac{3}{2I}\right)^{t-1-\ell}\eta_\ell^3 + 64L^2I^2c^2\sum_{\ell=\bar{t}_{s-1}}^{t-1}\left(1+\frac{33}{32I}\right)^{t-1-\ell}\eta_\ell^4\sum_{\bar{\ell}=\bar{t}_{s-1}}^{\ell}\eta_{\bar{\ell}}^2\sum_{k=1}^{K}\mathbb{E}\|d_{\bar{\ell}}^{(k)}-\bar{d}_{\bar{\ell}}\|^2$$

$$\overset{(a)}{\leq} \frac{KL}{2}\left(1+\frac{33}{32I}\right)^I\sum_{\ell=\bar{t}_{s-1}}^{t}\eta_\ell\mathbb{E}\|\bar{d}_\ell\|^2 + \frac{K\sigma^2c^2}{2bL}\left(1+\frac{33}{32I}\right)^I\sum_{\ell=\bar{t}_{s-1}}^{t}\eta_\ell^3$$

$$+ \frac{K\zeta^2c^2}{L}\left(1+\frac{33}{32I}\right)^I\sum_{\ell=\bar{t}_{s-1}}^{t}\eta_\ell^3 + 64L^2I^3c^2\left(\frac{1}{16LI}\right)^5\left(1+\frac{33}{32I}\right)^I\sum_{\bar{\ell}=\bar{t}_{s-1}}^{t}\eta_{\bar{\ell}}\sum_{k=1}^{K}\mathbb{E}\|d_{\bar{\ell}}^{(k)}-\bar{d}_{\bar{\ell}}\|^2$$

$$\overset{(b)}{\leq} \frac{3KL}{2}\sum_{\ell=\bar{t}_{s-1}}^{t}\eta_\ell\mathbb{E}\|\bar{d}_\ell\|^2 + \frac{3K\sigma^2c^2}{2bL}\sum_{\ell=\bar{t}_{s-1}}^{t}\eta_\ell^3 + \frac{3K\zeta^2c^2}{L}\sum_{\ell=\bar{t}_{s-1}}^{t}\eta_\ell^3$$

$$+ 192L^2I^3c^2\left(\frac{1}{16LI}\right)^5\sum_{\ell=\bar{t}_{s-1}}^{t}\eta_\ell\sum_{k=1}^{K}\mathbb{E}\|d_\ell^{(k)}-\bar{d}_\ell\|^2, \tag{23}$$

where inequality $(a)$ follows from the fact that $1+33/32I>1$ and $t-1-\ell\leq I$ for $t\in[\bar{t}_{s-1},\bar{t}_s-1]$ and $\ell\in[\bar{t}_{s-1},t]$ and inequality $(b)$ follows from the fact that $(1+33/32I)^I\leq e^{33/32}<3$ and $\eta_t\leq 1/16LI$ for all $t\in[T]$.

Multiplying (23) by $\eta_t$ and summing over $t=\bar{t}_{s-1}$ to $\bar{t}$ for $\bar{t}\in[\bar{t}_{s-1},\bar{t}_s-1]$ with $s\in[S]$

$$\sum_{t=\bar{t}_{s-1}}^{\bar{t}} \eta_t \sum_{k=1}^{K} \mathbb{E}\|d_t^{(k)} - \bar{d}_t\|^2 \le \frac{3KL}{2} \sum_{t=\bar{t}_{s-1}}^{\bar{t}} \eta_t \sum_{\ell=\bar{t}_{s-1}}^{t} \eta_\ell \mathbb{E}\|\bar{d}_\ell\|^2 + \frac{3K\sigma^2 c^2}{2bL} \sum_{t=\bar{t}_{s-1}}^{\bar{t}} \eta_t \sum_{\ell=\bar{t}_{s-1}}^{t} \eta_\ell^3$$

$$+ \frac{3K\zeta^2 c^2}{L} \sum_{t=\bar{t}_{s-1}}^{\bar{t}} \eta_t \sum_{\ell=\bar{t}_{s-1}}^{t} \eta_\ell^3 + 192L^2 I^3 c^2 \left(\frac{1}{16LI}\right)^5 \sum_{t=\bar{t}_{s-1}}^{\bar{t}} \eta_t \sum_{\ell=\bar{t}_{s-1}}^{t} \eta_\ell \sum_{k=1}^{K} \mathbb{E}\|d_\ell^{(k)} - \bar{d}_\ell\|^2$$

$$\overset{(a)}{\le} \frac{3KL}{2} \left(\sum_{t=\bar{t}_{s-1}}^{\bar{t}} \eta_t\right) \sum_{\ell=\bar{t}_{s-1}}^{\bar{t}} \eta_\ell \mathbb{E}\|\bar{d}_\ell\|^2 + \frac{3K\sigma^2 c^2}{2bL} \left(\sum_{t=\bar{t}_{s-1}}^{\bar{t}} \eta_t\right) \sum_{\ell=\bar{t}_{s-1}}^{\bar{t}} \eta_\ell^3$$

$$+ \frac{3K\zeta^2 c^2}{L} \left(\sum_{t=\bar{t}_{s-1}}^{\bar{t}} \eta_t\right) \sum_{\ell=\bar{t}_{s-1}}^{\bar{t}} \eta_\ell^3 + 192L^2 I^3 c^2 \left(\frac{1}{16LI}\right)^5 \left(\sum_{t=\bar{t}_{s-1}}^{\bar{t}} \eta_t\right) \sum_{\ell=\bar{t}_{s-1}}^{\bar{t}} \eta_\ell \sum_{k=1}^{K} \mathbb{E}\|d_\ell^{(k)} - \bar{d}_\ell\|^2$$

$$\overset{(b)}{\le} \frac{3K}{32} \sum_{t=\bar{t}_{s-1}}^{\bar{t}} \eta_t \mathbb{E}\|\bar{d}_t\|^2 + \frac{3K\sigma^2 c^2}{32bL^2} \sum_{t=\bar{t}_{s-1}}^{\bar{t}} \eta_t^3 + \frac{3K\zeta^2 c^2}{16L^2} \sum_{t=\bar{t}_{s-1}}^{\bar{t}} \eta_t^3$$

$$+ 192L^2 I^4 c^2 \left(\frac{1}{16LI}\right)^6 \sum_{t=\bar{t}_{s-1}}^{\bar{t}} \eta_t \sum_{k=1}^{K} \mathbb{E}\|d_t^{(k)} - \bar{d}_t\|^2$$

where inequality $(a)$ uses the fact that $t \in [\bar{t}_{s-1}, \bar{t}]$ and $(b)$ follows from the fact that we have $\eta_t \le 1/16LI$ for all $t \in [T]$. Rearranging the terms we get

$$\left[1 - 192L^2 I^4 c^2 \left(\frac{1}{16LI}\right)^6\right] \sum_{t=\bar{t}_{s-1}}^{\bar{t}} \eta_t \sum_{k=1}^{K} \mathbb{E}\|d_t^{(k)} - \bar{d}_t\|^2 \le \frac{3K}{32} \sum_{t=\bar{t}_{s-1}}^{\bar{t}} \eta_t \mathbb{E}\|\bar{d}_t\|^2$$

$$+ \frac{3K\sigma^2 c^2}{32bL^2} \sum_{t=\bar{t}_{s-1}}^{\bar{t}} \eta_t^3 + \frac{3K\zeta^2 c^2}{16L^2} \sum_{t=\bar{t}_{s-1}}^{\bar{t}} \eta_t^3$$

using the fact that $c \le 128L^2/bK$, $b \ge 1$, $K \ge 1$ and $I \ge 1$, we have $\left[1 - 192L^2 I^4 c^2 \left(\frac{1}{16LI}\right)^6\right] \ge \frac{4}{5}$, therefore, we get

$$\frac{33}{256K} \frac{(I-1)}{I} \sum_{t=\bar{t}_{s-1}}^{\bar{t}} \eta_t \sum_{k=1}^{K} \mathbb{E}\|d_t^{(k)} - \bar{d}_t\|^2 \le \sum_{t=\bar{t}_{s-1}}^{\bar{t}} \frac{\eta_t}{64} \mathbb{E}\|\bar{d}_t\|^2 + \frac{\sigma^2 c^2}{64bL^2} \sum_{t=\bar{t}_{s-1}}^{\bar{t}} \eta_t^3 + \frac{\zeta^2 c^2}{32L^2} \frac{(I-1)}{I} \sum_{t=\bar{t}_{s-1}}^{\bar{t}} \eta_t^3.$$

Hence, the lemma is proved. $\qquad\square$

### C.2.5 Proof of Theorem 3.1

Next, to prove Theorem 3.1 we first prove an intermediate theorem by utilizing Lemmas C.9 and C.7 derived above.

**Theorem C.10.** *Choosing the parameters as*

*(i)* $\bar{\kappa} = \dfrac{(bK)^{2/3}\sigma^{2/3}}{L}$,

*(ii)* $c = \dfrac{64L^2}{bK} + \dfrac{\sigma^2}{24\bar{\kappa}^3 LI} \overset{(i)}{=} L^2 \left(\dfrac{64}{bK} + \dfrac{1}{24(bK)^2 I}\right) \le \dfrac{128L^2}{bK}$,

*(iii)* *We choose* $\{w_t\}_{t=0}^{T}$ *as*

$$w_t = \max\left\{2\sigma^2, 4096L^3 I^3 \bar{\kappa}^3 - \sigma^2 t, \frac{c^3 \bar{\kappa}^3}{4096L^3 I^3}\right\} \overset{(i)(ii)}{\le} \sigma^2 \max\left\{2, 4096I^3(bK)^2 - t, \frac{512}{bKI^3}\right\}.$$

*Moreover, for any number of local updates, $I \geq 1$, batch sizes, $b \geq 1$, and initial batch size, $B \geq 1$, computed at individual WNs, STEM satisfies:*

$$\mathbb{E}\|\nabla f(\bar{x}_a)\|^2 \leq \left[\frac{32LI}{T} + \frac{2L}{(bK)^{2/3}T^{2/3}}\right](f(\bar{x}_1) - f^*) + \left[\frac{8bI^2}{BT} + \frac{bI}{2(bK)^{2/3}BT^{2/3}}\right]\sigma^2$$

$$+ \left[\frac{256^2I}{T} + \frac{64^2}{(bK)^{2/3}T^{2/3}}\right]\sigma^2 \log(T+1) + \left[\frac{256^2I}{T} + \frac{64^2}{(bK)^{2/3}T^{2/3}}\right]\zeta^2\frac{(I-1)}{I}\log(T+1).$$

*Proof.* Substituting the gradient consensus error derived in Lemma C.9 into the Potential function descent derived in Lemma C.7, we can write the descent of potential function for $\bar{t} \in [\bar{t}_{s-1}, \bar{t}_s - 1]$ with $s \in [S]$ as:

$$\mathbb{E}[\Phi_{\bar{t}+1} - \Phi_{\bar{t}_{s-1}}] \leq -\sum_{t=\bar{t}_{s-1}}^{\bar{t}} \left(\frac{27\eta_t}{64} - \frac{\eta_t^2 L}{2}\right)\mathbb{E}\|\bar{d}_t\|^2 - \sum_{t=\bar{t}_{s-1}}^{\bar{t}}\frac{\eta_t}{2}\mathbb{E}\|\nabla f(\bar{x}_t)\|^2$$

$$+ \frac{c^2\sigma^2}{32L^2}\sum_{t=\bar{t}_{s-1}}^{\bar{t}}\eta_t^3 + \frac{c^2\sigma^2}{64bL^2}\sum_{t=\bar{t}_{s-1}}^{\bar{t}}\eta_t^3 + \frac{c^2\zeta^2}{32L^2}\frac{(I-1)}{I}\sum_{t=\bar{t}_{s-1}}^{\bar{t}}\eta_t^3$$

$$\overset{(a)}{\leq} -\sum_{t=\bar{t}_{s-1}}^{\bar{t}}\frac{\eta_t}{2}\mathbb{E}\|\nabla f(\bar{x}_t)\|^2 + \frac{3c^2\sigma^2}{64L^2}\sum_{t=\bar{t}_{s-1}}^{\bar{t}}\eta_t^3 + \frac{c^2\zeta^2}{32L^2}\frac{(I-1)}{I}\sum_{t=\bar{t}_{s-1}}^{\bar{t}}\eta_t^3.$$

where $(a)$ follows from the fact that $\eta_t \leq 1/16LI$ for all $t \in [T]$ and $b \geq 1$. Taking $\bar{t} = \bar{t}_s - 1 = sI$, the above expression can be written as:

$$\mathbb{E}[\Phi_{\bar{t}_s} - \Phi_{\bar{t}_{s-1}}] \leq -\sum_{t=\bar{t}_{s-1}}^{\bar{t}_s-1}\frac{\eta_t}{2}\mathbb{E}\|\nabla f(\bar{x}_t)\|^2 + \frac{3c^2\sigma^2}{64L^2}\sum_{t=\bar{t}_{s-1}}^{\bar{t}_s-1}\eta_t^3 + \frac{c^2\zeta^2}{32L^2}\frac{(I-1)}{I}\sum_{t=\bar{t}_{s-1}}^{\bar{t}_s-1}\eta_t^3.$$

Summing over all the restarts, i.e, $s \in [S]$, we get:

$$\mathbb{E}[\Phi_{\bar{t}_S} - \Phi_{\bar{t}_0}] \leq -\sum_{t=\bar{t}_0}^{\bar{t}_S-1}\frac{\eta_t}{2}\mathbb{E}\|\nabla f(\bar{x}_t)\|^2 + \frac{3c^2\sigma^2}{64L^2}\sum_{t=\bar{t}_0}^{\bar{t}_S-1}\eta_t^3 + \frac{c^2\zeta^2}{32L^2}\frac{(I-1)}{I}\sum_{t=\bar{t}_0}^{\bar{t}_S-1}\eta_t^3.$$

Assuming that $T = SI$, then from the definition of $\bar{t}_s$ that $\bar{t}_0 = 1$ and $\bar{t}_S = SI + 1 = T + 1$, we get

$$\sum_{t=1}^{T}\frac{\eta_t}{2}\mathbb{E}\|\nabla f(\bar{x}_t)\|^2 \leq \mathbb{E}[\Phi_1 - \Phi_{T+1}] + \frac{3c^2\sigma^2}{64L^2}\sum_{t=1}^{T}\eta_t^3 + \frac{c^2\zeta^2}{32L^2}\frac{(I-1)}{I}\sum_{t=1}^{T}\eta_t^3$$

$$\overset{(a)}{\leq} f(\bar{x}_1) - f^* + \frac{bK}{64L^2}\frac{\mathbb{E}\|\bar{e}_1\|^2}{\eta_0} + \frac{3c^2\sigma^2}{64L^2}\sum_{t=1}^{T}\eta_t^3 + \frac{c^2\zeta^2}{32L^2}\frac{(I-1)}{I}\sum_{t=1}^{T}\eta_t^3$$

$$\overset{(b)}{\leq} f(\bar{x}_1) - f^* + \frac{\sigma^2}{64L^2}\frac{b}{B\eta_0} + \frac{3c^2\sigma^2}{64L^2}\sum_{t=1}^{T}\eta_t^3 + \frac{c^2\zeta^2}{32L^2}\frac{(I-1)}{I}\sum_{t=1}^{T}\eta_t^3. \quad (24)$$

where $(a)$ follows from the fact that $f^* \leq \Phi_{T+1}$ and $(b)$ results from application of Lemma C.3.

First, let us consider the last term of the (24) above, we have from the definition of the stepsize $\eta_t$

$$\sum_{t=1}^{T}\eta_t^3 = \sum_{t=1}^{T}\frac{\bar{\kappa}^3}{w_t + \sigma^2 t}$$

$$\overset{(a)}{\leq} \sum_{t=1}^{T}\frac{\bar{\kappa}^3}{\sigma^2 + \sigma^2 t}$$

$$= \frac{\bar{\kappa}^3}{\sigma^2}\sum_{t=1}^{T}\frac{1}{1+t}$$

$$\overset{(b)}{\leq} \frac{\bar{\kappa}^3}{\sigma^2} \ln(T+1). \tag{25}$$

where inequality $(a)$ above follows from the fact that we have $w_t \geq 2\sigma^2 > \sigma^2$ and inequality $(b)$ follows from the application of Lemma B.3.

Substituting (25) in (24), dividing both sides by $T$ and using the fact that $\eta_t$ is non-increasing in $t$ we have

$$\frac{1}{T}\sum_{t=1}^{T}\mathbb{E}\|\nabla f(\bar{x}_t)\|^2 \leq \frac{2(f(\bar{x}_1) - f^*)}{\eta_T T} + \frac{1}{\eta_T T}\frac{\sigma^2}{32L^2}\frac{b}{B\eta_0} + \frac{1}{\eta_T T}\frac{3c^2\bar{\kappa}^3}{32L^2}\log(T+1)$$

$$+ \frac{1}{\eta_T T}\frac{c^2\bar{\kappa}^3}{16L^2}\frac{\zeta^2}{\sigma^2}\frac{(I-1)}{I}\log(T+1)$$

$$\overset{(a)}{\leq} \frac{2(f(\bar{x}_1) - f^*)}{\eta_T T} + \frac{1}{\eta_T T}\frac{\sigma^2}{32L^2}\frac{b}{B\eta_0} + \frac{1}{\eta_T T}\frac{c^2\bar{\kappa}^3}{4L^2}\log(T+1)$$

$$+ \frac{1}{\eta_T T}\frac{c^2\bar{\kappa}^3}{4L^2}\frac{\zeta^2}{\sigma^2}\frac{(I-1)}{I}\log(T+1). \tag{26}$$

where $(a)$ above utilizes the fact that $1/16 < 3/32 < 1/4$.

Now considering each term of (26) above separately and using the definition of $\eta_t = \dfrac{\bar{\kappa}}{(w_t + \sigma^2 t)^{1/3}}$ we get from the coefficient of the first term:

$$\frac{1}{\eta_T T} = \frac{(w_T + \sigma^2 T)^{1/3}}{\bar{\kappa} T} \overset{(a)}{\leq} \frac{w_T^{1/3}}{\bar{\kappa} T} + \frac{\sigma^{2/3}}{\bar{\kappa} T^{2/3}} \overset{(b)}{\leq} \frac{16LI}{T} + \frac{L}{(bK)^{2/3}T^{2/3}}. \tag{27}$$

where inequality $(a)$ follows from identity $(x+y)^{1/3} \leq x^{1/3} + y^{1/3}$ and inequality $(b)$ follows from the definition of $\bar{\kappa}$ and $w_T$

$$w_T = \max\left\{2\sigma^2, 4096L^3I^3\bar{\kappa}^3 - \sigma^2 T, \frac{c^3\bar{\kappa}^3}{4096L^3I^3}\right\} \leq \sigma^2 \max\left\{2, 4096I^3(bK)^2 - T, \frac{512}{bKI^3}\right\},$$

where we used $4096L^3I^3\bar{\kappa}^3 > 4096L^3I^3\bar{\kappa}^3 - \sigma^2 T \geq \max\left\{2\sigma^2, \frac{c^3\bar{\kappa}^3}{4096L^3I^3}\right\}$. Note that this choice of $w_T$ captures the worst case guarantees for STEM.

Now, let us consider the second term of (26), we have from the definition of $\eta_0$ and $\eta_T$

$$\frac{1}{\eta_T T}\frac{\sigma^2}{32L^2}\frac{b}{B\eta_0} \leq \left(\frac{16LI}{T} + \frac{L}{(bK)^{2/3}T^{2/3}}\right) \times \frac{\sigma^2}{32L^2} \times \frac{bw_0^{1/3}}{B\bar{\kappa}}$$

$$\overset{(a)}{\leq} \left(\frac{16LI}{T} + \frac{L}{(bK)^{2/3}T^{2/3}}\right) \times \frac{\sigma^2}{32L^2} \times \frac{16LIb}{B}$$

$$\overset{(b)}{\leq} \frac{8bI^2}{BT}\sigma^2 + \frac{bI}{(bK)^{2/3}BT^{2/3}}\frac{\sigma^2}{2}. \tag{28}$$

where inequality $(a)$ follows from the identity $(x+y)^{1/3} \leq x^{1/3} + y^{1/3}$ and $(b)$ follows from the definition of $\bar{\kappa}$ and using $w_0 \leq 4096L^3I^3\bar{\kappa}^3$ and $w_T \leq 4096L^3I^3\bar{\kappa}^3$ (Similar to the approach in (27) this choice of $w_0$ and $w_T$ capture the worst case convergence guarantees for STEM.)

Finally, considering the term $\frac{1}{\eta_T T}\frac{c^2\bar{\kappa}^3}{4L^2}$ common to the last two terms in (26) above, we have from the definition of the stepsize, $\eta_t$,

$$\frac{1}{\eta_T T}\frac{c^2\bar{\kappa}^3}{4L^2} \leq \left(\frac{16LI}{T} + \frac{L}{(bK)^{2/3}T^{2/3}}\right) \times \left(\frac{128L^2}{bK}\right)^2 \times \frac{(bK)^2\sigma^2}{L^3} \times \frac{1}{4L^2}$$

$$\overset{(a)}{\leq} 256^2\sigma^2\frac{I}{T} + 64^2\sigma^2\frac{1}{(bK)^{2/3}T^{2/3}}. \tag{29}$$

where inequality $(a)$ follows from the identity $(x+y)^{1/3} \leq x^{1/3} + y^{1/3}$ and $(b)$ again uses $w_T \leq 4096L^3I^3\bar{\kappa}^3$ along with the definition of $\bar{\kappa}$ and $c$.

Finally, substituting the bounds obtained in (27), (28) and (29) into (26), we get

$$\mathbb{E}\|\nabla f(\bar{x}_a)\|^2 \le \left[\frac{32LI}{T} + \frac{2L}{(bK)^{2/3}T^{2/3}}\right](f(\bar{x}_1) - f^*) + \left[\frac{8bI^2}{BT} + \frac{bI}{2(bK)^{2/3}BT^{2/3}}\right]\sigma^2$$
$$+ \left[\frac{256^2 I}{T} + \frac{64^2}{(bK)^{2/3}T^{2/3}}\right]\sigma^2 \log(T+1) + \left[\frac{256^2 I}{T} + \frac{64^2}{(bK)^{2/3}T^{2/3}}\right]\zeta^2 \frac{(I-1)}{I}\log(T+1).$$

Hence, the theorem is proved. □

Next, using Theorem C.10 we prove Theorem 3.1.

**Theorem C.11** (Theorem 3.1: Trade-off: Local Updates vs Batch Sizes)**.** *With the parameters chosen according to Theorem C.10 and for any $\nu \in [0,1]$ at each WN we set the total number of local updates as $I = \mathcal{O}\big((T/K^2)^{\nu/3}\big)$, batch size, $b = \mathcal{O}\big((T/K^2)^{1/2-\nu/2}\big)$, and the initial batch size, $B = bI$. Then* **STEM** *satisfies:*

*(i) We have:*

$$\mathbb{E}\|\nabla f(\bar{x}_a)\|^2 = \mathcal{O}\left(\frac{f(\bar{x}_1) - f^*}{K^{2\nu/3}T^{1-\nu/3}}\right) + \tilde{\mathcal{O}}\left(\frac{\sigma^2}{K^{2\nu/3}T^{1-\nu/3}}\right) + \tilde{\mathcal{O}}\left(\frac{(I-1)}{I} \times \frac{\zeta^2}{K^{2\nu/3}T^{1-\nu/3}}\right).$$

*(ii) Sample Complexity: To achieve an $\epsilon$-stationary point* **STEM** *requires at most $\mathcal{O}(\epsilon^{-3/2})$ gradient computations. This implies that each WN requires at most $\mathcal{O}(K^{-1}\epsilon^{-3/2})$ gradient computations, thereby achieving linear speedup with the number of WNs present in the network.*

*(iii) Communication Complexity: To achieve an $\epsilon$-stationary point* **STEM** *requires at most $\mathcal{O}(\epsilon^{-1})$ communication rounds.*

*Proof.* The proof of statement (i) follows from the statement of Theorem C.10 and substituting the values of parameters $B$, $I$ and $b$ in the expression. First, replacing $B = bI$ in the statement of Theorem C.10 yields

$$\mathbb{E}\|\nabla f(\bar{x}_a)\|^2 \le \left[\frac{32LI}{T} + \frac{2L}{(bK)^{2/3}T^{2/3}}\right](f(\bar{x}_1) - f^*) + \left[\frac{8I}{T} + \frac{1}{2(bK)^{2/3}T^{2/3}}\right]\sigma^2$$
$$+ \left[\frac{256^2 I}{T} + \frac{64^2}{(bK)^{2/3}T^{2/3}}\right]\sigma^2 \log(T+1) + \left[\frac{256^2 I}{T} + \frac{64^2}{(bK)^{2/3}T^{2/3}}\right]\zeta^2 \frac{(I-1)}{I}\log(T+1).$$

Then using the fact that $I = \mathcal{O}\big((T/K^2)^{\nu/3}\big)$ and $b = \mathcal{O}\big((T/K^2)^{1/2-\nu/2}\big)$ yields the expression of statement $(i)$.

Next, we compute the computation and communication complexity of the algorithm.

- *Sample Complexity* [Theorem C.11(ii)]: From the statement of Theorem C.11(i), total iterations required to achieve an $\epsilon$-stationary point are:

$$\tilde{\mathcal{O}}\left(\frac{1}{K^{2\nu/3}T^{1-\nu/3}}\right) = \epsilon \quad \Rightarrow \quad T = \tilde{\mathcal{O}}\left(\frac{1}{K^{2\nu/(3-\nu)}\epsilon^{3/(3-\nu)}}\right). \tag{30}$$

  In each iteration, each WN computes $2b$ stochastic gradients, therefore, the total gradient computations at each WN are $2bT$. Using $b = \mathcal{O}\big((T/K^2)^{1/2-\nu/2}\big)$, we get the total gradient computations required at each WN as:

$$bT = \tilde{\mathcal{O}}\left(\frac{T^{3/2-\nu/2}}{K^{1-\nu}}\right) \overset{(30)}{=} \tilde{\mathcal{O}}\left(\frac{1}{K\epsilon^{3/2}}\right)$$

  This implies that the sample complexity is $\tilde{\mathcal{O}}(\epsilon^{-3/2})$.

- *Communication Complexity* [Theorem C.11(iii)]: The total rounds of communication to achieve an $\epsilon$-stationary point are $T/I$, with $I = \mathcal{O}\big((T/K^2)^{\nu/3}\big)$ and $T$ given in (30), therefore, we have the communication complexity as:

$$\frac{T}{I} = \tilde{\mathcal{O}}\big(T^{1-\nu/3}K^{2\nu/3}\big) \overset{(30)}{=} \tilde{\mathcal{O}}\left(\frac{1}{\epsilon}\right).$$

Hence, the theorem is proved. □

**Corollary 2** (FedSTEM: Local Updates)**.** *With the choice of parameters given in Theorem C.10. At each WN, setting constant batch size, $b \geq 1$, number of local updates, $I = (T/b^2 K^2)^{1/3}$, and the initial batch size, $B = bI$. Then STEM satisfies the following:*

(i) *We have:*

$$\mathbb{E}\|\nabla f(\bar{x}_a)\|^2 = \mathcal{O}\left(\frac{f(\bar{x}_1) - f^*}{(bK)^{2/3}T^{2/3}}\right) + \tilde{\mathcal{O}}\left(\frac{\sigma^2}{(bK)^{2/3}T^{2/3}}\right) + \tilde{\mathcal{O}}\left(\frac{\zeta^2}{(bK)^{2/3}T^{2/3}}\right).$$

(ii) *Sample Complexity: To achieve an $\epsilon$-stationary point FedSTEM requires at most $\tilde{\mathcal{O}}(\epsilon^{-3/2})$ gradient computations while achieving linear speedup with the number of WNs.*

(iii) *Communication Complexity: To achieve an $\epsilon$-stationary point FedSTEM requires at most $\tilde{\mathcal{O}}(\epsilon^{-1})$ communication rounds.*

*Proof.* The proof of statement (i) follows from substituting the values of the parameters $b$, $I$ and $B$ as defined in the statement of the Corollary in the statement of Theorem C.10.

Next, we compute the sample and communication complexity of the algorithm.

- *Sample Complexity:* From the statement of Corollary 2(i), total iterations, $T$, required to achieve an $\epsilon$-stationary point are:

$$\tilde{\mathcal{O}}\left(\frac{1}{(bK)^{2/3}T^{2/3}}\right) = \epsilon \qquad \Rightarrow \qquad T = \tilde{\mathcal{O}}\left(\frac{1}{bK\epsilon^{3/2}}\right). \tag{31}$$

At each iteration the algorithm computes $2b$ stochastic gradients. Therefore, the total number of gradient computations required at each WN are of the order of $2bT$, which is $\tilde{\mathcal{O}}(K^{-1}\epsilon^{-3/2})$. Therefore, the sample complexity of the algorithm is $\tilde{\mathcal{O}}(\epsilon^{-3/2})$.

- *Communication Complexity:* Total rounds of communication to achieve an $\epsilon$-stationary point is $T/I$, therefore we have from the choice of $I$ that

$$\frac{T}{I} = \tilde{\mathcal{O}}\big((bK)^{2/3}T^{2/3}\big) \stackrel{(31)}{=} \tilde{\mathcal{O}}\left(\frac{1}{\epsilon}\right).$$

Hence, the corollary is proved. □

An alternate design choice for the algorithm is to design large batch-size gradients and communicate more often. The next corollary captures this idea.

**Corollary 3** (Corollary 1: Minibatch STEM)**.** *With the choice of parameters given in Theorem C.10. At each WN, choosing the number of local updates, $I = 1$, the batch size, $b = T^{1/2}/K$, and the initial batch size, $B = bI$. Then STEM satisfies:*

(i) *We have:*

$$\mathbb{E}\|\nabla f(\bar{x}_a)\|^2 = \mathcal{O}\left(\frac{f(\bar{x}_1) - f^*}{T}\right) + \tilde{\mathcal{O}}\left(\frac{\sigma^2}{T}\right).$$

(ii) *Sample Complexity: To achieve an $\epsilon$-stationary point Minibatch STEM requires at most $\tilde{\mathcal{O}}(\epsilon^{-3/2})$ gradient computations while achieving linear speedup with the number of WNs.*

(iii) *Communication Complexity: To achieve an $\epsilon$-stationary point Minibatch STEM requires at most $\tilde{\mathcal{O}}(\epsilon^{-1})$ communication rounds.*

*Proof.* The proof of statement (i) follows from substituting the values of the parameters $b$, $I$ and $B$ given in the statement of the Corollary in the statement of Theorem C.10.

Next, we compute the sample and communication complexity of the algorithm.

- *Sample Complexity:* From the statement of Corollary 3(i), total iterations, $T$, required to achieve an $\epsilon$-stationary point are:

$$\tilde{\mathcal{O}}\left(\frac{I}{T}\right) = \epsilon \qquad \Rightarrow \qquad T = \tilde{\mathcal{O}}\left(\frac{I}{\epsilon}\right). \tag{32}$$

In each iteration, each WN computes $2b$ stochastic gradients, therefore, the total gradient computations at each WN are $2bT$. Using the fact that $b = \mathcal{O}\left(\frac{T^{1/2}}{I^{3/2}K}\right)$. The total gradients computed at each WN to reach an $\epsilon$-stationary point are:

$$\tilde{\mathcal{O}}\left(\frac{I}{\epsilon} \times \frac{I^{1/2}}{\epsilon^{1/2}I^{3/2}K}\right) = \tilde{\mathcal{O}}\left(\frac{1}{K\epsilon^{3/2}}\right).$$

Therefore, the communication complexity if $\tilde{\mathcal{O}}(\epsilon^{-3/2})$.

- *Communication Complexity:* The total rounds of communication required to reach an $\epsilon$-stationary point are $T/I$, therefore we have

$$\frac{T}{I} \stackrel{(32)}{=} \tilde{\mathcal{O}}\left(\frac{1}{\epsilon}\right).$$

Hence, the corollary is proved. □