# OpenReview forum: "STEM: A Stochastic Two-Sided Momentum Algorithm Achieving Near-Optimal Sample and Communication Complexities for Federated Learning"
_NeurIPS.cc/2021/Conference — NeurIPS 2021 Poster_

### Official Review · Reviewer_9RXX · 2021-07-13

**Rating:** 6
**Confidence:** 4

**Summary:**

This paper proposes and analyzes an optimization algorithm (STEM) with two-sided momentum and local steps for (non-convex) Federated Learning. The authors prove its convergence and
They also claim that it improves over the previous methods in terms of join communication and sample complexity.

**Limitations And Societal Impact:**

Limitations of the work were mostly addressed. Negative societal impact concerns are not applicable due to the theoretical nature of the paper.

**Main Review:**

This work considers a problem of non-convex federated optimization, where communication is the bottleneck. A local stochastic gradient method with momentum at the server and workers is suggested. STEM enjoys $\tilde{\mathcal{O}}(\epsilon^{-3/2})$ sample and $\tilde{\mathcal{O}}(\epsilon^{-1})$ communication complexity if mini-batch size and number of local updates are chosen appropriately (e.g. as $\mathcal{O}(\epsilon^{-1/2})$). The authors show a trade-off between the mini-batch sizes and local update frequency.
Experimental comparison to FedAvg and SCAFFOLD on two image classification tasks (with convolutional neural networks) confirmed the benefits of the proposed method.

Related work was mostly adequately cited and compared with.

Overall, the paper is well-organized and leaves an impression of good work, but I think that in its current stage, it is not ready for publication.
The first reason is the theoretical contributions. The method does not suggest a very novel technique because algorithms with momentum and local steps were previously analyzed in the federated setting.
The so-called “near-optimal” sample and communication complexities are obtained for mini-batches and number of local updates of the order of $\mathcal{O}(\epsilon^{-1/2})$, which means that for achieving an $\epsilon$-stationary point for $\epsilon = 10^{-10}$ (which corresponds to gradient norm $\||\nabla f(x_t)\|| \leq 10^{-5}$) the method requires very big mini-batches or the number of local updates, which seems not very practical. At the same time, for methods like SCAFFOLD and MIME, it is not required. That is why I think that theoretical comparison is not completely fair from the practical side.

The second issue is the lack of experimental comparison to the methods presented in the comparison Table 1. The chosen benchmarking methods (FedAvg and SCAFFOLD) are the easiest competitors to STEM because the first one is the most basic baseline, while the second is more suited for convex problems in the cross-silo settings. Both of them also have worse theoretical complexities. To show real practical benefits of the proposed method, it is required to compare against FedPD and MIME/FedGLOMO as they have the same sample or communication complexities to STEM. Furthermore, the hyperparameter tunning seems very limited (just two values for $\overline{\kappa}$ and $\overline{c}$). I also recommend multiple runs for every experiment to show mean values and deviations as the methods are stochastic. This is a basic standard to show the consistent superiority of the proposed approach over the baselines.

In addition, real experiments supporting the point on the trade-off between mini-batch size and the number of local updates are quite limited to basically a couple of values (of $b$ and $I$), which does not seem enough to show this effect and provide valuable guidelines for practitioners. If the computational resources are the problem, I suggest the authors using a simpler non-convex model [2]. Showing plots with gradient norm values would be also more consistent with the theory in this case.

To conclude, I think that the current results are not very important nor valuable for the field.

Unfortunately, chosen notation (e.g., for stochastic gradients) complicates the algorithm description and makes convergence proofs very hard to follow and understand. I greatly encourage authors to simplify it. E.g. for local gradient estimates $\nabla f^{kj}$ could be used in case of finite sum losses $f^{(k)} = \sum_{j=1}^J f^{kj}$.

*Minor issues/comments:*

I think that the authors missed an important reference [1].

There is a typo in the Appendix on page 18 in equation (8). In the RHS of the (b) inequality, there should be superscript (j) after the last $\nabla$ symbol.

[1] R. Sashank, et al. "Adaptive federated optimization." arXiv preprint arXiv:2003.00295 (2020)

[2] L. Zhao, et al. "From Convex to Nonconvex: A Loss Function Analysis for Binary Classification," 2010 IEEE International Conference on Data Mining Workshops, 2010, pp. 1281-1288, doi: 10.1109/ICDMW.2010.57

**Time Spent Reviewing:**

10

---

> ### Author Response · Authors · 2021-08-10
> **Clarification on Batch Size and Number of Local Updates**
>
> >**Your Comment:** 1. Overall, the paper is well-organized and leaves an impression of good work, but I think that in its current stage, it is not ready for publication. The first reason is the theoretical contributions. The method does not suggest a very novel technique because algorithms with momentum and local steps were previously analyzed in the federated setting. The so-called "near-optimal” sample and communication complexities are obtained for mini-batches and number of local updates of the order of $\mathcal{O}(\epsilon^{-1/2})$, which means that for achieving an $\epsilon$-stationary point for  (which corresponds to gradient norm $\|| \nabla f(x_t)\||^2 \leq 10^{-5}$) the method requires very big mini-batches or the number of local updates, which seems not very practical. At the same time, for methods like SCAFFOLD and MIME, it is not required. That is why I think that theoretical comparison is not completely fair from the practical side.
>
> **Our Response:** Thank you for the positive comment on the organization of our work. Please see below for our response to your comments about originality, and about the size of the mini-batch.
>
> First,  we respectfully disagree with the reviewer's comment about the novelty of this work. Note that we completely agree that the gradient estimator itself is not new, as we have already pointed out in the original paper (please see discussion from Line 137 to 147 in the original paper). However, to our knowledge, it is not clear how to make it work in the FL setting so that optimal complexities are achieved. Importantly, despite the fact that there are some existing FL algorithms that utilize momentum in their local updates [1, 2, 3], to our knowledge, none of them has achieved the following three desired properties simultaneously:
>
> - Near-optimal communication rounds to achieve a desired solution.
> - Optimal computation complexity.
> -  Linear speed-up with the number of nodes.
>
> Note that achieving all the three objectives above is even more challenging when the data across the nodes is heterogeneous. The reason is that, with heterogeneous data, the local iterates at each worker node tend to drift further from the global optimal solution with a larger number of local updates. Moreover, the analysis presented in the paper reveals a fundamental trade-off between the batch sizes and the number of local updates required to achieve these optimal guarantees, and these tradeoffs have not been discovered before. More importantly, STEM uniformly improves the guarantees provided by any algorithm using momentum directions.
>
> Next, we would like to clarify the usage of a large number of local updates/batch sizes.
>
> Usually, the goal of FL algorithms is to minimize the total communication rounds required to achieve a desired convergence accuracy, which is referred to as *communication complexity* in the literature.
> To achieve an improved communication complexity, it is often desired that nodes perform multiple local computations before parameter sharing. In fact, it is an advantage of STEM that it allows the nodes to perform a larger number of local updates (or larger batch gradients, or trade-off batch gradients with a number of local updates) to achieve a near-optimal communication complexity. Importantly, STEM achieves near-optimal communication complexity while attaining optimal **overall** sample complexity. More specifically, even if nodes are performing multiple local updates within each communication round, the total samples required throughout the entire convergence process is $\mathcal{O}(\epsilon^{-3/2})$, which is optimal for stochastic non-convex problems. Moreover, note that even if we choose a **constant** number of local updates with fixed batch size (as suggested by the reviewer), our algorithm still achieves better sample and communication complexities of $\mathcal{O}(\epsilon^{-3/2})$ compared to those of SCAFFOLD, and it is the same as MIME. Please note that this point has already been mentioned in the original paper, Remark 3, where we commented that if both the batch size and iteration number are chosen as $\mathcal{O}(1)$, the total number of communication rounds remains to be $\mathcal{O}(\epsilon^{-3/2})$.
>
>  $[1]$ Yu et al, On the linear speedup analysis of communication efficient momentum sgd for distributed non-convex optimization
>
>  $[2]$ Das et al, Improved convergence rates for non-convex federated learning with compression
>
>  $[3]$  Karimireddy, Mime: Mimicking centralized stochastic algorithms in federated learning}
>
> > **Your Comment:** 2. The second issue is the lack of experimental comparison to the methods presented in the comparison Table 1. The chosen benchmarking methods (FedAvg and SCAFFOLD) are the easiest competitors to STEM because the first one is the most basic baseline, while the second is more suited for convex problems in the cross-silo settings. Both of them also have worse theoretical complexities. To show real practical benefits of the proposed method, it is required to compare against FedPD and MIME/FedGLOMO as they have the same sample or communication complexities to STEM. Furthermore, the hyperparameter tunning seems very limited (just two values for $\bar{\kappa}$ and $\bar{c}$). I also recommend multiple runs for every experiment to show mean values and deviations as the methods are stochastic. This is a basic standard to show the consistent superiority of the proposed approach over the baselines.
>
> **Our Response:** We thank the reviewer for raising this concern. In the revised version of the paper, we will add experiments with comparison of STEM with state-of-the-art algorithms with all the parameter settings described in detail. Moreover, we will also add the comparison for STEM with other algorithms for the Shakespeare dataset. A summary of the experiments is included in the **General Response** section. We also note that the parameters $\bar{\kappa}$ and $\bar{c}$ were not only tuned for two parameter values but for the range of values described in lines 261 and 262 of the paper.
>
> > **Your Comment:** 3. In addition, real experiments supporting the point on the trade-off between mini-batch size and the number of local updates are quite limited to basically a couple of values (of $b$ and $I$), which does not seem enough to show this effect and provide valuable guidelines for practitioners. If the computational resources are the problem, I suggest the authors using a simpler non-convex model [2]. Showing plots with gradient norm values would be also more consistent with the theory in this case.
> >
> > To conclude, I think that the current results are not very important nor valuable for the field.
>
> **Our Response:** In the new set of experiments we have conducted experiments with additional batch-size choices. We have included a summary of the results at the beginning of the response in the **General Response** section. In our opinion, the gradient norm is not the important factor in capturing the performance of the FL algorithms, therefore, we choose to present the training and testing performance of the algorithms on real datasets.
>
> > **Your Comment:** 4. Unfortunately, chosen notation (e.g., for stochastic gradients) complicates the algorithm description and makes convergence proofs very hard to follow and understand. I greatly encourage authors to simplify it. E.g. for local gradient estimates $\nabla f^{kj}$  could be used in case of finite sum losses $f^{(k)} = \sum_{j = 1}^J f^{kj}$.
>
> **Our Response:** We thank the reviewer for raising this concern. In the revised version of the manuscript, we will make an attempt to simplify the notation.
>
>
> > Minor issues/comments:
> >
> > **Your Comment:** 5. I think that the authors missed an important reference [1].
>
> **Our Response:** Thanks, we will add the reference to the revised version of the paper.
>
> >**Your Comment:** 6. There is a typo in the Appendix on page 18 in equation (8). In the RHS of the (b) inequality, there should be superscript $(j)$ after the last symbol.
>
> **Our Response:** Thanks for catching this, we will correct it in the revised version of the paper.
>
> $[1]$ R. Sashank, et al. Adaptive federated optimization
>
> $[2]$ L. Zhao, et al. From Convex to Nonconvex: A Loss Function Analysis for Binary Classification,

---

> > ### Comment · Reviewer_9RXX · 2021-08-11
> > **Rebuttal response**
> >
> > I would like to appreciate the authors' response to my critique.
> >
> > **1.** The first answer clarified my confusion, so I would like to raise my rating of the paper to at least 5 (also taking into account additional experiments). I apologize for missing this important point. Please add this result to Table 1 with the comparison of the methods.
> >
> > **2.** It is nice to see that the authors conducted additional experiments. Nevertheless, I can not currently vote for clear acceptance because the work lacks details on hyperparameters tuning for all considered methods. So, it is not clear whether the benefits of the proposed approach are due to the novel algorithm or cherry-picked parameters. A proper reproducible experimental report (like in [1]) could greatly strengthen the results and be much more convincing. I think that it is essential because you consider a non-convex (Deep Learning) setting where it is not possible to use theoretical parameters. Besides, the constants coming from the convergence analysis seem very unpractical for other settings (where analytical expressions may be used).
> >
> > I have a few additional questions:
> >
> > **3.** Why you did not compare with the methods proposed in the work [1]?
> >
> > **4.** Could you please clarify what version of the MIME algorithm did you use in your experiments?
> >
> > **5.** What is meant by _"near-optimal complexity"_? I think it should be mentioned in the paper.
> >
> > ***
> >
> > [1] R. Sashank, et al. "Adaptive federated optimization." arXiv preprint arXiv:2003.00295 (2020)

---

> > > ### Author Response · Authors · 2021-08-14
> > > **Hyperparameter Selection**
> > >
> > > > I would like to appreciate the authors' response to my critique.
> > >
> > > We thank the reviewer for the positive comment. Below we address the comments in a point-by-point manner.
> > >
> > > > **Your comment** 1. The first answer clarified my confusion, so I would like to raise my rating of the paper to at least 5 (also taking into account additional experiments). I apologize for missing this important point. Please add this result to Table 1 with the comparison of the methods.
> > >
> > > **Our response:** We thank the reviewer for the suggestion, we will add the result to Table 1.
> > >
> > > > **Your comment** 2. It is nice to see that the authors conducted additional experiments. Nevertheless, I can not currently vote for clear acceptance because the work lacks details on hyperparameters tuning for all considered methods. So, it is not clear whether the benefits of the proposed approach are due to the novel algorithm or cherry-picked parameters. A proper reproducible experimental report (like in [1]) could greatly strengthen the results and be much more convincing. I think that it is essential because you consider a non-convex (Deep Learning) setting where it is not possible to use theoretical parameters. Besides, the constants coming from the convergence analysis seem very unpractical for other settings (where analytical expressions may be used).
> > >
> > > **Our response:** We thank the reviewer for the positive comment. We did not include the details of hyperparameter selection in the interest of space with the response to the reviewers. We would like to clarify that the advantage of the proposed algorithm is not because of cherry-picked parameters, we have tuned the parameters for all the algorithms for best performance. Below, we provide the details of parameter selection for each algorithm for both Shakespeare and CIFAR-10 datasets.
> > >
> > > - Shakespeare Dataset: The Shakespeare dataset considers a classification problem of next character prediction with 80 classes in total. We associate with each node a different speaking role (same setting as in FedProx). We have a total of 143 nodes with a total of 517,106 samples that are split among 143 nodes with each node having 3616 samples on average. We randomly split the data on each node into an 80% training set and a 20% testing set. We randomly sample 10 nodes out of 143 for the training purpose. For this task, we utilize a 2-Layer LSTM network with 100 hidden units and an 8-D embedding layer at each node. For each algorithm, we select a batch size of 128 and tune for the rest of the hyperparameters whose details will be presented shortly below.
> > >
> > >
> > > - CIFAR-10 Dataset: Note that for the CIFAR-10 dataset, we choose the same model and data settings as described in the experiment section of the paper. Moreover, in the experiments, we included an additional very-high heterogeneity setting where each node can access data from only one or two classes. We also conducted experiments for an additional batch-size setting of 128 as suggested by the reviewers. Next, we describe the hyper-parameter selection for each algorithm.
> > >
> > > For all the algorithms, parameters are chosen based on the values given in their respective papers. The details are given below:
> > >
> > > - For FedProx and FedDyn we fix the regularization constant to be 0.1. The momentum parameters for FedGLOMO and MIME are set based on the choices given in the respective papers. Specifically, for FedGLOMO we choose the parameter $\beta_k = 0.2$ and design the momentum gradient using a damping factor given in Appendix A.4 of FedGLOMO. Moreover, for MIME we choose the momentum parameter as 0.9.
> > >
> > > - For the proposed STEM algorithm, recall that we had the step-size $\eta_t = \bar{\kappa}/(w_t +  \sigma^2 t)^{1/3}$ and the momentum parameter $a_t = c \eta_t^2$. The step-size is used to update the iterates whereas the momentum parameter is used to construct the stochastic gradient estimate (please see Algorithm 1 and Theorem 3.1 in the paper). For the experiments, we set $w_t = 1$, $\sigma^2 = 1$ and $c = \bar{c}/\bar{\kappa}^2$ and tune for $\bar{\kappa} \in \\{10^1, 10^0, 10^{-1}, 10^{-2} \\}$ for Shakespeare dataset and in the range $\bar{\kappa} \in  [10^{-1}, 10^{-2} ]$ for CIFAR-10 dataset. For both the datasets, we tune for $\bar{c}$ in the range $[1,10]$.
> > >
> > > - For rest of the algorithms, we choose tune the step-size from the set $\\{10^1, 10^0, 10^{-1}, 10^{-2} \\}$.
> > >
> > > Note that for each of the algorithms, its parameters are selected either using what has been suggested in their respective papers and/or the best performance among the hyperparameters selection strategy outlined above. We believe that we did our best to present a fair comparison between all the algorithms, given the time and computational resource constraints in the response period. Moreover, we plan to release the codes with complete implementation in the future.
> > >
> > > About the constant, let us emphasize that we have carefully checked their choice, and we believe that they can be only marginally improved in the analysis leading to no change in the theoretical convergence guarantees. Moreover, we have chosen the constants to keep them, exponents of $2$, whenever possible.
> > >
> > > Finally, we would like to point out that the paper’s goal was to develop an algorithm that achieves the best theoretical performance guarantees for FL algorithms. In addition to the theoretical contributions, we have conducted extensive experiments to show that the algorithm in fact works well in practice.  We have also followed the reviewer's advice to further improve the quality of the numerical results. We hope that our response above can further clarify the reviewer's doubts, but we also fully understand if the review still cannot vote for clear acceptance (score "7"). However, it was a bit not clear to us if the reviewer meant to raise the score to "6" (weak acceptance) rather than ``5" (weak reject). Thank you for your time and valuable suggestions.
> > >
> > > > I have a few additional questions:
> > >
> > > > **Your comment** 3. Why you did not compare with the methods proposed in the work [1]?
> > >
> > > **Our response:** The work in [1] considers adaptive algorithms, for a fair comparison we will compare the works in [1] with the adaptive version of our algorithm. We are currently in the process of extending our algorithm which supports adaptive implementation.
> > >
> > > > **Your comment** 4. Could you please clarify what version of the MIME algorithm did you use in your experiments?
> > >
> > > **Our response:** In our experiments we included, we have used MimeLite which practically has the same performance as MIME.
> > >
> > > > **Your comment** 5. What is meant by "near-optimal complexity"? I think it should be mentioned in the paper.
> > >
> > > **Our response:** We have discussed the “near-optimal complexity” in Remark 1 of the paper.
> > >
> > > [1] R. Sashank, et al. "Adaptive federated optimization."

---

> > > > ### Comment · Reviewer_9RXX · 2021-08-29
> > > > **Concluding comment**
> > > >
> > > > Thanks for your response and provided details of the experimental evaluation.
> > > >
> > > > I agree that given the obtained clarifications and improved experimental results, I can increase my score from "5" (the consequence of my conservatism) to "6". Nevertheless, in my view, the theoretical contributions of this work are not significant enough (due to *improved* rates in the case of parameters depending on $\epsilon$) to guarantee acceptance on its own. This is why I changed my evaluation due to extended experimental results showing the benefits of the proposed approach. If it was done right from the beginning, the discussion would have been different.

---

> > > > > ### Author Response · Authors · 2021-09-01
> > > > > **Clarification of parameter dependence on \epsilon**
> > > > >
> > > > > **Our response:** We thank the reviewer for his/her valuable time and suggestions. We would like to point out that it is standard for the parameters of an FL algorithm, e.g., the number of local updates and/or the batch sizes, to depend on the desired solution accuracy $\epsilon$. To the best of our knowledge, the algorithms that achieve near-optimal communication complexity of $\mathcal{O}(\epsilon^{-1})$ either require the number of local updates or the batch-sizes that depend on the solution accuracy $\epsilon$.
> > > > >
> > > > > > - For example, FedProx [1], FedPD [2] and FedDyn [3] rely on solving the "local problems" to achieve $\epsilon$-accuracy, which implies that the number of local updates (or the batch sizes) implicitly depends on the desired solution accuracy $\epsilon$. Specifically, if we use vanilla SGD as the local solver and the "local problem" at each client is strongly convex, then FedProx, FedPD, and FedDyn require $\mathcal{O}(\epsilon^{-1})$ local updates/gradient evaluations before each communication round to solve the (local) problem to $\epsilon$-accuracy.
> > > > >
> > > > > > - Similarly, as shown in [4] and [5] the communication complexity of FedAvg and its momentum version can be improved from $\mathcal{O}(\epsilon^{-2})$ to $\mathcal{O}(\epsilon^{-3/2})$ when the number of local updates (or batch size) is chosen as  $\mathcal{O}(\epsilon^{-1/2})$ (please see Section 3.2 of the paper for a detailed discussion).
> > > > >
> > > > > Moreover, we note that even without the local updates and the batch-sizes depending on $\epsilon$ (with $b$ and $I$ are both chosen as $\mathcal{O}(1)$), STEM achieves optimal sample complexity of $\mathcal{O}(\epsilon^{-3/2})$ and the same communication complexity of $\mathcal{O}(\epsilon^{-3/2})$.
> > > > >
> > > > > In the revised version of the paper, we will add a discussion regarding the dependence of parameters on $\epsilon$.
> > > > >
> > > > > - [1] Li et al., Federated optimization in heterogeneous networks, 2018.
> > > > > - [2] Zhang et al., Fedpd: A federated learning framework with optimal rates and adaptivity to non-iid data, 2020.
> > > > > - [3] Acar et al, Federated Learning Based on Dynamic Regularization, 2020.
> > > > > - [4] Yu et al., Parallel restarted SGD with faster convergence and less communication: Demystifying why model averaging works for deep learning,” 2018.
> > > > > - [5] Yu et al., On the linear speedup analysis of communication efficient momentum SGD for distributed non-convex optimization,” 2019.

---

### Official Review · Reviewer_2QZ9 · 2021-07-17

**Rating:** 5
**Confidence:** 4

**Summary:**

This paper studies non-convex stochastic optimization for federated learning under the mean-squared smoothness assumption and shows that better sample convergence and communication complexity are possible with large batch sizes.

**Limitations And Societal Impact:**

Yes

**Main Review:**

To me, this is a borderline paper. This paper has its technical merit by establishing stronger sample/communication complexity bounds. However, there seem to be some fundamental issues on the interpretation correctness and presentation clarity.

The proposed Algorithm 1 is named " Stochastic Two-Sided Momemtum (STEM)", which also appears in the paper title. It is named so because the authors interpret line 9 in Algorithm 1 as "server side momentum step" and argue its novelty and importance for better convergence. However, I don't think it is necessary to let the server do the momentum SGD update in line 9.  We could let each client do local momentum SGD updates in line 8 and then simply average all local x at line 9.  With such an equivalent description,there is no "server side momentum" concept needed. It is really hard for me to appreciate the necessity of server side momentum. In fact, I think line 7-10 is mathematically equivalent to " updating each local node with momentum SGD at each iteration and average all local x at the server when t mod I = 0", which is just the standard "FedAvg with momentum SGD local solver "or "local SGD with momentum".  In this regard, it seems this paper does not propose a new algorithm but just analyzes the existing FedAvg with momentum SGD local solvers under the stronger Assumption 1.  I would also suggest the authors revisit their proof and comment if the proof also follows similar lines as in early works.

Table 1 lists the complexity of earlier works that can be misleading to general readers who do not have sufficient optimization backgrounds. The current comparison is not apple to apple since the complexities are attained under different technical assumptions. Please add an extra column to list the needed assumption for each row.

The idea of using large batch sizes to achieve faster convergence may let practitioners frown since large batch sizes may hurt the final model accuracy. In the experiment, the methods are only tested with b=8 or b=64. (These settings are usually considered as O(1) batch size.) More experiments with larger b might be desired.







**Time Spent Reviewing:**

3

---

> ### Author Response · Authors · 2021-08-10
> **Technical Merit**
>
> >**Your Comment:** 1. To me, this is a borderline paper. This paper has its technical merit by establishing stronger sample/communication complexity bounds. However, there seem to be some fundamental issues on the interpretation correctness and presentation clarity.
>
> **Our Response:** We thank the reviewer for the comments, below we address the comments in a point-by-point manner.
>
> > **Your Comment:** 2. The proposed Algorithm 1 is named "Stochastic Two-Sided Momemtum (STEM)", which also appears in the paper title. It is named so because the authors interpret line 9 in Algorithm 1 as "server side momentum step" and argue its novelty and importance for better convergence. However, I don't think it is necessary to let the server do the momentum SGD update in line 9. We could let each client do local momentum SGD updates in line 8 and then simply average all local x at line 9. With such an equivalent description, there is no "server side momentum" concept needed. It is really hard for me to appreciate the necessity of server side momentum. In fact, I think line 7-10 is mathematically equivalent to " updating each local node with momentum SGD at each iteration and average all local x at the server when t mod I = 0", which is just the standard "FedAvg with momentum SGD local solver "or "local SGD with momentum". In this regard, it seems this paper does not propose a new algorithm but just analyzes the existing FedAvg with momentum SGD local solvers under the stronger Assumption 1. I would also suggest the authors revisit their proof and comment if the proof also follows similar lines as in early works.
>
>
> **Our response:** Thank you for the insightful comment. Please see our response below.
>
> First, we agree with the reviewer that the server-side momentum can be done at the client as well. However, to keep the method general and allow the server to have the freedom to choose the learning rate independently of the clients in practice, we have chosen to present the algorithm in the current form. In this way, the presentation is consistent with a line of recent works in FL, which utilizes *two-sided* learning rate for FL algorithms, where both the clients and server perform gradient-based updates [4,5]. In the revised work, we will make the above relation clear.
>
> Nevertheless, we would like to emphasize that, no matter how the algorithm is presented, the underlying analysis remains challenging. We agree that the algorithm uses local steps and combines the iterates similar to FedAvg with Momentum SGD solvers. However the resulting algorithm is still new, and to our knowledge, there has been no analysis for such an algorithm. Intuitively, to see why this kind of Momentum SGD version of FedAvg is not easy to analyze, we can compare the relation between the vanilla SGD  and the SGD version of FedAvg. Despite the fact that vanilla SGD is relatively easy to analyze, the FedAvg with SGD as a local solve is challenging, and analyzing this kind of algorithm has attracted tremendous research efforts in recent years [1, 2 ,3].  The main culprit is that, when data heterogeneity is present, and when we allow the clients to perform multiple local SGD updates, then the clients' updates can deviate significantly from the true SGD updates, and the occasional simple averaging cannot ensure that iterates will not deviate again.
>
> Finally, let us mention that, beyond analyzing the communication/sample complexity for the proposed algorithm, we also provide guidelines for choosing batch sizes and the number of local updates such that the optimal performance of STEM can be maintained. Such kind of optimal trade-off between batch sizes and the number of local updates is unique to the FL setting (since for conventional SGD applied to our setting, the clients do not perform multiple local updates), and to our knowledge, this is the first time that it has been discovered.
>
> $[1]$ McMahan et al, Communication-efficient learning of deep networks from decentralized data
>
> $[2]$ Yu et el, On the linear speedup analysis of communication efficient momentum sgd for distributed non-convex optimization
>
> $[3]$ Woodworth et al, Minibatch vs Local SGD for Heterogeneous Distributed Learning
>
>  $[4]$ Reddi et al, Adaptive Federated Optimization
>
>  $[5]$ Karimireddy et al, Scaffold: Stochastic controlled averaging for on-device federated learning
>
> > **Your Comment:** 3. Table 1 lists the complexity of earlier works that can be misleading to general readers who do not have sufficient optimization backgrounds. The current comparison is not apple to apple since the complexities are attained under different technical assumptions. Please add an extra column to list the needed assumption for each row.
>
> **Our Response:** We thank the reviewer for bringing up this point. In the original paper, we have already included some of the details regarding assumptions of the corresponding algorithms in the caption. Based on the reviewer's suggestion we will move the assumptions in the caption of the table and include them as a separate column in the Table.
>
> > **Your Comment:** 4. The idea of using large batch sizes to achieve faster convergence may let practitioners frown since large batch sizes may hurt the final model accuracy. In the experiment, the methods are only tested with $b=8$ or $b=64$. (These settings are usually considered as $O(1)$ batch size.) More experiments with larger $b$ might be desired.
>
> **Our Response:**  Thank you for your comment. We first note that STEM does not necessarily require large batch sizes to achieve the optimal sample and communication complexity. This follows from Theorem 1, and Remark 2, that if each node uses $O(1)$ batch size we can increase the number of local updates such that the (near) optimal communication complexity of STEM is maintained. Moreover, note that even if we choose an $\mathcal{O}(1)$ number of local updates with $\mathcal{O}(1)$ batch size per update, our algorithm achieves the sample and communication complexities of $\mathcal{O}(\epsilon^{-3/2})$, which match the performance of MIME and FedGLOMO. This result comes from our trade-off analysis discussed in Remark 3 of the original paper.
>
>  Additionally, we have also evaluated the performance of the algorithm with additional batch-size choices; see Table R1-R3 in the **General Response**. We have also conducted a comparison of STEM with other algorithms for the Shakespeare dataset; see Table R4 in the **General Response** section at the beginning of this response.

---

### Official Review · Reviewer_pfmU · 2021-07-22

**Rating:** 6
**Confidence:** 3

**Summary:**

This paper introduced a two-step momentum method for federated learning. The proposed method achieves the best achievable sample and communication complexities for first-order stochastic FL algorithms. The theoretical analysis provides guidance on selecting synchronization interval and batch size for achieving the nearly optimal complexities.

**Limitations And Societal Impact:**

Yes

**Main Review:**

This paper introduced an interesting idea for achieving near-optimal sample and communication complexities for federated learning. Overall, the paper is well written, but I would suggest authors explicitly state Definitions 2.1 and 2.2 before referencing them. More detailed comments are shown as follows

1. Why do we need to use the consecutive iterates x^k_{t+1} and x^k_t to evaluate the estimator? Can't we just use x^k_{t+1}? How does it affect the complexity bounds?

2. As pointed out by (Khaled, Ahmed, Konstantin Mishchenko, and Peter Richtárik, 2019), a bound on the dissimilarity is not realistic for heterogeneous data setting. Under this assumption, the data is very close to i.i.d.

3. The experiments seems weak in terms of model selection and data scale. Federated EMNIST dataset is a better benchmark data for federated learning (Caldas, Sebastian, et al. 2018).

Khaled, Ahmed, Konstantin Mishchenko, and Peter Richtárik. "First analysis of local gd on heterogeneous data." arXiv preprint arXiv:1909.04715 (2019).
Caldas, Sebastian, et al. "Leaf: A benchmark for federated settings." arXiv preprint arXiv:1812.01097 (2018).

**Time Spent Reviewing:**

1

---

> ### Author Response · Authors · 2021-08-10
> **Clarifications and Additional Experiments**
>
> > **Your Comment:**. This paper introduced an interesting idea for achieving near-optimal sample and communication complexities for federated learning. Overall, the paper is well written, but I would suggest authors explicitly state Definitions 2.1 and 2.2 before referencing them. More detailed comments are shown as follows
>
> **Our Response:** We thank the reviewer for the positive comment. In the revised version of the paper, we will move Definitions 2.1 and 2.2 at the start before they are being referenced.
>
> > **Your Comment:** 1. Why do we need to use the consecutive iterates $x^k_{t+1}$ and $x^k_t$ to evaluate the estimator? Can't we just use $x^k_{t+1}$? How does it affect the complexity bounds?
>
> **Our Response:** We note that the use of two iterates to construct the stochastic gradient helps improve the performance of non-convex optimization algorithms by reducing the variance of the gradient estimator iteratively. Specifically, it is well known that $\mathcal{O}(\epsilon^{-2})$ performance of vanilla SGD can be improved (under additional assumptions, please see Assumption 1 in the paper) via the use of variance reduced gradient estimates [1,2,3,4,5], evaluated using two iterates. Note that with gradient evaluated using a single iterate, the only way to reduce the variance is by increasing the batch size, however, such increased batch size does not lead to improved sample complexity.  Therefore, to improve the performance of vanilla SGD algorithms (evaluated using only a single iterate) it is necessary to utilize two stochastic gradients evaluated at two different iterates. Please see Lemma C.6 in the appendix, where we show how the gradient error contracts with the use of the gradient estimator given in Step 6 of Algorithm 1. Moreover, note that the gradient estimator used for STEM is equivalent to FedAvg with the choice of momentum parameter $a_t = 1$, which utilizes only $x_{t+1}^{(k)}$ for the gradient estimator.  Please see Algorithm 2 and Section 3.2 in the paper for the detailed sample and communication complexities for the generalized FedAvg algorithm.
>
> $[1]$ Reddi et al, Stochastic Variance Reduction for Nonconvex Optimization
>
> $[2]$ Lei et al, Nonconvex Finite-Sum Optimization Via SCSG Methods
>
> $[3]$ Fang et al, Spider: Near-optimal non-convex optimization via stochastic path-integrated differential estimator
>
> $[4]$ Cutkosky et al, Momentum-based variance reduction in non-convex SGD
>
> $[5]$ Dihn et al, Hybrid stochastic gradient descent algorithms for stochastic nonconvex optimization
>
> >**Your Comment:** 2.  As pointed out by (Khaled, Ahmed, Konstantin Mishchenko, and Peter Richtarik, 2019), a bound on the dissimilarity is not realistic for heterogeneous data setting. Under this assumption, the data is very close to i.i.d.
> >
> >Khaled, Ahmed, Konstantin Mishchenko, and Peter Richtárik. "First analysis of local gd on heterogeneous data." arXiv preprint arXiv:1909.04715 (2019).
>
> **Our Response:** We thank the reviewer for raising this concern. The gradient dissimilarity assumption mentioned by the reviewer, i.e., Assumption 2 in the paper,  is given by:
> $$||\nabla f^{(k)}(x) - \nabla f^{(\ell)}(x)||^2 \leq \zeta^2 \quad \forall x, y \in \mathbb{R}^d,  \forall k, \ell \in [K]..$$
> We agree that this assumption is rather strong. In Ahmed et al, the authors consider convex problems, and utilized the following measure for data heterogeneity:
> $$ \frac{1}{K} \sum_{k=1}^K ||\nabla f^{(k)} (x^\ast) ||^2 = \zeta^2, \quad (1)$$
> where $x^\ast$ is a minimizer of $f (x) = \frac{1}{K} \sum_{k = 1}^K f^{(k)}(x)$. The authors state that for strongly convex problems (and convex problems) the above assumption is milder than the bounded dissimilarity assumption used for STEM (Assumption 2), as the bounded dissimilarity assumption cannot be satisfied for strongly convex problems. We would like to point out that the work Ahmed et al, considers a convex deterministic optimization problem which is not directly comparable to our work which considers a stochastic non-convex optimization problem. Moreover, for non-convex problems a non-i.i.d assumption similar to (1) is adopted [1], stated as
> $$
> \frac{1}{K} \sum_{k = 1}^K  || \nabla f^{(k)}(x)||^2 \leq B^2 ||\nabla f(x)||^2 \quad \forall x \in  \\{ x \in \mathbb{R}^d |~ ||\nabla f(x)||^2 > \epsilon \\}
> $$
> we note that this assumption is much stronger than the bounded dissimilarity assumption in the paper (for details please see discussion in FedPD [2]).
>
> In summary, we agree with the reviewer that the gradient dissimilarity assumption used in our work is a strong assumption, but it is still the most popular assumption used for describing data heterogeneity for non-convex FL algorithms; for example, please see [3,4,5].
>
>
> $[1]$ Li et al, Federated optimization in heterogeneous networks
>
> $[2]$ Zhang et al, Fedpd: A federated learning framework with optimal rates and adaptivity to non-iid data
>
> $[3]$ Yu et el, On the linear speedup analysis of communication efficient momentum sgd for distributed non-convex optimization
>
> $[4]$ Das et al, Improved convergence rates for non-convex federated learning with compression
>
> $[5]$ Woodworth et al, Minibatch vs Local SGD for Heterogeneous Distributed Learning
>
> >**Your Comment:** 3. The experiments seems weak in terms of model selection and data scale. Federated EMNIST dataset is a better benchmark data for federated learning (Caldas, Sebastian, et al. 2018).
> >
> >Caldas, Sebastian, et al. ``Leaf: A benchmark for federated settings." arXiv preprint arXiv:1812.01097 (2018).
>
>   **Our Response:** Thank you for the helpful comment. In the revised manuscript, we have conducted additional experiments on the Shakespeare dataset from the LEAF repository. We note that the Shakespeare dataset is much bigger in scale and considers a relatively difficult classification problem compared to EMNIST. Moreover, we have compared the performance of our algorithm with many state-of-the-art algorithms including FedProx, MIME, FedGLOMO, FedDyn, SCAFFOLD, and FedAvg. The summary of the results are included in the **General Response** section at the beginning of the response.

---

### Official Review · Reviewer_4cGF · 2021-08-23

**Rating:** 6
**Confidence:** 4

**Summary:**

The paper proposes a new method to solve the finite sum problem in federated learning where each local model is an expectation function. The STEM algorithm uses momentum variance-reduced estimator in local updates (similar to STORM estimator in stochastic optimization) and also send these estimators to the server when performing server-side update. Therefore, STEM includes momentum updates for both local and server updates. Convergence analysis reveals that STEM nearly match the best known sample complexity in centralized setting and it also matches the lower bound communication complexity up to a log factor. Numerical experiments on neural network training not only show advantage of STEM compared with FedAvg and SCAFFOLD but also presenting the tradeoff between communication frequency and minibatch size.

**Limitations And Societal Impact:**

Regarding limitations, I have not seen the authors state this clearly in the paper. I don't see any immediate negative societal impact of this work.

**Main Review:**

Pros:
- The paper introduces a new framework to incorporate the momentum gradient estimators in federated learing compared to 2 related works which are MIME and FedGLOMO. Given FedGLOMO is rather new, I consider it as concurrent work.
- Convergence analysis provides intuition on the tradeoff between communication and sample complexities (perform more local updates vs communicate more). This effect is further illustrated in the experiments
- Complexity results are also new.
- The analysis of number of local updates and minibatch size

Cons:
- The paper presents FedAvg with full-device participation and compare the results with this version. However, FedAvg or SCAFFOLD do allow partial-participation in their convergence analysis while STEM seems to require full device participation.
- Apart from the local model, STEM also sends "local momentum" which is an estimate of the local gradient. This framework does not exactly match the federated learning framework described in [https://arxiv.org/pdf/1610.05492.pdf, page 2] where local nodes can only send local model back to the server. Moreover, these gradient estimates are susceptible to privacy attack in federated learning [https://proceedings.neurips.cc/paper/2020/file/c4ede56bbd98819ae6112b20ac6bf145-Paper.pdf]. Although this is not exactly the true gradient, but it is still a good estimate (if not then the algorithm is not working well) so it is good to avoid communicating the first-order information. I am aware that the same criticism can be applied to SCAFFOLD, MIME or FedGLOMO.

Minor comments:
- It is better to elaborate the steps from (b) to (c) for the second term at line 504 similar to (9) at line 523 for clarity.
- There should be a $\nabla f^{(k)}(x^{(k)}_t;\xi_t^{(k)})$ at line 561.

**Time Spent Reviewing:**

7

---

### Author Response · Authors · 2021-08-10
**General Response**

We thank the reviewers for taking their valuable time to provide insightful comments and suggestions for the paper. We believe that the thoughtful reviews and the recommendations made by the reviewers have substantially improved the quality of the paper. Based on reviewers' suggestions we have conducted additional numerical experiments to compare the performance of STEM to other algorithms under different settings. In this response, we include a summary of the results obtained. We will provide the complete details of parameter selection and algorithm implementation in the revised manuscript.

We evaluate the performance of STEM against multiple algorithms for (i) different heterogeneity settings, (ii) multiple batch size choices and, (iii) two different datasets, specifically, in addition to CIFAR-10 dataset, we also include additional results for the Shakespeare dataset [1]. Since we have additionally included FedDyn [2] in our results, we choose not to include FedPD since they are known to be very closely related.

### Additional algorithms and heterogeneity setting

In Tables R1, R2, and R3 we present the results for different algorithms for CIFAR-10 dataset under different heterogeneity and batch-sizes choices (note, in Table R1 and R2, newly included algorithms are highlighted using bold letters; Table R3 is completely new since it represents a new high heterogeneous setting). Note that STEM performs uniformly well under all the conditions. Moreover, note from Table R2 that FedGLOMO diverges for the case when the number of local updates is high. Also, note from Table R3 that when the heterogeneity is high, FedProx performs well while SCAFFOLD's performance degrades significantly.

> **Table R1**  CIFAR-10 data is randomly split among nodes (moderate heterogeneity setting). We choose $b = 64$ and $I = 7$ after $500$ rounds of communication. Highlighted rows are new.\
> **Algorithm**  $\quad$   **Training Accuracy(%)** $\quad$    **Test Accuracy(%)**
>---
>FedAvg    $~\qquad \qquad \quad$       78.2     $\qquad \qquad \quad \qquad$                74.1
>
> **FedProx** $\qquad ~~~\quad \quad$   79.2 $\qquad \qquad \quad \qquad$  74.8
>
> **FedDyn** $~~\qquad ~~\quad \quad$  68.9 $\qquad \qquad \quad \qquad$  66.0
>
>SCAFFOLD $\quad \quad \qquad$  71.9 $\qquad \qquad \quad \qquad$  74.0
>
> **MIME** $~~~\qquad \qquad \quad $  82.6 $\qquad \qquad \quad \qquad$  76.8
>
> **FedGLOMO** $~~\quad \quad \quad $  76.1 $\qquad \qquad \quad \qquad$  72.8
>
>STEM $~~~~\qquad  \quad \qquad$  80.1$\qquad \qquad ~\quad \qquad$  78.8


> **Table R2**  CIFAR-10 data is split among nodes such that each node can only access $5$ classes (high heterogeneity setting). We choose $b = 8$ and $I = 61$ after $500$ rounds of communication. Highlighted rows are new.\
> **Algorithm**  $\quad$   **Training Accuracy(%)** $\quad$    **Test Accuracy(%)**
>---
>FedAvg    $~\qquad \qquad \quad$       73.6     $\qquad \qquad \quad \qquad$                75.4
>
> **FedProx** $~~~\quad \qquad \quad$   80.0 $\qquad \qquad \quad \qquad$  75.2
>
> **FedDyn** $~~\qquad ~~\quad \quad$  76.1 $\qquad \qquad \quad \qquad$  71.3
>
> SCAFFOLD $\quad \quad \qquad$  72.5 $\qquad \qquad \quad \qquad$  73.7
>
> **MIME** $~~~\qquad \qquad \quad $  61.5 $\qquad \qquad \quad \qquad$  58.6
>
> **FedGLOMO** $~~\quad \quad \quad $  10.0 $\qquad \qquad \quad \qquad$  10.0
>
>STEM $~~~~\qquad  \quad \qquad$  81.1 $\qquad \qquad \quad \qquad$  78.5


> **Table R3**  CIFAR-10 data is split among nodes such that each node can only access $2$ classes (extreme heterogeneity setting). Training and testing accuracy for $b = 128$ and $I = 6$ after $300$ rounds of communication. All the results in this table are new.\
> **Algorithm**  $\quad$   **Training Accuracy(%)** $\quad$    **Test Accuracy(%)**
>---
>FedAvg    $~\qquad \qquad \quad$       57.6     $\qquad \qquad \quad \qquad$               57.1
>
>FedProx $\qquad \qquad \quad$   59.1  $\qquad \qquad \quad \qquad$  58.5
>
>FedDyn $~~~\qquad ~~\quad \quad$  51.2 $\qquad \qquad \quad \qquad$  51.3
>
>SCAFFOLD $\;\quad \quad ~~~\quad$  40.3 $\qquad \qquad \quad \qquad$  41.3
>
> MIME $~~~~\qquad \qquad \quad $  56.1 $\qquad \qquad \quad \qquad$  55.1
>
>FedGLOMO $~~~\quad \quad \quad $  56.8 $\qquad \qquad \quad \qquad$  56.1
>
>STEM $~~~~\qquad  \quad \qquad$  58.5  $\qquad \qquad \quad \qquad$  57.4

### Federated learning Benchmarking Dataset

In Table R4, we evaluate the performance of STEM against other algorithms for the task of next-character prediction on the Shakespeare dataset [1]. Note that the Shakespeare dataset is a popular benchmarking dataset for FL algorithms. We choose the number of local updates as the number of updates required to complete a single pass over each node's data. We present the training and testing accuracy in Table R4. Note from Table R4 that STEM outperforms the rest of the algorithms. We also note that because of time and computation constraints we were unable to average the experiments over multiple runs. In the revised version of the manuscript, we will average the performance of all the algorithms over multiple runs for the Shakespeare dataset and report the performance with error bars. Moreover, we will include the details of all the parameter tuning and experiment settings in the updated manuscript.


> **Table R4**  Performance comparison for Shakespeare dataset with $b = 128$. We report the performance after $50$ rounds of communication. All the results in this table are new.\
> **Algorithm**  $\quad$   **Training Accuracy(%)** $\quad$    **Test Accuracy(%)**
>---
>FedAvg    $~\qquad \qquad \quad$       40.1    $\qquad \qquad \quad \qquad$                39.2
>
>FedProx $\qquad ~~~\quad \quad$   43.5  $\qquad \qquad \quad \qquad$  43.2
>
>FedDyn $~~\qquad ~~\quad \quad$  43.7 $\qquad \qquad \quad \qquad$  43.2
>
>SCAFFOLD $\quad \quad \qquad$  40.3 $\qquad \qquad \quad \qquad$  41.3
>
> MIME $~~~\qquad \qquad \quad $  32.1 $\qquad \qquad \quad \qquad$  32.1
>
>FedGLOMO $~~\quad \quad \quad $  40.3 $\qquad \qquad \quad \qquad$  40.1
>
>STEM $~~~~\qquad  \quad \qquad$  44.5 $\qquad \qquad \quad \qquad$  43.8

$[1]$ Caldas et al, LEAF: A Benchmark for Federated Settings

$[2]$ Acar et al, Federated Learning Based on Dynamic Regularization

---

### Comment · Area_Chair_N4BB · 2021-09-24
**Additional Comments from Area Chairs and Senior Area Chair Discussion**

--Given that $O(1/\epsilon^{3/2})$ sample complexity has been achieved in previous work on federated learning ([17,18]), this paper seems to trade the mini-batch size (or the number of local updates) for better communication complexity, but the overall sample complexity is no better than the best existing results.  In order to achieve $O(1/\epsilon)$ communication complexity, the proposed algorithm needs $O(1/\epsilon^{0.5})$ mini-batch size or $O(1/\epsilon^{0.5})$ number of local updates, or some combination in-between, this may not be feasible, if some clients do not have that many data points, or if each local update is energy-consuming.

--The proposed algorithm needs all the clients to be updated in each round, and this is less interesting in the most general federated learning setting, where only a subset of clients is updating their local models.

---

### Decision · Program_Chairs · 2021-09-28

**Decision:**

Accept (Poster)

**Comment:**

This is a borderline paper. The reviewers agree that the communication complexity and sample complexity results are new in the federated learning setting, but there are questions on the novelty of the algorithm as the main ideas and techniques (including the complxities) are adapted from existing variance-reduced estimators for stochastic optimization. The are also concerns on additional vectors to communicate, the large local batch-size or number of iterations, and the requirement of full participation. After discussion between the reviewers and assessment of two Area Chairs and senior AC, we think the results are not significant enough and have to recommend rejection.

**Consistency Experiment:**

NeurIPS has a long history of experimentation. In 2014, NeurIPS ran an experiment in which 10% of submissions were reviewed by two independent committees to quantify the randomness in the review process. This year, we repeated a variant of this experiment to see how the quality of the review process has changed over time.  This paper was part of the experiment and was therefore assigned to two committees (consisting of reviewers, an Area Chair, and a Senior Area Chair) that reached independent decisions.  If both committees made the same recommendation, this recommendation was followed. If a single committee recommended acceptance, the paper was accepted (with the exception of a few cases in which the other committee identified what we considered a fatal flaw, e.g., an error in a key result).

This copy’s committee reached the following decision: **Reject**

The other committee assigned to the paper recommended **Accept (Poster)**.  You can find the other set of reviews, along with any follow up discussion with the authors here:
https://openreview.net/forum?id=J28lNO4p3ki